# Spatio-temporal transcriptomics of chromothriptic SHH-medulloblastoma identifies multiple genetic clones that resist treatment and drive relapse

Paediatric medulloblastomas with chromothripsis are characterised by high genomic instability and are among the tumours with the worst prognosis. However, the molecular makeup and the determinants of the aggressiveness of chromothriptic medulloblastoma are not well understood. Here, we apply spatial transcriptomics to profile a cohort of 13 chromothriptic and non-chromothriptic medulloblastomas from the same molecular subgroup. Our data reveal a higher extent of spatial intra-tumour heterogeneity in chromothriptic medulloblastomas compared to non-chromothripictic tumours, which is associated with increased proliferation and stemness, but lower immune infiltration and differentiation. Spatial mapping of genetic subclones of the same tumour identify a regionally distinct architecture and clone-specific phenotypic features, with distinct degrees of differentiation, proliferation and immune infiltration between clones. We conduct temporal profiling of 11 samples from patient-derived xenografts from a patient with chromothriptic medulloblastoma, covering the transition from the minimal residual disease stage to treatment-resistant regrown tumours. In chromothriptic medulloblastoma, an ecosystem of cells from multiple genetic clones resist treatment and lead to relapse. Finally, we identify tumour microtubes in chromothriptic medulloblastoma, calling for exploration of cell network communication as a putative target.

Medulloblastomas with chromothripsis, a form of genome instability leading to massive genome rearrangements, are incurable paediatric brain tumours. Children with Li Fraumeni syndrome (LFS, germline variant in *TP53*, OMIM Entry # 151623) who develop such tumours have the worst prognosis from all molecular subgroups of medulloblastoma, with a five-year overall survival rate of 27%[1]. The unmet clinical need highlighted by this dismal clinical outcome calls for a better understanding of the mechanisms of resistance, to lay the basis for future therapy development. Thought to occur in a single catastrophic event, chromothripsis results in massive, complex chromosomal rearrangements that are observed in close to all LFS medulloblastomas[2]. In the context of germline variant in *TP53*, the risk of secondary tumours induced by treatment-related DNA damage further complicates therapy selection[3]. Affected children frequently develop relapsed tumours, and eventually die from the relapse in most cases[4]. Despite recent efforts[5–12], the molecular basis of the aggressiveness of LFS medulloblastoma remains largely unknown. In particular, the specific cell populations and the role of different genetic clones in the formation of relapse tumours have not been comprehensively characterised.

✉ e-mail: o.stegle@dkfz.de; a.ernst@dkfz.de

Spatial transcriptomics analyses in cancer have emerged as a powerful tool to dissect the complex biology of human malignancies, and in particular open up the possibility to study how tumour cell populations interact and organise across the tissue landscape within their microenvironment[13,14]. In the normal cerebellum, spatial transcriptomics analyses of genetic mouse models[15] have been used to characterise the tissue architecture. However, human medulloblastomas, and in particular the highly aggressive subgroup of LFS medulloblastomas, have not been characterised by spatially resolved transcriptomics.

Beyond a deeper knowledge of the molecular basis of patient tumours, patient-derived xenograft (PDX) models are becoming an essential tool in oncology[16]. Pre-clinical studies in PDX have led to the identification of novel therapeutic strategies in medulloblastoma[17–19]. However, the lack of efficient treatment for relapsed tumours highlights the need to better characterise resistant cell populations and to leverage spatial and temporal profiling. PDX models provide the unique possibility to locate residual brain tumour cells after treatment and to study the kinetics and the spatial organisation underlying treatment resistance.

In this study, we seek to achieve a better understanding of the aggressiveness of chromothriptic medulloblastomas by harnessing molecular signatures obtained from the local tissue architecture and microenvironment in both patient tumours and PDX models. To gain insights into the aggressiveness of LFS medulloblastomas, we generate comparative high-resolution maps of chromothriptic medulloblastomas as well as their non-chromothriptic counterparts from the same molecular subgroup, namely SHH ($n = 13$ SHH medulloblastomas, all untreated at diagnosis except for one matched pair of diagnostic and post-treatment relapse biopsies, as well as patient-derived xenografts). Our analysis provides a characterization of the spatial composition of malignant as well as non-malignant cell types and identifies critical cell communication networks in these tumours.

## Results

### Molecular and clinical features of the chromothriptic and non-chromothriptic medulloblastoma cohorts

In order to identify molecular features that potentially contribute to the aggressiveness of LFS medulloblastoma, we applied spatial transcriptomics (Visium, 10x Genomics) to a cohort of 13 medulloblastoma specimen (8 SHH-*TP53*mut medulloblastomas, 5 non-chromothriptic sporadic SHH-*TP53*wt medulloblastomas, abbreviated LFS1 to LFS8 and S1 to S5, respectively) and leveraged previously generated matched single-cell DNA and RNA sequencing data[12]. All tumours belonged to the SHH-medulloblastoma group, as confirmed by DNA methylation-based clustering of bulk methylomes using a reference cohort comprising all major molecular subgroups[20] (Supplementary Fig. 1A). DNA sequencing revealed germline variants in *TP53* in 8/8 LFS samples, as expected, but none in the SHH-*TP53*wt medulloblastomas (Supplementary Data 1). Sample S2 displayed a germline variant in *PTCH1*. As reported previously for SHH medulloblastoma[1,2], massive rearrangements due to chromothripsis were detected in all LFS cases, in stark contrast with the relatively small number of copy number variations detected in the SHH-*TP53*wt tumours from the control group (Supplementary Fig. 1B). Among the SHH-*TP53*wt medulloblastomas, four patients showed a loss of chromosome 9q including the *ELP1* and the *PTCH1* loci, an alteration which was previously associated with a good prognosis[21]. Pathological review of all tumours confirmed anaplastic or desmoplastic histology but did not identify any specific feature that might explain the aggressiveness of LFS (SHH-*TP53*mut) medulloblastomas as compared to SHH-*TP53*wt medulloblastomas (Supplementary Data 1, Supplementary Fig. 2). Hereafter, we refer to the SHH-*TP53*mut medulloblastomas as LFS or chromothriptic medulloblastomas and to the SHH-*TP53*wt medulloblastomas as non-chromothriptic or sporadic medulloblastomas, respectively.

### Higher transcriptional heterogeneity and distinct cell type composition in LFS medulloblastoma

We first set out to identify distinct spatially coherent tissue zones that are characterised by a shared expression profile within each tumour (using SpatialDE2[22]). This spatially aware clustering analysis revealed a markedly higher number of distinct tissue zones in LFS medulloblastomas, indicating a higher degree of spatial heterogeneity in this subtype (Fig. 1A–B, Supplementary Figs. 2–3). As alternative approaches, we also considered Leiden clustering (Fig. 1B, Supplementary Fig. 4) without using spatial information, as well as the quantification of the gene expression variance of individual genes that could be attributed to the spatial makeup (Supplementary Fig. 5), consistently supporting higher heterogeneity in LFS medulloblastomas as compared to sporadic medulloblastomas. Next, to gain insights into the cell-type composition of sporadic and LFS tumours, we applied Cell2location[23] to estimate cell type fractions at individual Visium locations based on reference cell-type signatures derived from matched scRNA-seq[12] (Fig. 1C, Supplementary Fig. 6). Importantly, normal cell types like macrophages, glial, endothelial, or meningeal cells were significantly more abundant in sporadic medulloblastoma as compared to LFS medulloblastoma, indicating subtype-specific differences in terms of their tumour microenvironment (Fig. 1C–D, Supplementary Fig. 6, Supplementary Data 2). In LFS medulloblastoma, we also observed limited infiltration of anti-tumour macrophages (distinguished from pro-tumour macrophages using established signatures, see Methods) (Fig. 2A–B, Supplementary Fig. 7A). To assess the extent to which these differences in cell type composition generalise, we conducted deconvolution of bulk RNA sequencing in a larger cohort ($n = 763$ medulloblastomas[5]). Specifically, we compared immune cell populations between the SHH alpha subtype, enriched for LFS medulloblastomas, and the remaining 11 molecular subtypes[5] but also between the SHH subgroup and the other three major molecular subgroups (WNT, group 3 and group 4). Consistent with the spatial transcriptomics data, this analysis revealed lower proportions of T-cells but higher proportions of pro-tumour macrophages in the SHH alpha subtype as compared to molecular subtypes with low LFS and low chromothripsis prevalence (Fig. 2C, Supplementary Fig. 7B). Altogether, these data point to a stronger immune exclusion in medulloblastomas from LFS patients as compared to other subtypes, even though medulloblastoma is a notoriously cold tumour type in general[24]. Notably, expression levels of canonical markers of immune cells correlated with survival outcome within the SHH alpha subtype enriched for LFS medulloblastomas (Fig. 2D, Supplementary Fig. 7C). To test whether lower immune infiltration might be a feature of *TP53* mutant tumours beyond medulloblastoma and possibly in the context of sporadic *TP53* mutations as well, we re-analysed a pan-cancer cohort of 8955 patients, again employing bulk deconvolution. This analysis indicated less CD8 positive T-cells in tumours displaying somatic mutations in *TP53* in a subset of tumour types, suggesting a possible link between p53 dysfunction and immune exclusion, beyond the context of genetic predisposition (Fig. 2E, Supplementary Fig. 7D, Supplementary Data 3–4).

We next investigated the spatial distribution of individual malignant cell types in LFS and sporadic medulloblastomas. Major cancer drivers as well as microenvironment-related genes such as proinflammatory factors showed higher expression levels in LFS medulloblastomas but were also characterised by increased spatial expression heterogeneity, with regions characterised by high expression of one gene but lower expression of the other, and vice versa (Fig. 3A, Supplementary Figs. 8–11, Supplementary Data 5). This heterogeneous pattern of expression of cancer drivers and microenvironment factors, together with differences of the overall expression levels, might contribute to tumour cells escaping the treatment in LFS medulloblastomas, with multiple drivers active in different tumour regions. In contrast to oncogenes, tumour suppressor genes such as *FSTL5* were

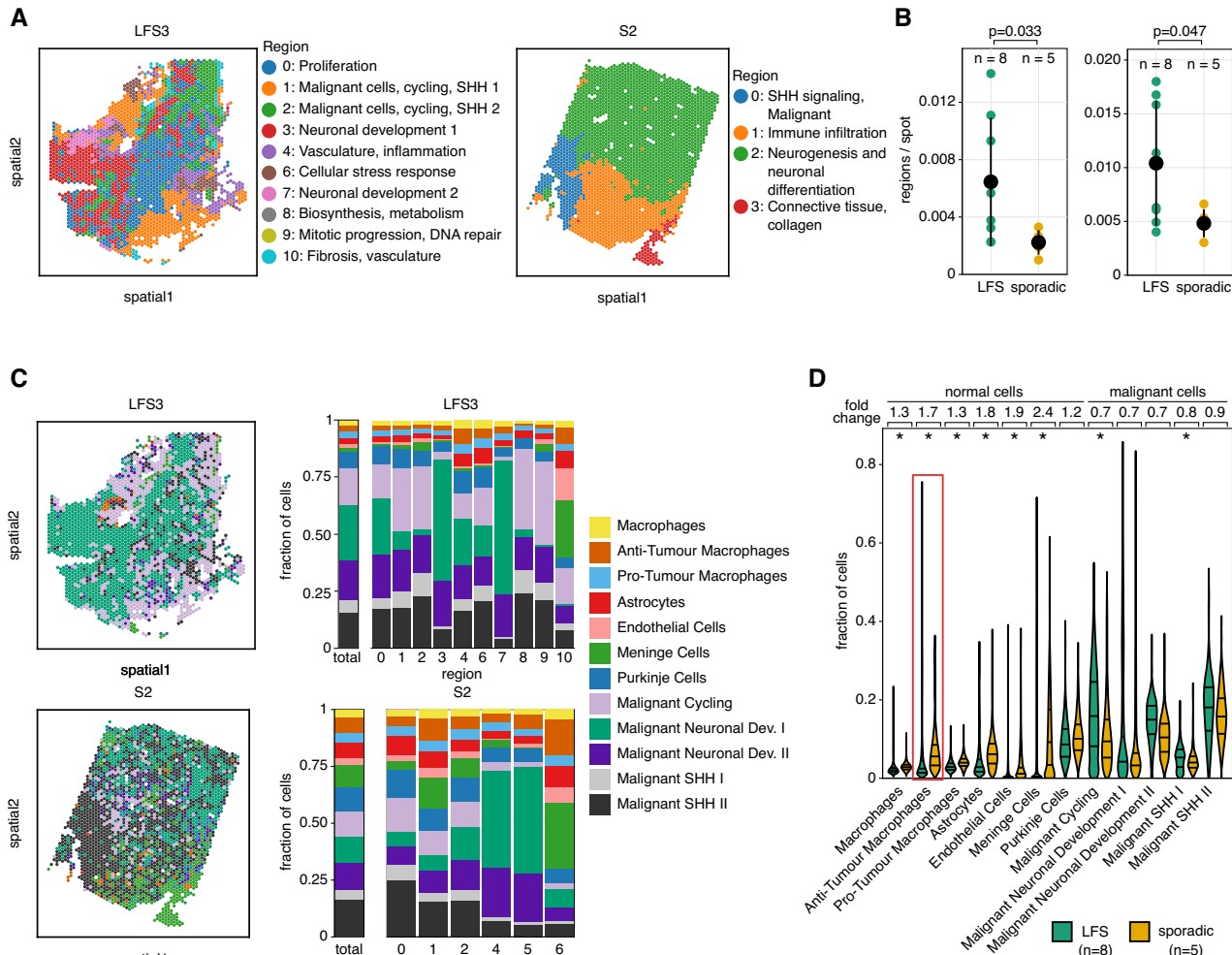

**Fig. 1 | Spatial transcriptomics reveals higher intra-tumour transcriptional heterogeneity in LFS medulloblastoma as compared to sporadic medulloblastoma. A** Spatial clustering of human medulloblastoma tissue Visium data identifies connected tissue regions with common transcriptional patterns. Shown are two representative samples of LFS (left, $n = 2469$ spots) and sporadic (right, $n = 2621$ spots) medulloblastoma. **B** Quantification of the number of regions normalised to the tissue size for all medulloblastomas as a measure of spatial heterogeneity (LFS, $n = 8$; sporadic, $n = 5$). Left: Regions were determined using SpatialDE2. Right: Regions were determined by Leiden clustering. Statistical significance was calculated using a one-sided Mann-Whitney U-test. Mean ± standard deviation per group is shown in black. **C** Comparative analysis of the cell type composition for a representative LFS (top, $n = 2469$ spots) and sporadic medulloblastoma (bottom, $n = 2621$ spots). Left, dominant cell type for each spot for representative LFS and sporadic medulloblastoma. Right, bar graphs displaying the cell type abundance for each region as in **A**. **D** Cell type abundance estimates for all medulloblastomas (LFS, $n = 8$; sporadic, $n = 5$). Mean fold change and the first, second, and third quartiles are indicated. *$p < 0.05$ (two-sided Mann-Whitney U-test, Benjamini-Hochberg multiple testing correction).

homogeneously lost in LFS medulloblastomas throughout the tumour tissue, but not in the sporadic medulloblastomas (Supplementary Figs. 8–10). Markers previously associated with putative medulloblastoma stem cells such as *SOX2*[25] showed significantly higher expression levels in LFS medulloblastomas, whereas differentiation markers were expressed at significantly lower levels (Supplementary Figs. 11 and 26, Supplementary Data 6). However, despite using the normal cell fractions as covariates, we cannot rule out that these differences are in part due to the different levels of expression of these genes in normal cells. Among the genes displaying a higher spatial variance in LFS as compared to sporadic medulloblastomas, angiogenesis factor *VEGFA* was prominent (Fig. 3A).

Three major malignant cell clusters in our dataset, both in LFS and in sporadic SHH medulloblastomas, were characterised by i) high Sonic Hedgehog (SHH) signalling ii) high proliferation and iii) neuronal development and differentiation, respectively[12] (Fig. 3B). Cells with SHH signatures and high cycling activity were frequently co-localized, suggesting possible transitory states of one unique cell population,

such as progenitor-like cells, which are inherently proliferating. In contrast, cells with a more differentiated signature were typically located in separate tumour areas (Fig. 3B, right column). This was true for LFS as well as sporadic medulloblastomas. More generally, we assessed pairwise spatial correlations between all major cell types, which identified both negative and positive correlations in LFS medulloblastoma (Supplementary Fig. 12). Sporadic medulloblastomas were characterised by overall slightly weaker spatial cell-type correlation, again suggesting a more homogeneous spatial structure in sporadic medulloblastoma, even though the observed effect was modest (Supplementary Fig. 12). To better understand the signal transduction pathways that drive interactions across cell types, we tested for signatures of spatial cell-cell communication using COMMOT[26]. This analysis identified midkine signalling as a major signalling pathway in LFS medulloblastoma (Fig. 3C, E). Midkine signalling was more active in LFS compared to sporadic medulloblastomas (two-fold higher score for cell-cell communication), suggesting a critical role for this pathway for the communication between cell populations

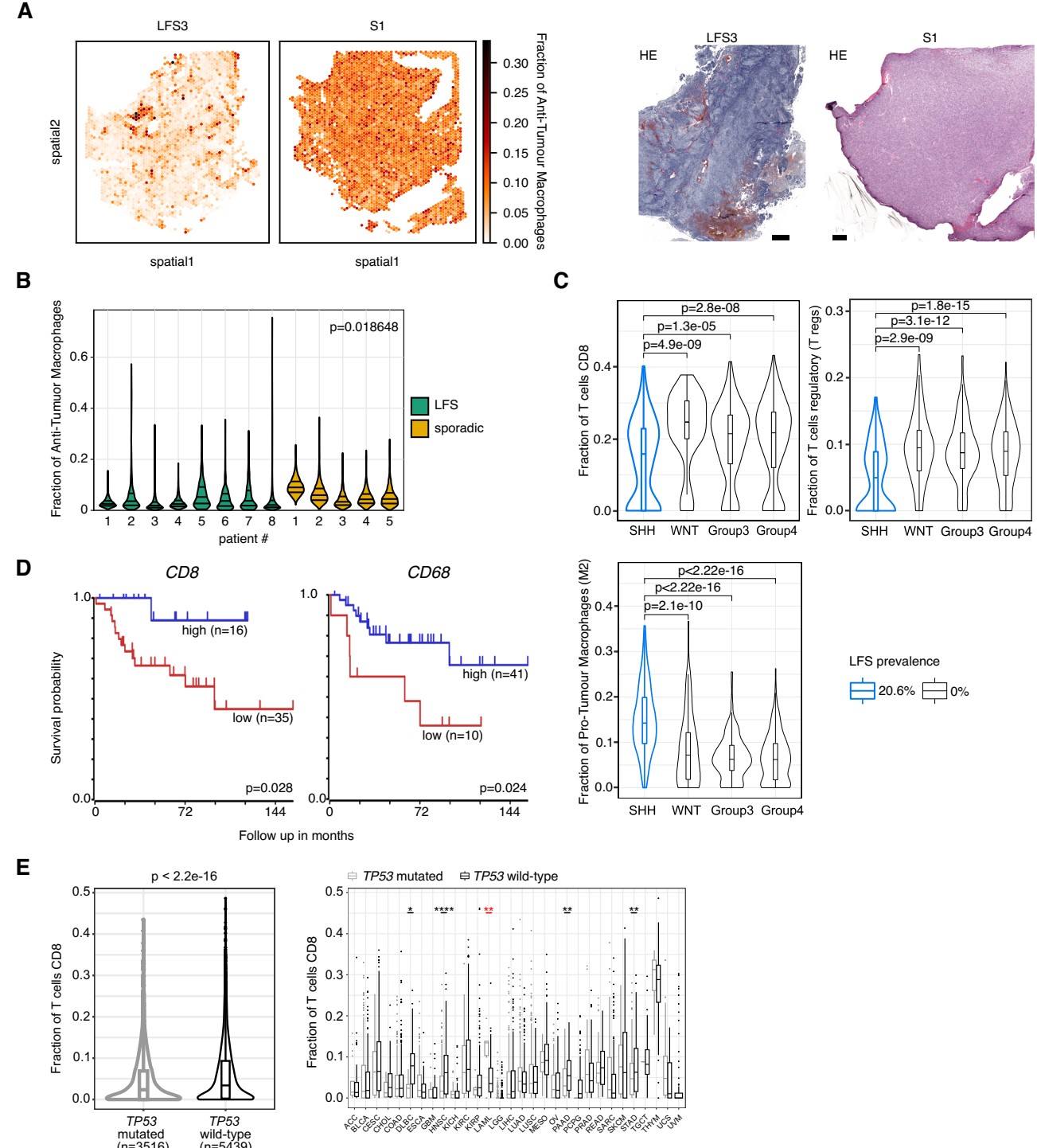

**Fig. 2 | Immune cell infiltration in LFS medulloblastomas is lower as compared to sporadic medulloblastomas. A** Left, fraction of anti-tumour macrophages for each Visium spot in one representative LFS ($n$ = 2469 spots) and one sporadic ($n$ = 2966 spots) medulloblastomas. Right, matched haematoxylin and eosin stain. Scale bars, 500 μm. **B** Quantification of anti-tumour macrophages for all medulloblastomas (LFS, $n$ = 8; sporadic, $n$ = 5). First, second, and third quartiles are indicated. Two-sided Mann-Whitney U-test was used. **C** Cell types fraction estimates from the deconvolution of bulk RNA-seq ($n$ = 763 patients), identifying lower T-cells but higher pro-tumour (M2) macrophage fractions in the SHH medulloblastoma subgroup, which is enriched for LFS medulloblastomas. Two-sided Mann-Whitney U-test was used. **D** Kaplan-Meier survival curves for SHH alpha subtype ($n$ = 51

patients) from a primary medulloblastoma cohort[5] grouped by *CD8* and *CD68* expression. Low expression of *CD8* and *CD68* is associated with poor outcome in the SHH alpha medulloblastoma subtype (log-rank test, Kaplan-Meier plots generated using the R2 database, see Methods). **E** Less T cells in *TP53* mutant tumours (re-analysis of a pan-cancer cohort, deconvolution from bulk RNAseq[49], $n$ = 8955 patients). Left, fraction of CD8 positive T cells shown for all *TP53* mutated and *TP53* wild-type patients. Right, fraction of CD8 positive T cells shown for each tumour type. LAML is the only tumour type for which the fraction of CD8 positive T cells is significantly higher in *TP53* mutant tumours. Two-sided Mann-Whitney U-test was used. Boxes show the interquartile range (IQR), line indicates the median, whiskers indicate 1.5x IQR.

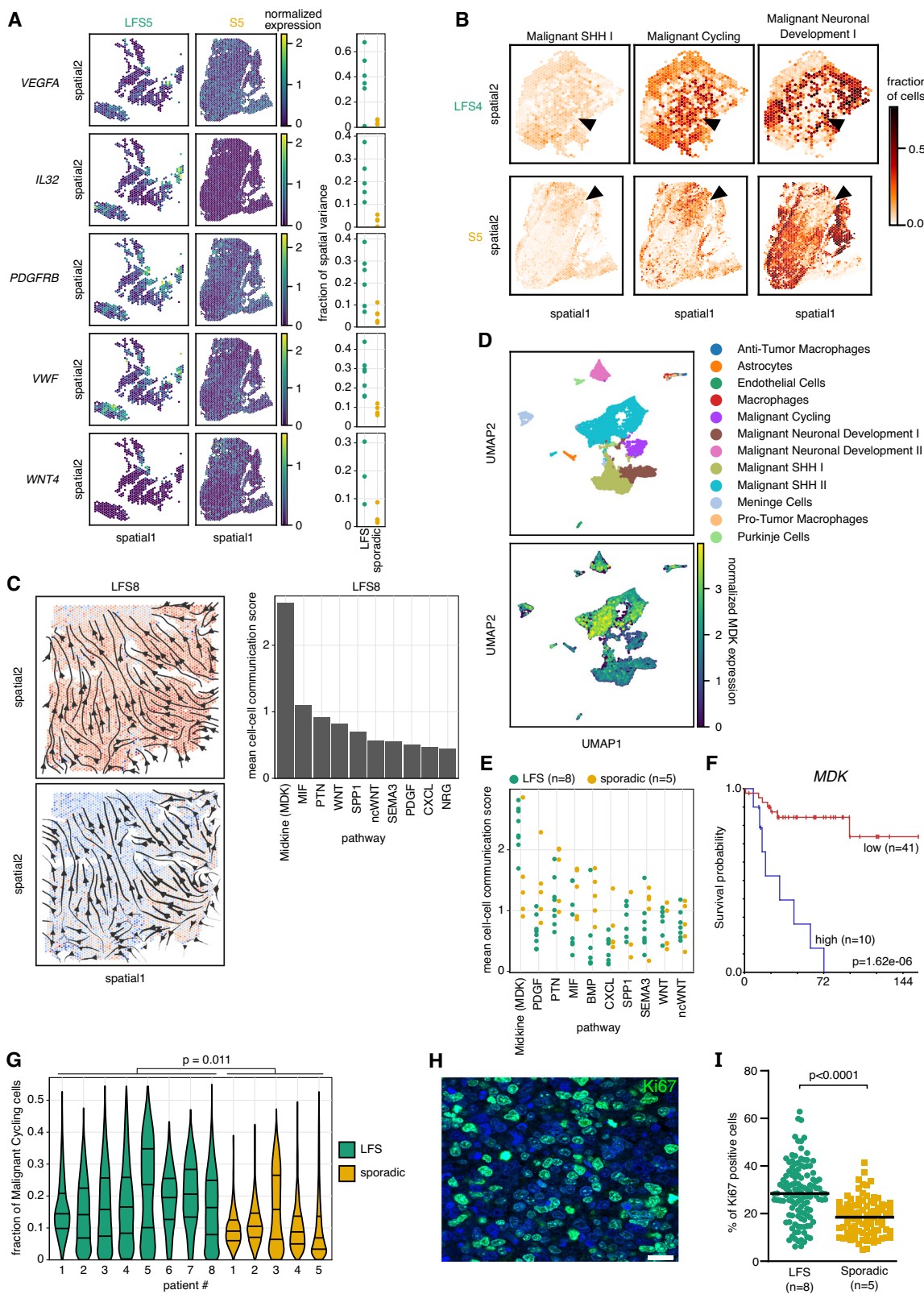

in LFS medulloblastoma. The midkine ligand *MDK* was expressed by malignant cells, as shown in matched scRNA-seq data of the same LFS medulloblastomas (Fig. 3D, Supplementary Fig. 13, Supplementary Data 7). Re-analysis of existing bulk RNA-seq data identified high expression of *MDK* as linked with poor prognosis in medulloblastoma (Fig. 3F, SHH alpha subtype, enriched for LFS medulloblastomas, Supplementary Fig. 14 including all four medulloblastoma subtypes).

At the same time, other pathways (e.g., PDGF, BMP) were identified as more relevant in sporadic medulloblastomas (Fig. 3E). Further studies will be needed to investigate the precise role of midkine signalling in LFS medulloblastoma, as this pathway is involved in cancer cell survival and proliferation, inflammation and angiogenesis, evasion of apoptosis, cancer cell invasion and metastasis, as well as immunosuppression in different tumour types[27,28].

**Fig. 3 | Higher spatial intra-tumour heterogeneity, proliferation and stemness in LFS medulloblastoma as compared to sporadic medulloblastoma. A** Spatial distribution of cancer drivers. Heterogeneous expression of oncogenes and essential microenvironment factors in LFS ($n = 500$ spots) and sporadic ($n = 2723$ spots) medulloblastoma. Left, representative examples; right, distribution of the fraction of spatial variance for the corresponding genes for LFS ($n = 8$) and sporadic ($n = 5$) tumours. Two-sided Mann-Whitney U-test was applied, non-significant after Benjamini-Hochberg multiple testing correction. **B**. Fractions of malignant cell types for each Visium spot in one representative LFS ($n = 881$ spots) and one sporadic ($n = 2723$ spots) medulloblastomas. Malignant cell clusters of SHH-high and cycling cells co-localize but neuronal development clusters are located in separate regions. **C** Cell-cell communication analysis, revealing Midkine signalling (MDK) as essential driver for cell-cell communication in LFS medulloblastoma. Left, overview of midkine signalling in a representative sample ($n = 3983$ spots). Colour indicates the extent of signal emitted by Visium spot in the upper plot and the received signal in the lower plot (from blue to red). Arrows indicate the overall direction of signalling. Right, average COMMOT cell-cell communication score for the leading 10 pathways. **D** UMAPs ($n = 15{,}265$ cells) based on matched 10x RNA-seq data (4 LFS medulloblastomas) with cell type annotation and expression of the MDK ligand. **E** COMMOT cell-cell communication scores, averaged across locations, for all medulloblastomas (LFS, $n = 8$; sporadic, $n = 5$), displaying the 10 leading pathways. n.s., two-sided Mann-Whitney U-test with Benjamini-Hochberg multiple testing correction. **F** Kaplan-Meier survival curves for SHH alpha subtype ($n = 51$, enriched for LFS medulloblastomas) grouped by *MDK* expression. High expression of *MDK* is associated with poor outcome in the SHH alpha medulloblastoma subtype (log-rank test, Kaplan-Meier plots generated using the R2 database, see Methods). **G** Fraction of the Malignant Cycling cells across spatial locations in LFS medulloblastomas ($n = 8$) as compared to sporadic medulloblastomas ($n = 5$). Horizontal lines denote the first, second, and third quartiles. A two-sided Mann-Whitney U-test was used. **H** Validation at protein level by immunofluorescence using Ki67 as a proliferation marker. Representative image of one sample is shown. Scale bar, 20 μm. **I** Quantification of Ki67-positive cells for all medulloblastomas (LFS, $n = 8$; sporadic, $n = 5$). Median is indicated. Statistical significance was calculated using a two-sided Mann-Whitney U-test.

---

Beyond signalling pathways active in LFS medulloblastomas, a further feature contributing to the aggressiveness of LFS tumours was a significantly higher proportion of cells with elevated cycling activity and SHH-high signatures as compared to sporadic medulloblastomas (Fig. 3G), possibly due to progenitor-like cells. We confirmed the quantifications of the cycling activity at protein level by immunofluorescence using Ki67 as a proliferation marker and NEUROD1 as a differentiation marker (Fig. 3H–I, Supplementary Fig. 15). Collectively, our results point to increased spatial intra-tumour heterogeneity, elevated signatures of proliferation and stemness, reduced immune infiltration and differentiation in LFS medulloblastomas compared to sporadic tumours.

## Spatial mapping of genetic clones identifies phenotypic features associated with genomic profiles

In order to relate the spatial heterogeneity of LFS medulloblastomas to their underlying genetic make-up, we next set out to infer the spatial composition of genetic clones. Briefly, we utilised previously generated matched single-cell DNA sequencing data from a primary-relapse pair of LFS medulloblastomas[12] to estimate single-cell copy number profiles. Analysis of the copy number profiles identified two distinct genetic clones—an ancestral clone B and a more recent clone with an additional chromothriptic event (Fig. 4A, Supplementary Fig. 16). Focusing on chromosome regions with variable numbers of copies between these two main clones (chromosomes 8 and 12), we used the expression levels of genes located on these chromosomes to estimate the spatial locations in the Visium slides (Methods, Fig. 4B, Supplementary Fig. 16). To confirm the spatial clone assignments, we employed FISH on adjacent tissue sections. Briefly, we quantified the number of signals for two probes specific to loci anticipated to exhibit varying copy numbers between the two primary clones (previously detected by single-cell DNA sequencing) on chromosome 8 and chromosome 12, respectively. Indeed, the distribution of FISH signals in the target areas from the major clone A was distinct from those observed in the evaluated areas from the major clone B (Fig. 4C, Supplementary Fig. 17). The ancestral clone was mostly diploid for these two loci, whereas the daughter clone harboured an additional chromothriptic event and showed a high fraction of polyploid cells, in line with the tight connection between polyploidy and chromothripsis[29]. Having established the validity of the clone mapping, we compared the clone-associated regions phenotypically. The two regions associated with the two major clones were corroborated by distinct histological features (e.g., higher cellularity versus more necrosis) as well as different variability in the nuclear morphology (Fig. 4D), including a significant difference in the size of the nuclei (Fig. 4E). The tumour region associated with the daughter clone A was also enriched for higher cycling signature, whereas the area with the

ancestor clone B was enriched for differentiated cells (Fig. 4F, G, Supplementary Data 8). Hence, the spatial mapping of genetic clones identified phenotypic features associated with each clone, with phenotypic variation between clones molecularly related to the differences detected between LFS and sporadic tumours (c.f. Figs. 1, 3).

## Defining genomic features and signalling pathways active in treatment-resistant tumour cells in patient-derived xenografts of chromothriptic medulloblastoma

Chromothriptic medulloblastomas frequently relapse. In order to survey the spatial makeup and the clonal composition of the tumours across the time course of treatment, we considered PDX models derived from the same patient (matched primary-relapse pair, see analysis of the patient tumour in Fig. 4). Specifically, we selected PDX models established from the relapse biopsy, and focused on combinations of particle radiation with PARP inhibitors (Fig. 5A). Conventional photon radiotherapy and DNA-damaging chemotherapy are not successful for these patients and increase the risk of secondary malignancies. We previously showed that the pronounced homologous recombination deficiency in these tumours offer vulnerabilities that can be therapeutically utilised[17,30]. In particular, combining PARP inhibitors and carbon ions was successful in the context of primary chromothriptic medulloblastomas[17]. In the PDX models of pre-treated tumours utilised in this study (derived from a patient who experienced a relapse after therapy), the majority of mice ($n = 22$) exhibited tumour regrowth after a phase of minimal residual disease (Fig. 5B, Supplementary Data 9). We selected $n = 11$ mice for spatial transcriptomics analysis, covering the key stages of resistance, to characterise the tumour cell population(s) that persist through the treatment and lead to tumour regrowth (Fig. 5A, Supplementary Fig. 18).

We first compared the PDX models to the matched patient tumours. Notably, the two-dimensional neighbourhood structures from patient tumours were qualitatively preserved in matched PDX, indicating that, even though a deeper analysis of the mouse tumour microenvironment will be necessary for a fine-grained comparison of unique and common patterns between human medulloblastomas and these models, the latter globally reflect cellular interactions from human medulloblastomas (Supplementary Fig. 19A). Additionally, the PDX models more closely resembled the spatial patterns observed in the LFS medulloblastomas than sporadic tumours (Supplementary Fig. 19A-D), supporting the overall faithfulness of these in vivo models.

Next, we analysed correlations between cell types in two-dimensional neighbourhoods, which were markedly consistent across PDX treatment groups and disease stages. In line with the patterns observed in LFS patient tumours (c.f. Supplementary Fig. 12), the cell-type correlation structure in the PDX reflected the general patterns of negative correlations between regions dominated by high

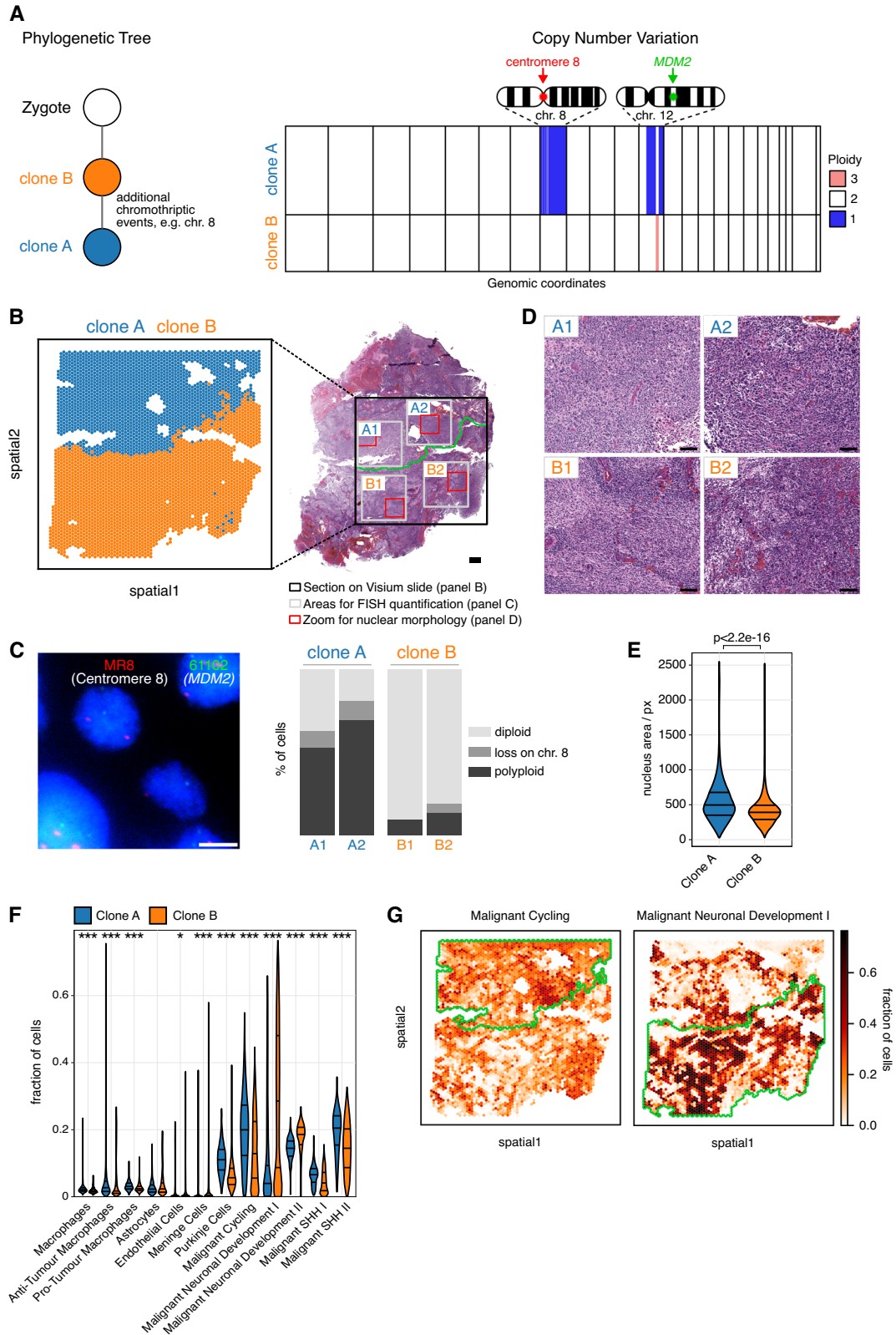

cycling activity and regions enriched for differentiated tumour cells (Supplementary Fig. 20).

**Subclones co-exist across the treatment response time course**
Application of the transcriptome-guided clone mapping approach to the PDX samples identified both major genetic clones present in the original patient tumour from which the PDX model was derived (Fig. 6A,

Supplementary Fig. 21). Importantly, the two main clones A and B were also detected in the regrown tumours that had passed the minimal residual disease stage (Supplementary Fig. 22), indicating a high degree of genetic stability up to resistance. To further investigate the genetic makeup across the treatment time course, we again applied the FISH probe panel considered in the primary tumour analysis (c.f. Figure 4), providing evidence that both clones are sustained (Supplementary

**Fig. 4 | Spatial mapping of major genetic clones identifies clone-specific molecular features. A** Schematic representation of the evolutionary relationship of two major clones identified by single-cell DNA sequencing, and visualisation of distinct copy numbers differences between clones that were used for spatial mapping (Note: polyploid clones are not called by HMMcopy, hence ploidy-normalised copy numbers are shown). **B** Inference of the spatial mapping of the major genetic clones as in A in one LFS medulloblastoma (patient LFS8, $n = 3983$ spots). Right, matched haematoxylin and eosin stain of the medulloblastoma tissue. Scale bar, 500 μm. **C** FISH validation on the consecutive tissue section of patient LFS8 using probes for centromere 8 and *MDM2*, located in two chromosome regions with differential copy numbers between clones A and B. Left, representative image of nuclei. Right, quantification of FISH signals of each probe in areas A1, A2, B1 and B2 from clones A and B. Scale bar, 10 μm. **D** Zoom-in on regions from B corresponding to different clones. Differences in nuclear morphology and histological subtype between tumour areas enriched for distinct genetic clones. Scale bars, 100 μm. **E** Morphological feature analysis of nuclei supports the clone mapping results. Clone B (which contains more diploid cells) has smaller nuclei as compared to the daughter clone A, which has a higher fraction of polyploid cells, consistent with the larger nuclei. A two-sided Mann-Whitney U-test was used (clone A, $n = 126868$ nuclei; clone B, $n = 197189$ nuclei). **F** Cell type fractions for individual locations for clone A and clone B. First, second, and third quartiles are indicated. $*p < 0.01$, $***p < 1e-10$ (two-sided Mann-Whitney U-test, Benjamini-Hochberg multiple testing correction (clone A, $n = 1713$ spots; clone B, $n = 2270$ spots). **G** Specific transcriptional programs are enriched in tumour areas comprising distinct genetic clones (patient LFS8, $n = 3983$ spots).

Fig. 23). As a third approach, independent of CNVs, we also used allele frequency distributions of somatic SNVs to identify subclones and estimate their clonal fractions across the time course, again supporting the stability of the major clones (Supplementary Fig. 24B). To further characterise resistant cells, we performed immunofluorescence analysis of SOX2 (previously described as stem cell marker in medulloblastoma[25]), which showed widespread SOX2 expression at minimal residual disease on adjacent sections of the tissue used for FISH (Supplementary Fig. 25). In addition, *SOX2* expression in the Visium data identified positive cell populations in both major genetic clones A and B in PDX as well as throughout the human medulloblastomas (Supplementary Fig. 26). Altogether, this points to the existence of resistant cell populations in multiple genetic clones, with putative stem-like compartments in different clones.

Our results suggest that the treatment resistance may not originate from the selection of one homogenous resistant cell population, but that treatment-tolerant cells from multiple genetic clones survive and collectively lead to relapsed tumours. We used FISH to evaluate the clonal composition of invading tumours in PDX, whereby tumour cells had migrated from the mouse cerebellum to the frontal cortex (Supplementary Fig. 27). The distribution of genetic clones in the invaded cortex was close to that observed in the initial cerebellar medulloblastomas, suggesting a multi-clone migration to the frontal cortex.

Having established the presence of both clones on broad terms, we set out to map out the changes in the cell composition within these clones across the treatment time course (Fig. 6C). Briefly, we projected the major cellular programs identified from spatial transcriptomics profiling of the human relapse tumour, which segregate across the major clones (Fig. 6C) onto the PDX time course, estimating changes in the relative composition in response to the treatment (Fig. 6C–E; Supplementary Fig. 28). While again, this analysis suggests that cell types from both clones are present across the time course, it identified cellular treatment response patterns. Interestingly, cells from region types 3 and 6 (differentiated malignant cells) initially decreased upon treatment but reappeared in the regrown tumours (Fig. 6E). This suggests that differentiated medulloblastoma cells are more sensitive to treatment and less abundant at the stage of minimal residual disease, but this compartment is re-established upon tumour regrowth. Cells from region type 2 (resistant tumour cells) were enriched at minimal residual disease but decreased in the regrown tumours. This resistance signature included genes involved in processes such as the regulation of apoptosis, adaptation to stress, resistance to nutrient deprivation or inhibition of cell growth by prolonging the G0/G1 phase. Altogether, tracking tumour cells over the course of the treatment applied in vivo showed within-clone changes in the cell types and identified putative cell populations that lead to relapse.

## Tumour microtubes are disrupted by the treatment but regrow in relapsed tumours

Beyond changes in the cell type composition from the minimal residual disease stage to the regrown tumours, we set out to identify gene signatures associated with resistance, to gain further insights into the spatial and temporal properties of treatment resistance. Comparing tumours from the control group, the minimal residual disease stage and regrown tumours identified *GAP43* as strongly differentially expressed, with a low expression at the minimal residual disease stage, but increasing in the regrown tumours (Fig. 7A–B). Of particular relevance in the context of treatment resistance, GAP43 was previously described as a major factor associated with tumour microtubes in glioblastoma[31]. Ultra-long membrane protrusions were described in glioblastoma cells, which use these microtubes for invasion, proliferation and to interconnect over long distances. These tumour microtube networks allow multicellular communication in tumours through microtube-associated gap junctions[31,32]. Genes previously associated with a tumour microtube connectivity signature in glioblastoma[31] were downregulated at the minimal residual disease stage (enrichment tests for tumour microtube genes in the Visium data, hypergeometric test: enrichment of TM-high genes in downregulated genes, tm_high enrichment pval: 1.63e-08 and TM-low genes in upregulated genes, tm_low enrichment pval: 0.008). To search for potential tumour microtubes in LFS medulloblastoma xenografts, we used immunofluorescence and identified nestin-positive tumour microtubes in primary and relapsed PDX (Fig. 7C), similar to those previously described in glioblastoma[31]. These tumour microtubes were positive for STEM121 (marker of human tumour cells in xenografts) and TOM20 (mitochondrial marker, previously linked with tumour microtubes[32]). In line with the *GAP43* expression patterns, quantification of the tumour microtube area per cell identified significantly larger tumour microtube areas per cell in the untreated relapsed PDX as compared to the matched untreated primary PDX tumours, and also showed a decrease in the tumour microtubes at the minimal residual disease stage and an increase in the regrown tumours (Fig. 7D). Therefore, tumour microtubes might potentially play a role in treatment resistance in LFS medulloblastoma. In line with this, we also identified tumour microtubes in 15 out of 17 analysed patient medulloblastomas (Fig. 7E, Supplementary Fig. 29). The tumour microtube marker nestin showed a significantly higher expression in the medulloblastoma subgroup enriched for LFS (SHH alpha, $n = 51$) as compared to sporadic medulloblastoma, as well as a significant association with overall survival (Supplementary Fig. 30A-B, bulk RNA sequencing; this was also true for other tumour microtube network relevant markers such as *TTYH1* and *GJA1*, see Supplementary Fig. 30C) and also a widespread expression throughout tumours (Supplementary Fig. 31–32, $n = 8$ Visium LFS samples, difference to the 5 sporadic samples non-significant). To test whether tumour microtubes in medulloblastoma build functional networks that have been related to vivid and rhythmic intracellular Ca transients in other brain tumour entities[32,33], we performed live-cell imaging of calcium signalling activity in p53-deficient medulloblastoma cells with chromothripsis (Fig. 7F–H, Supplementary Movies 1–3). Medulloblastoma cells showed rhythmic calcium activity, as visualised using different calcium indicators (Fig. 7F–H, Supplementary Movies 1–3). In addition, staining of

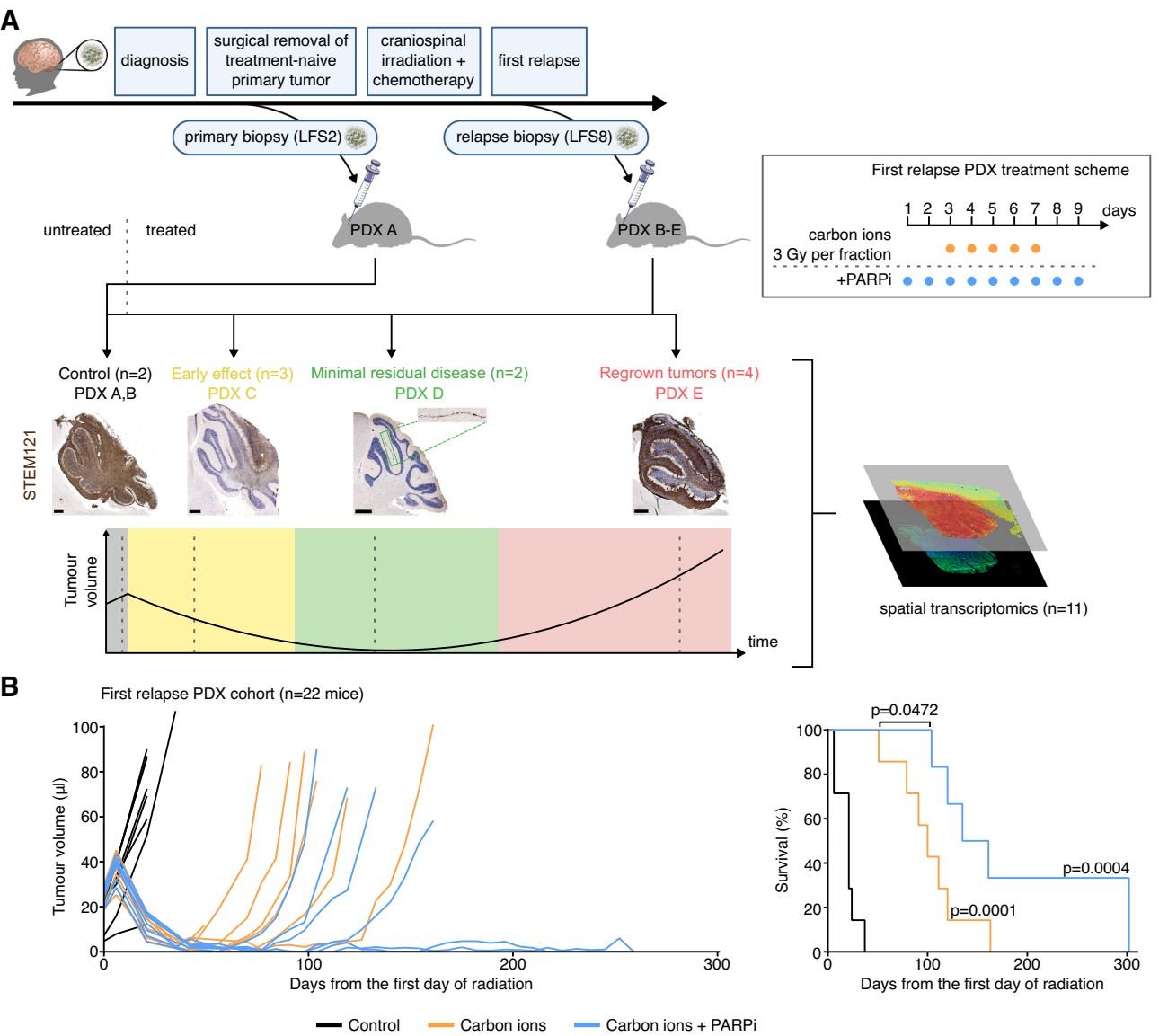

**Fig. 5 | Spatial and Temporal Transcriptomics in patient-derived xenografts to characterise cell populations that cause treatment resistance and give rise to recurrence. A** Top: Clinical history of LFS medulloblastoma patient from which patient-derived xenografts were derived. Immunocompromised mice orthotopically injected with medulloblastoma cells from the relapsed tumour (LFS8) were randomised based on tumour volumes and treated with particle radiation (five fractions of carbon ions) alone or in combination with a PARP inhibitor. Bottom: Time points for treatment of animals used for spatial transcriptomics experiment ($n = 11$ mice). STEM121 was used to visualise human cells in the mouse brain (brown stain). Scale bars, 500 μm. Head and brain icons are adapted from the Reactome icon library, used under CC BY 4.0 (https://creativecommons.org/licenses/by/4.0/). Mouse, syringe, and cancerous cell icons are adapted from Bioicons (user Servier), used under CC BY 3.0 (https://creativecommons.org/licenses/by/3.0/). **B** Effects of fractionated particle radiation (5×3 Gy) alone or in combination with PARP inhibitor BGB290 on tumour growth and on survival ($n = 8$ for controls, $n = 7$ for carbon ions, $n = 7$ for carbon ions + PARP inhibitor) in PDX models from a pre-treated SHH medulloblastoma-LFS patient (PDX B in **A**). Tumour volumes were measured by MRI. The results of log-rank (Mantel-Cox) tests on Kaplan–Meier curves indicate the survival benefit of each treatment group in comparison to control but also between combination treatment and irradiation only.

medulloblastoma cells with membrane dye Lipilight560 in combination with the Rhod-2 staining showed that medulloblastoma cells form membrane tube connections between each other (Fig. 7G, Supplementary Fig. 33). Altogether, tumour microtubes in LFS medulloblastoma may contribute to the aggressiveness of these tumours. Future studies investigating the function of such networks may lay the basis for potential pharmacological targeting of tumour microtubes.

## Discussion
Our spatial and temporal transcriptomic analyses of medulloblastoma with chromothripsis identified marked differences between LFS and sporadic medulloblastoma but also specific features of tumour cell populations that give rise to relapsed tumours. Our data question

models supporting a homogeneous pool of tumour stem cells at the top of a hierarchy, as medulloblastoma cells from different genetic clones survived treatment, with distinct phenotypic compartments within each genetic clone. Even though we cannot formally exclude the possibility that one single genetic clone survived the treatment and rapidly regenerated the daughter clones following the exact same evolutionary path, it seems plausible that tumour cell populations from distinct genetic clones survive. Such resistant cells with distinct copy number profiles may potentially be more quiescent than the treatment-sensitive tumour cells from each clone and positively selected upon treatment due to their dormancy state. Alternatively, these resistant cells may enter a transition state of dormancy upon treatment. Importantly, the PDX models that we selected here already

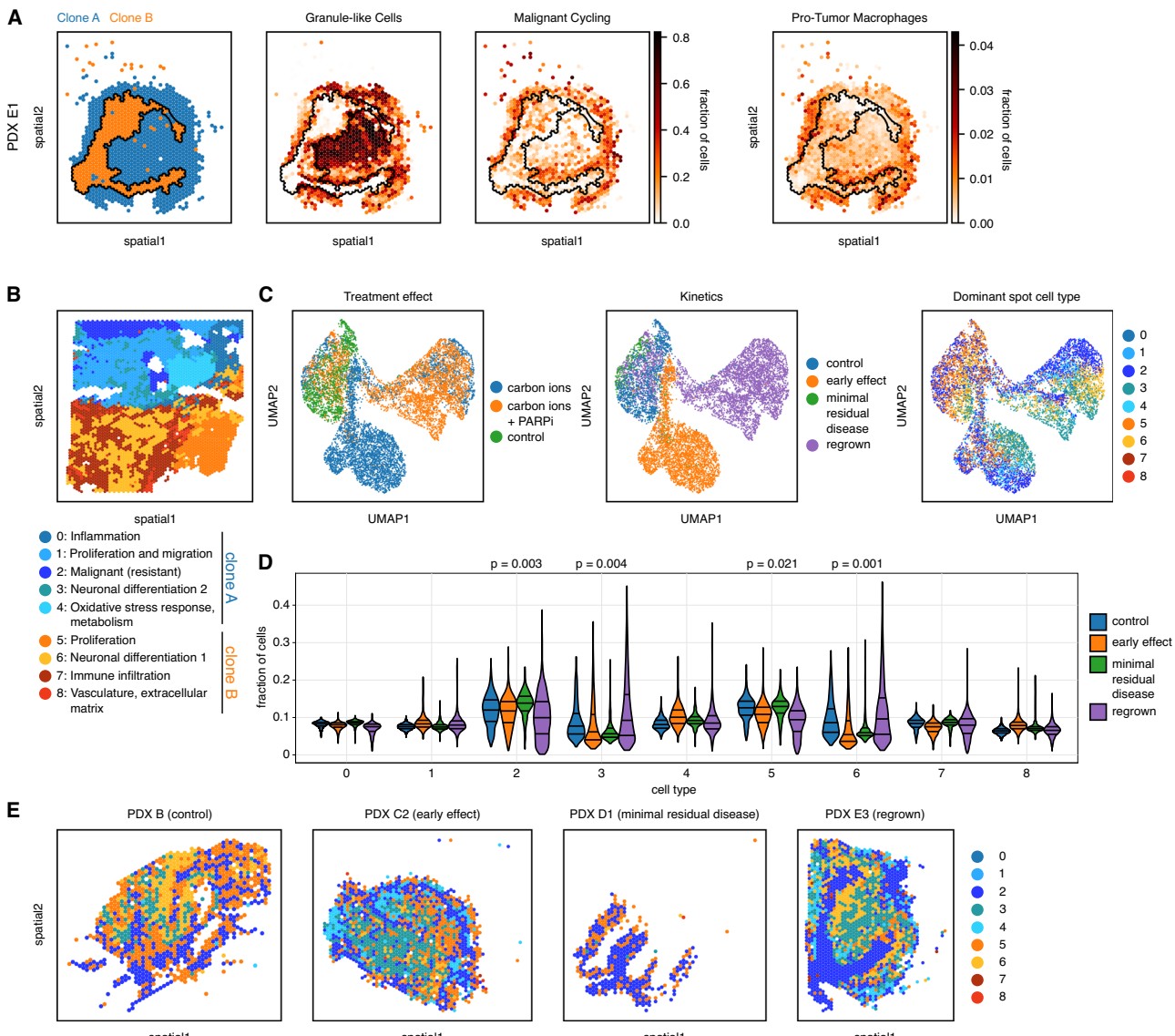

**Fig. 6 | Monitoring the response to treatment in PDX models. A** Mapping of genetic clones as identified from the patient tumour (c.f. Fig. 4B) to spatial transcriptomics locations (*n* = 1147 spots) from xenografted mouse brain tissue, which identifies both major clones in the xenografts. More progenitor-like cells, more cycling cells and more pro-tumour macrophages are within or close to clone B **B** Spatial segmentation of a relapsed LFS medulloblastoma sample (*n* = 3983 spots). **C** UMAPs of an integrated dataset across all 11 PDX samples, across time points and treatment groups (*n* = 9989 spots in total). Left: Colour-coded by treatment group. Middle: Colour-coded by time point. Right: Colour-coded by the most abundant cell type as inferred by cell2location, where cell types are defined by tissue regions

of C. **D** Projections of the tissue region signatures from the original patient tumour on the combined PDX dataset. Violin plots show the evolution over the course of the treatment of cell types from all regions of the original patient tumour. First, second, and third quartiles are indicated. Statistical significance was assessed by a regression *t*-test on the medians (*n* = 11 samples, control, *n* = 1193 spots; early effect, *n* = 3391 spots; MRD, *n* = 547 spots; regrown, *n* = 4858, spots). **E** Representative samples across the time course coloured by the estimated dominant cell type at each Visium location (PDX B, *n* = 1166 spots; PDX C2, *n* = 1610 spots; PDX D1, *n* = 416 spots, PDX E3, *n* = 1741 spots).

went through one evolutionary bottleneck induced by the treatment received by the patient. As the current treatment scheme fails for LFS medulloblastoma and as relapsed tumours are fatal, testing novel therapeutic options in large cohorts of PDX models of relapsed tumours but also dissecting the response of the different tumour cell populations will be essential. Furthermore, multiome studies will be essential to disentangle to which extent transcriptional heterogeneity arises from CNVs or from transcriptional diversity, and to ultimately target genetic-driven as well as transcriptional plasticity-mediated resistance. Unsupervised approaches offering a higher resolution will allow the detection of minor genetic clones, to evaluate the possible contribution of such clones in resistance and to provide a more fine-

grained resolution to the current picture of overall stable genetic composition. We discovered tumour microtubes in LFS medulloblastoma similar to those previously reported in other aggressive brain tumours[32]. In glioblastomas, cells that are part of a microtube-connected tumour network show increased stemness and have a higher potential to reinitiate brain tumour growth[31]. Future studies will be needed to understand the precise function of tumour microtube network formation and integration in treatment resistance in medulloblastoma. Microtubes in LFS medulloblastomas may represent a feature contributing to the aggressiveness of these tumours, and potential benefits of pharmacological targeting of tumour microtubes may be investigated in future studies.

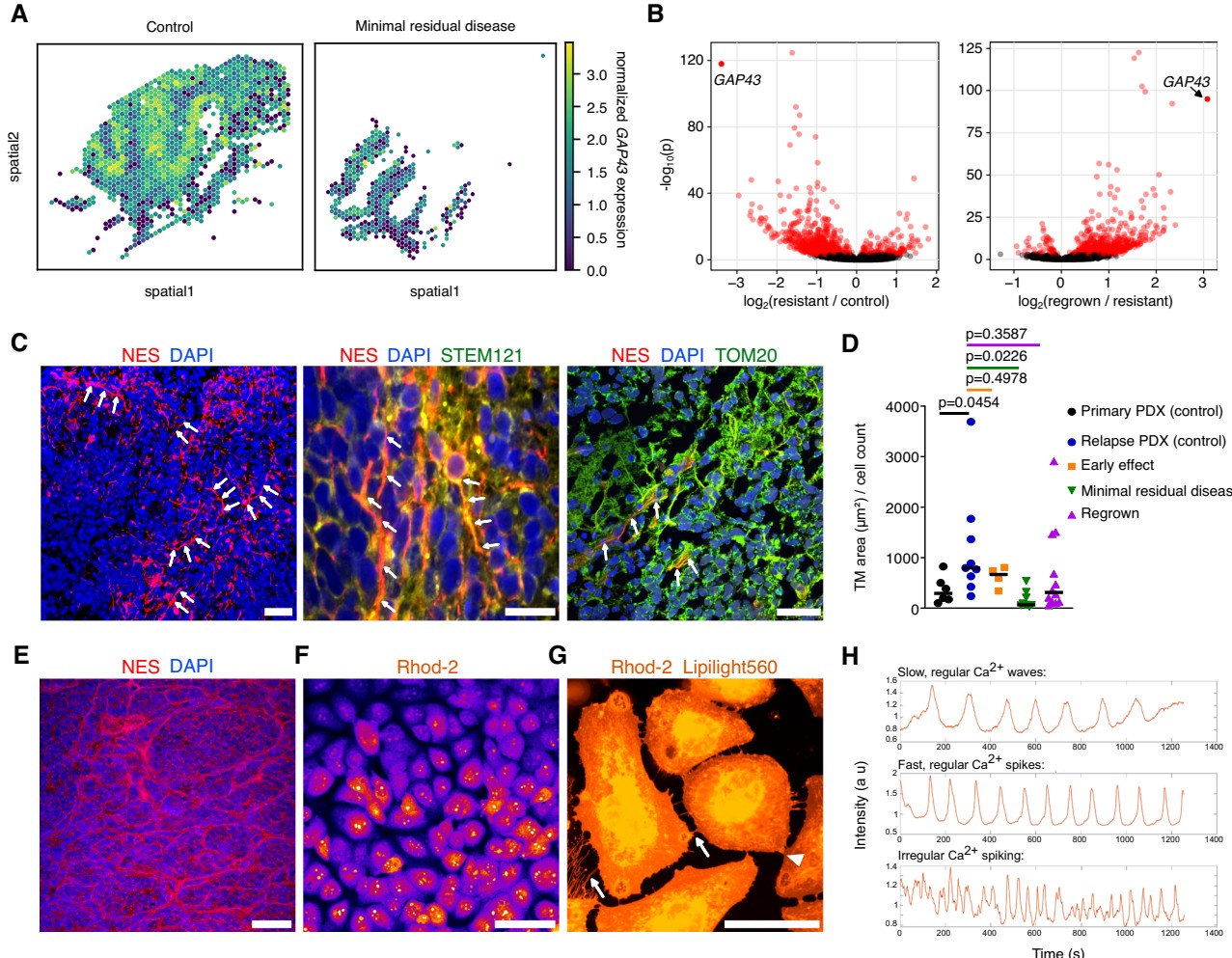

**Fig. 7 | Tumour microtubes in LFS medulloblastoma. A** Low expression of the marker of tumour microtubes *GAP43* at minimal residual disease (*n* = 416 spots, one representative sample) compared to control (*n* = 1166 spots, one representative sample). **B** *GAP43* belongs to the most differentially expressed genes, with a strongly decreased expression at minimal residual disease but increased in the regrown tumours. Analysis was performed with DESeq2, red dots indicate differentially expressed genes at FDR 0.01. **C** Visualisation of tumour microtubes in PDX of LFS medulloblastoma by immunofluorescence analysis. Nestin (NES) was used as a marker of tumour microtubes (image representative of 15 PDX samples), STEM121 as a marker of human cells in the mouse brain (image representative of one PDX sample), TOM20 as a mitochondrial marker (image representative of 5 PDX samples). Scale bars: 50 µm (left), 20 µm (middle and right). Arrows indicate examples of tumour microtubes. **D** Quantification of the tumour microtube area per cell. Median is indicated. Significance testing between the untreated primary PDX (control, *n* = 2 mice) and matched untreated relapse PDX (control, *n* = 3 mice) was performed using a one-tailed unpaired *t*-test. To compare the three treatment

groups (*n* = 3 mice/group for early effect and minimal residual disease; *n* = 4 for the regrown time point) to their matched untreated relapse PDX (control, *n* = 3 mice), Ordinary one-way ANOVA with Dunnett's multiple comparison test was used. **E** Representative image of nestin staining in LFS medulloblastoma primary tissue, showing the presence of tumour microtubes in medulloblastoma patients. Nestin staining was repeated with similar results in 15 out of 17 analysed medulloblastomas (LFS, *n* = 9 patients; sporadic, *n* = 8 patients). Scale bar, 50 µm. **F** Medulloblastoma cells, grown in 2D monolayer, exhibit $Ca^{2+}$ activity as demonstrated with fluorescent calcium indicator Rhod-2. Experiment was repeated three times with similar results (*n* = 3 biological replicates). Scale bar, 100 µm. **G** Medulloblastoma cells in 2D monolayer assay with Lipilight560 and Rhod-2 staining. Arrowhead represents membrane connections, while arrows denote thin processes. Representative image of 4 technical replicates is shown (4 wells of a 96-well plate in which DAOY cells were seeded and stained once for the combination of Lipilight560 and Rhod-2 staining). Scale bar, 50 µm. **H** Three different patterns of calcium oscillations observed in DAOY medulloblastoma cells.

## Methods

### Material and data collection
All experiments in this study involving human tissue or data were conducted in accordance with the Declaration of Helsinki. Clinical data and tissue of all cases in the study were collected after receiving written informed consent from the respective patients or their legal representatives and after approval by the ethics committee of Heidelberg University. Participants did not receive compensation and were informed of this prior to enrolment in the study. The therapy for the LFS medulloblastoma patient from which the PDX models were established included surgery, craniospinal irradiation and vincristin, as well as

chemotherapy six weeks after the surgery (cisplatin, vincristin, CCNU). There was a local relapse after two blocks of chemotherapy in the resection margin.

### DNA-methylation profiling and copy number variation analysis
DNA methylation profiling was performed using the Infinium Human Methylation 450k and EPIC BeadChips as previously described[20]. The Heidelberg Brain Tumor Methylation Classifier v11b6 (https://www.molecularneuropathology.org) was applied for molecular classification. Copy number variation analysis from 450k and EPIC methylation array data was performed using the conumee Bioconductor package version 1.12.0[34].

## Visium spatial gene expression for fresh frozen tissue

**RNA quality evaluation.** To assess RNA quality of frozen tissue blocks, the RNA Integrity Number (RIN) was determined. Sections of 10 μm thickness were cut and placed into pre-cooled microcentrifuge tubes. Using the Qiagen RNeasy Mini Kit (#74104), RNA was isolated. The RIN was quantified using Agilent RNA 6000 Nano Kit (#5067-1511) on a 2100 Agilent BioAnalyzer (#G2939BA) following the manufacturer's instructions.

**Tissue handling, staining and imaging.** Patient-derived xenograft tissue was prepared following the Visium Spatial User Guide **CG000240_Demonstrated_Protocol_VisiumSpatialProtocols_TissuePreparationGuide_RevD**.

TissueTek O.C.T. Compound (#4583, Sakura Finetek) embedded frozen Patient-derived xenograft mouse brains were sectioned at a section thickness of 10 μm. The section was then trimmed to 6.5 × 6.5 mm, placed on the pre-chilled Visium Spatial Gene Expression slide (#1000187 10x Genomics), and stuck firmly by warming the back of the slide. After all sections were placed, the slide was stored at −80 °C in a sealed container until further processing.

Methanol Fixation and Immunofluorescence Staining was performed according to the **CG000312_Demonstrated Protocol Methanol Fixation and IF Staining_RevC**. In brief, the slide was placed on the thermal cycler adapter on a thermal cycler (Mastercycler Gradient, Eppendorf) at 37 °C for 1 min and fixed in pre-chilled methanol at −20 °C for 30 min. Primary antibody staining was performed with blocking, incubation of the primary antibodies Ki67 (#11882, Cell Signaling, 1:50) and NeuroD1 (#AF2746, Biotechne, 1:50) and repeated washing. This step was then repeated for the secondary antibody staining (#705-165-003, Jackson Immuno, 1:500). After immersing the slide in 3X SSC buffer to remove any excess reagents, mounting medium which was prepared according to the User Guide and a coverslip were added.

Imaging of the tissues was performed using the Slideview VS200 research slide scanner (Olympus, Hamamatsu ORCA-Fusion camera, 20x objective/NA 0.80). 1.18 μm thick z-stacks were recorded and EFI (Extended Focus Image) projections were used for downstream analyses.

To remove the coverslip, slides were immersed in 3X SSC buffer until separation. Then slides were placed in the Visium slide cassette and a wash buffer added.

The following steps were performed according to the Visium Spatial Gene Expression User **GuideCG000239_Visium_Spatial_Gene_Expression_User_Guide_Rev_D**.

**Permeabilization and cDNA synthesis.** Permeabilization was performed on a pre-heated thermal cycler at 37 °C with Permeabilization Enzyme for 12 min. Permeabilization time was determined in the previous Tissue Optimization Experiment following the Visium Spatial User Guide **CG000238_VisiumSpatialTissueOptimizationUserGuide_RevC** using the *Visium Spatial Tissue Optimization Slide & Reagents Kit* (#1000193, 10x Genomics), by permeabilizing tissue for different times, synthesising fluorescent cDNA, imaging, and determining the optimal signal.

The following steps were Reverse Transcription, Second Strand Synthesis and Denaturation, performed according to instructions in the User Guide.

**cDNA amplification and quality checks.** Optimum number of cycles for cDNA amplification were determined by qPCR analysis using the StepOnePlus™ Real-time PCR system (#4376600, Applied Biosystems) and the KAPA SYBR® Fast qPCR kit master mix (2X) Universal (#KK4601, Roche Diagnostics). Reaction was set up as per the user guide (#CG000239).

Size selections were performed as per the user guide using AMPure XP beads (#A63881, Beckman Coulter) and 10x Magnetic Separator (1000194 230003).

Cleanup was followed by QC and quantification of cDNA by Agilent TapeStation (#G2991BA, Agilent) with a High Sensitivity D5000 ScreenTape (#5067-5592, Agilent) at a dilution of 1:10.

**Library preparation and sequencing.** Libraries were prepared following the user guide with the steps Fragmentation, End Repair and A-tailing; Post Fragmentation End Repair and A-tailing Double Sided Size Selection, Adaptor Ligation, Post Ligation Cleanup, Sample Index PCR using a previously determined number of cycles from cDNA yield, Post Sample Index PCR Double Sided Size Selection and Post Library Construction QC. C1000 Touch™ thermal cycler (#1851196, Bio-Rad) was used for all steps when necessary.

Quantification of library concentration was performed using Qubit™ 3.0 Fluorometer (#Q33216, Invitrogen). 1 μl of each sample was used for quantification using the Qubit™ dsDNA High sensitivity assay kit (#Q32854, Invitrogen).

Average size of library fragments was determined using the 4200 Tapestation system (Agilent Technologies). 1 μl of each cDNA sample was analysed using the D5000 reagents kit (#5067-5589, Agilent Technologies) and ScreenTape (#5067-5588). 1 μl of each spatial gene expression library sample was analysed using the D1000 reagents kit (#5067-5583) and ScreenTape (#5067-5582).

Sequencing was done at the NGS Core Facility at the DKFZ following the user guide (#CG000239). Libraries were pooled and sequenced on a NovaSeq6000 using paired-end sequencing with Read 1: 28 cycles i7 Index: 10 cycles i5 Index: 10 cycles Read 2: 90 cycles and at least 50,000 read-pairs per spot.

## Visium spatial gene expression for FFPE tissue

Preparation of FFPE tumour tissue was done according to the Tissue Preparation Guide **CG000408_Demonstrated_Protocol_VisiumSpatialProtocolsFFPE_TissuePreparationGuide_RevB**.

**RNA quality assessment.** Assessment of RNA quality was performed for FFPE tissue blocks prior to proceeding to selection of samples for the Visium Spatial Gene Expression experiment. After cleaning surfaces and instruments thoroughly with RNaseZap, sections (2−4 sections, depending on tissue size) were cut on a microtome with a section thickness of 10 μm and placed in a pre-cooled microcentrifuge tube kept on ice. RNA was extracted using the RNeasy FFPE Kit (#73504, Qiagen) following the manufacturer's instructions. Extracted RNA was stored at −80 °C until the next step. Using Nanodrop (#NDLPLUSGL, Thermofisher) RNA concentration was measured and diluted. Then, on a 2100 BioAnalyzer with an Agilent RNA 6000 Pico Kit (#5067-1513, Agilent), DV200, the percentage of RNA fragments >200 nucleotides, was evaluated.

**Section collection and placement.** Blocks were placed in an ice bath and incubated for approximately 15 min. Microtome blade (#14035838382, Leica) was cleaned with 100% Ethanol, surfaces wiped with RNaseZap (#AM9780, Invitrogen).

The water bath was filled with Milli-Q Water and warmed up to 39 °C.

Sectioning thickness was set to 5 μm, and the sections were placed on the water bath with a clean brush, discarding the first few sections that had been exposed to air. Before and after the section used for the Visium Spatial experiment, 5-10 sections were collected on glass slides (#11270, Engelbrecht) for subsequent staining and analysis.

After all sections were placed on the Visium Spatial Gene Expression Slide, the slide was placed in a slide drying rack and incubated in an oven (#SE400, Memmert) at 42 °C for 3 h. Afterwards, the

slide was placed in a slide mailer in an airtight bag with desiccant and kept at room temperature overnight to ensure drying. The slides were stored at room temperature for a maximum of 2 weeks, until proceeding to the next steps.

**Tissue adhesion test.** Using the tissue adhesion slide kit (Visium Tissue Section Test Slide #PN-2000460, 10x Genomics), adherence of sections to the Visium slides was tested. For this, the sections were cut, placed and dried according to the Tissue Preparation Guide from 10x Genomics **CG000408_Demonstrated_Protocol_VisiumSpatialProtocolsFFPE_TissuePreparationGuide_RevB**. After deparaffinization and H&E staining, slides were visually inspected for tissue detachment.

**Immunofluorescence staining and imaging.** Immunofluorescence staining and imaging was done following the Demonstrated Protocol **CG000410_Demonstrated_Protocol_VisiumSpatialFFPE_Deparaffin_IF_RevB** using the Visium Spatial Gene Expression Slide Kit (4 rxns, #PN-1000188). Antibody staining was optimised for the protocol prior to the experiment.

For deparaffinization the slide was placed in a slide rack and incubated in an oven at 60 °C for 2 h. The slide was then immersed and incubated in consecutive baths of Xylene (2, for 10 min each), 100% Ethanol (3, for 3 min each), 96% Ethanol (2, for 3 min each), 85% Ethanol (for 3 min), 70% Ethanol (for 3 min) and finally in Milli-Q water for 20 sec. For decrosslinking, the slide was incubated on a thermal cycler on the Thermocycler Adaptor (#1000194, 3000380, 10x Genomics) for 3 min at 37 °C with an open lid. Then, it was placed in the Visium Slide Cassette and washed once with TE Buffer, pH 9. TE Buffer was added again, the slide covered with a Slide Seal and incubated at 70 °C on the thermal cycler for 1 h. After incubation, TE Buffer was removed, and the slide was incubated in PBS for 5 min.

Immunofluorescence staining consisted of a Blocking step for 5 min, then primary antibody solution for Ki67 (#M724029-2, Agilent, 1:100) and NeuroD1 (#AF2746, Biotechne, 1:50) was added for 1 hr, followed by 3 washing steps. Then, a secondary antibody solution (#705-165-003, Jackson Immuno, 1:500; #715-175-151, Jackson Immuno, 1:500) was added for 1 hr, again followed by 3 washing steps. After the staining was done, the slide was removed from the cassette and immersed in PBS. Then, 5 drops of SlowFade Diamond Antifade Mountant or 85% Glycerol and a coverslip were applied.

Imaging of the tissues was performed using the Slideview VS200 research slide scanner (Olympus, Hamamatsu ORCA-Fusion camera, 20x objective/NA 0.80). 1.18 μm thick z-stacks were recorded and EFI (Extended Focus Image) projections were used for downstream analyses.

The following steps were done according to the Visium Spatial Gene Expression User Guide which can be referred to for more detailed information on the protocol: **CG000407_VisiumSpatialGeneExpressionforFFPE_UserGuide_RevB**.

**Probe hybridization and ligation.** After imaging and removal of the coverslip, the slides were again inserted in the Visium Slide Cassette and covered in PBS. PBS was removed and Pre-Hybridization mix added for 15 min at room temperature. Probe Hybridization Mix was prepared using the human transcriptome probe panel, added to the slide and covered with a Slide Seal. The mix was incubated overnight at 50 °C for at least 18 h on a Thermal Cycler. 3 ×5 min washes were done with preheated Post-Hyb Wash Buffer. Then, 2X SSC Buffer was added and the cassette left to cool to room temperature.

For probe ligation, Probe Ligation Mix was prepared and added to the slide, which was left to incubate on the thermal cycler for 1 hr at 37 °C and then cooled to 4 °C. Post Ligation Wash Buffer was added for consecutive washes at room temperature and at 57 °C, followed by washes with 2X SSC Buffer.

**Probe release and extension.** For RNA Digestion, RNase Mix was prepared, added to the slide cassette, and incubated at 37 °C for 30 min on a thermal cycler. Permeabilization followed with Permeabilization Mix at 37 °C for 40 min. Then followed 2 washes with 2X SSC Buffer. Probe Extension Mix was prepared and added to the slide cassette for 15 min at 45 °C and then cooled down to 4 °C on a thermal cycler.

After a washing step with 2X SSC, Probe Elution was done with 0.08 M KOH incubated at room temperature for 10 min. The solution was then transferred to an 8-tube strip, 5 μl 1 M Tris-HCl pH 7.0 added, vortexed and placed on ice.

**Library construction.** To determine the cycle number for the sample index PCR, qPCR was performed on a StepOnePlus™ Real-time PCR system (#4376600, Applied Biosystems) and the KAPA SYBR® Fast qPCR kit master mix (2X) Universal (#KK4601, Roche Diagnostics). Using TS Primer Mix A from the Kit and KAPA SYBR FAST qPCR Master Mix. 25 cycles were run (98 °C for 3 min, 98 °C for 5 sec, 63 °C for 30 sec, Read Signal, Go to Step 2). The Cq Value was recorded for each sample, threshold set along the exponential phase of the amplification plot, at around 25% of the peak fluorescence value.

Cycle numbers determined for the different samples ranged from 13 to 16. Sample Index PCR was run (98 °C for 1 min, 98 °C for 15 sec, 63 °C for 20 sec, 72 °C for 30 sec, Go to Step 2, Cq value as total number of cycles, 72 °C for 1 min, hold at 4 °C), after Master Mix was prepared using the samples, Amp Mix and Dual Index TS Set A Primers (96 rxns #PN-1000251, 3000511, 10x Genomics).

Size selection was performed as per the user guide using AMPure XP beads (#A63881, Beckman Coulter) and Magnetic Separator (#1000194 230003, 10x Genomics).

Quantification of library concentration was performed using Qubit™ 3.0 Fluorometer (#Q33216, Invitrogen). 1 μl of each sample was used for quantification using the Qubit™ dsDNA High sensitivity assay kit (#Q32854, Invitrogen).

Average size of library fragments was determined using the 4200 Tapestation system (Agilent Technologies). 1 μl of each spatial gene expression library sample was analysed using the D1000 reagents kit (#5067-5583) and ScreenTape (#5067-5582).

**Sequencing.** Sequencing was done by the NGS Core Facility at the DKFZ following the user guide (#CG000407). Libraries were pooled and sequenced on a NovaSeq6000 using paired-end sequencing with Read 1: 28 cycles i7 Index: 10 cycles i5 Index: 10 cycles Read 2: 90 cycles and at least 25.000 read pairs per spot covered with tissue.

## Heterogeneity quantification by clustering

We used two complementary approaches to cluster the Visium samples: SpatialDE2[22], which takes spatial information into account, and Leiden[35], which works only with gene expression data.

For spatial clustering, we first identified spatially variable genes using SpatialDE2's Gaussian Process model. We then used the 2000 most highly expressed spatially variable genes as input to SpatialDE2's tissue segmentation algorithm, which incorporates a spatial smoothness assumption: Visium spots in close proximity are more likely to belong to the same cluster.

For Leiden clustering, we normalised the raw expression data to the total number of UMIs per cell, followed by log-transformation. We then identified 2000 most highly variable genes using the Seurat v3 algorithm[36] as implemented in Scanpy[37] and used these to perform principal components analysis. A 15-nearest-neighbour graph was constructed based on the first 20 principal components, which was then used as input to the Leiden clustering algorithm.

For both approaches, we used the same settings for all samples, such that differences in the number of clusters reflect properties of the

sample. To obtain a metric for sample heterogeneity, we normalized the number of clusters by the number of Visium spots in the sample while excluding clusters containing less than 5 spots.

## Fluorescence in situ hybridization

Two-colour interphase FISH[38] was performed on human medulloblastoma sections and patient-derived xenograft sections using a rhodamine-labelled probe (centromere 8) and a FITC-labelled probe (*MDM2*; 611O2). The probes were indirectly labelled via Nick translation. Pretreatment of slides included use of Sodium thiocyanate (1 M NaSCN) at 80 °C. Digestion was performed with pepsin solution (1 mg/ml) for 30 min. Probes were applied on the slide, followed by denaturation step for 10 min at 75 °C. Due to indirect probe labelling, signal detection was done on the next day, after washing slides with 50% Formamid, followed by 0.5x SSC buffer. Samples showing sufficient FISH efficiency (>90% nuclei with signals) were evaluated.

## R2 genomics analysis and visualization platform

R2 Genomics Analysis and Visualization Platform (Website: https://r2platform.com) was used to compare survival data and generate Kaplan-Meier Plots using the data set Tumour Medulloblastoma – Cavalli – 763 – rma_sketch – hugene11t[5], a minimal group size of 10, and separating by a single gene (*CD4*, *CD8*, *CD68*). More specifically, shh was selected for subgroup, or shh_alpha for subtype.

## Confocal microscopy

Leica SP8 confocal microscope was used for the imaging of thick sections (15–30 μm) of frozen mouse tissue stained for human nestin. Imaging was performed using the Leica LAS X software. Z-stacks were imaged with a step size of 0.5 μm. The start and end points of the Z-stack were set manually. Maximal intensity projections were made with Fiji with the Z projection function where the projection type was set to maximal intensity where all the stacks imaged were included in the projection.

## Imaging and scanning of human tissue sections

All H&E, Ki67, STEM121, Nestin and SOX2 stained sections were imaged using the Zeiss Axioscan 7 at the light microscopy facility at the DKFZ. Sections were imaged at 20x magnification with settings adapted from the master fluorescence and brightfield profile for the respective fluorophores. Sample detection was performed by manual selection and focal points were assigned manually for each section.

**SOX2 stain and microscopy.** Five μm thick formalin-fixed paraffin-embedded mouse brain sections were deparaffinized in xylene and subsequently hydrated. Antigen retrieval was carried out by the incubation of formalin fixed mouse brain tissue in a 10 mM citrate buffer adjusted to pH 6.0 at 95–100 °C in a steam cooker for 40 minutes. The sections were blocked with 10% donkey serum diluted in 1x PBS with 0.2% Triton X-100. Anti-SOX2 antibody raised in rabbit (Cell signalling Technology, #23064) at a dilution of 1:200 was used and kept for overnight incubation. Secondary donkey anti-rabbit IgG Alexa Fluor 594 antibody (Invitrogen, #A21207) was used at a dilution of 1:500. Vector® TrueVIEW® Autofluorescence Quenching Kit (#SP-8400) was used to quench autofluorescence from the brain tissue using Reagents A, B and C in equal volumes. Mounting of slides was performed with DAPI Fluoromount-G mounting medium (Southern Biotech, #0100-20).

**Nestin staining.** Five μm thick formalin-fixed paraffin-embedded mouse brain sections were deparaffinized in xylene and subsequently rehydrated. Heat-induced antigen retrieval was carried out by the incubation of deparaffinized sections in Tris-EDTA buffer (10 mM Tris base and 1 mM EDTA solution) adjusted to pH 8.3 at 95–100 °C in a steam cooker for 18 minutes. The sections were blocked with 10%

donkey serum for 1 hour. Recombinant anti-nestin antibody (Abcam, #ab105389) raised in rabbit was used at a dilution of 1:100 and kept for overnight incubation. Secondary donkey anti-rabbit Alexa 594 antibody (Invitrogen, #A21207) was used at a dilution of 1:500. Primary and secondary antibodies were diluted in 1x PBS with 0.2% Triton-X-100.

Thirty μm thick frozen mouse brain sections were fixed with 2% PFA followed by blocking with 10% of donkey serum for 1 hour.

The co-staining of rabbit anti TOM20 (Proteintech, #11802-1-AP) at a dilution of 1:400 and chicken anti nestin antibody (Novus Biologicals, #NB100-1604) at a dilution of 1:500 was performed on 5 μm thick frozen sections. Secondary donkey anti-chicken CF 488A antibody (Sigma Aldrich, #SAB4600031) was used at a dilution of 1:500.

Mounting of slides was performed with DAPI Fluoromount-G mounting medium (Southern Biotech, #0100-20).

**Quantification of microtubes.** For the analysis of tumour microtubes, we extracted three crops of $400 \times 400\,\mu m$ from the mouse whole cerebellum images or the patient tumour sections using Imagej (version 2.0). We then used the software AIVIA (version 10.1) for each of the data sets. After training the pixel classifier for each dataset to detect background, cells, and microtubes differentially, the smart segmentation tool was applied to measure cells and microtubes. Then, the data was post-processed in R to exclude structures detected as cells that were $<10\,\mu m^2$ and for microtubes use structures $> 2\,\mu m^2$. The cell count and microtube area data was then analysed in Prism (version 9.0). *T*-tests were used for the statistical testing.

## In vitro 2D calcium imaging assay

The in vitro 2D calcium imaging assay was performed similarly as previously described[33]: DAOY cells (ATCC® HTB-186™) were obtained from the American Type Culture Collection. Prior to the assay, the cells were authenticated by Single Nucleotide Polymorphism (SNP)-profiling and tested negative for mycoplasma as well as fungal and other bacterial contaminations. The cells were cultured in high-glucose DMEM/F12 medium, supplemented with 10% foetal calf serum, 1 % 200 mM Glutamine and 1% Penicillin-Streptomycin. For the assay, 5.000 – 15.000 cells were seeded into a 96-well plate on a Matrigel matrix (Corning®, #354230). After three days, cells were labelled with fluorescent indicators for intracellular calcium: 1 μM Rhod-2 AM (Thermo Scientific, #R1244), 1 μM Fluo-4 AM (Thermo Scientific, #F14202) or 1 μM Calbryte™ 590 (AAT Bioquest, #20700). To stain any cellular processes, 0.2 μM Lipilight 560 (MemBright, MCO-MEM 560-1906) was combined with the Rhod-2 staining. Following a 30-minute incubation at 37 °C and 5% $CO_2$, $Ca^{2+}$ imaging was performed at the same temperature and $CO_2$ concentration using a Zeiss LSM 980 Airyscan NIR confocal microscope, utilising a 20X objective for calcium imaging and a 63X oil objective for imaging of cellular processes. Every time series was recorded over a 35-minute period, utilising either bidirectional scanning at a speed of 1.26 seconds per frame or monodirectional scanning at a speed of 1.89 seconds per frame. The first five minutes were later cut to accommodate initial stage movement in the z-axis, resulting in videos of 30 minutes each.

**Image processing and analysis.** For image processing, ZEISS ZEN Blue Software and Fiji[39] (ImageJ 2.14.0) were employed. For cellular segmentation, a maximum intensity projection of each raw data video was created. After optimisation of brightness and contrast of this maximum intensity projection in Fiji[39], a custom trained Cellpose[40,41] model was applied. Cellular masks were then used on the raw data in Fiji[39] to measure and save location and mean fluorescence intensities of all cells over time. Next, network identification and analysis was conducted in MATLAB, making use of custom-modified versions of code based on Hausmann et al. 2023[33] and Smedler et al. 2014[42].

## Animal studies

Patient-derived xenograft tumour cells were injected into the cerebellum of 6-10-week-old female mice (NRGS strain: *NOD.Cg-Rag1[tm1Mom]Il2rg[tm1Wjl]*, DKFZ breeding), as described previously[17]. All animal experiments were performed in accordance with ethical and legal regulations for animal welfare and approved by the governmental council (Regierungspräsidium Karlsruhe, Germany). Female mice were selected for this study to allow for effective randomization into groups with comparable tumour sizes following the first positive MRI measurements. Male mice would have been less suitable for group housing due to their potential aggressive behaviour. Tumour growth was followed by Magnetic Resonance Imaging (MRI) and animals were randomised into treatment groups based on tumour volume measurements. A maximum tumour volume was not defined as a termination criterion. The criteria for terminating animal experiments, as approved by the animal protocol, were all strictly adhered to and involved regular monitoring for the following defined symptoms: skull bulging, ataxia (impaired balance and movement indicative of brain damage), hyperactivity, central or peripheral paralysis, reduced movement, lack of food or water intake, behavioural signs of pain, and weight loss exceeding 20%. Animals were euthanized under deep anesthesia by gradual $CO_2$ exposure followed by decapitation. Housing conditions for the mice included a 12-hour light/12-hour dark cycle, an ambient temperature of 20–24 °C, and relative humidity of 45–65%.

**In vivo radiation.** Prior to cerebellum particle irradiation, mice were anaesthetized by inhalation anaesthesia, a mixture of 2% isoflurane with 2 L/min flow rate compressed medical air. Six anaesthetized mice were placed and fixed on a specifically designed acrylic glass frame on which the beam plans irradiated each of the target volumes individually, without any overlap between the six subfields. The whole cerebellum, 7×7 mm region around the cell injection site was irradiated with 3 Gy of carbon ions or protons for 5 consecutive fractions at the Heidelberg Ion Beam Therapy Center (HIT). The horizontal beams were oriented in the ventral direction, within a SOBP of 6 mm at a water equivalent depth of 122 mm with PMMA shielding of 10 cm thickness (linear energy transfer, carbon: 102 keV/μm, range 84–144 keV/μm). For the radiation experiment with PARP inhibitor, 6 mg/kg of BGB290 (Pamiparib) was administered p.o. (in 0.5% methylcellulose) twice a day for 9 days (two days before, during and two days after irradiation).

**Magnetic resonance imaging (MRI).** Imaging was carried out by the DKFZ small animal imaging core facility using a Bruker BioSpec 3 T (Ettlingen, Germany) with ParaVision software 360 V1.1. Before imaging, mice were anaesthetised with 3.5% sevoflurane in air. For lesion detection, T2 weighted imaging was performed using a T2_TurboRARE sequence: TE = 48 ms, TR = 3350 ms, FOV 20×20 mm, slice thickness 1.0 mm, averages = 3, Scan Time 2 min 40 s, echo spacing 12 ms, rare factor 10, slices 20, image size 192×192. MRI tumour volume was determined by manual segmentation using Bruker ParaVision software 6.0.1.

**Hematoxylin and eosin stainings, immunohistochemistry and immunofluorescence.** All stainings were performed on 5 μm formalin-fixed paraffin-embedded sagittal mouse brain sections. Hematoxylin and eosin (HE) staining was done using standard procedure. For STEM121 (1:5000, Takara, #Y40410) and Ki67 (1:100, DAKO, #M7240) immunohistochemistry, brain sections were deparaffinized and cooked for 18 min in Tris-EDTA pH 8.0. Slides were washed in PBS, incubated in 3% hydrogen peroxide for 10 min, blocked with 1% BSA for 30 min and incubated with primary antibody diluted in 1% BSA overnight at 4 °C. Supervision 2 HRP-polymer kit (DCS, #PD000POL) was applied the next day. Imaging was performed on Axio Zeiss Imager.M2 microscope.

**Quantification of immunofluorescence signals.** The tissue sections used for the Visium spatial experiments were stained with DAPI Fluoromount G and Ki67-Cy5 (1:100 DAKO #M7420). The individual channels corresponding to DAPI and Cy5 were scanned and saved as TIFF files. The channels were merged to form a composite image using imageJ. The images were then divided into 3 slices for further analysis. A macro was developed to identify Ki67 positive nuclei. The macro subtracted background from the DAPI channel, applied gaussian blur on the nuclei and the nuclei were segmented by finding the centres of each DAPI stained nucleus. Background subtraction was performed for the Cy5 channel subsequently and a manual thresholding was applied to select positively stained nuclei. All the single points within the centre of the DAPI signal were included in the analysis. The final output was the total number of DAPI positive cells and the Cy5 positive cells within the area of the DAPI segmented regions. Each data point corresponds to the percentage of Ki67 positive cells in a region of an image taken for analysis. Each sample consists of 30 data points where 10 regions in three image slices were taken for analysis. The macro was used on the entire tissue area of samples LFS4 and LFS7 due to the small size of the tissue where only one data point was plotted for each sample. Samples LFS5, LFS6, S4 and S5 were excluded from the image analysis due to several artefacts in staining, small tissue size and high background signals.

## Visium data processing and quality control

Spots covering the tissue were manually annotated using the 10x Loupe software. For human data, FASTQ sequence files were processed with the 10x SpaceRanger pipeline (version 1.3.1) using a pre-built human GRCh38 reference genome downloaded from the 10x Genomics website. For PDX data, reverse read FASTQ files were first separated into human and mouse reads using xengsort[43], a k-mer based read sorter specifically developed for PDX sequencing experiments, with the human GRCh38 and mouse mm10 reference genomes downloaded from the 10x Genomics website. Spatial barcodes contained in forward read FASTQ files were separated according to the xengsort results using a custom script. FASTQ files with human reads were then processed as above.

Further analysis was performed in Python using scanpy[37]. A standard scanpy QC workflow was applied and spots with low numbers of reads or detected genes were excluded from the analysis. Thresholds were determined individually for each sample based on read/gene count histograms to exclude background noise from empty droplets or ambient RNA while retaining as much signal as possible. Similarly, genes detected in less than 10% of spots (human samples) or less than 10 spots in total (PDX samples) were excluded from further analysis.

## Visium cell type deconvolution

Deconvolution of Visium data was performed using cell2location[23]. For human samples, we used the annotated primary tumour snRNAseq reference dataset that we generated previously[12] containing 10 cell types, of which 6 are malignant. For PDX samples, we used both the primary tumour snRNAseq and the PDX scRNAseq dataset that we generated previously[12], correcting for technology differences using the categorical_covariate parameter of cell2location[23]. Cell type annotation in both reference datasets was performed by identifying differentially expressed genes in clusters of cells using a two-sided Mann-Whitney U-test with Benjamini-Hochberg multiple testing correction followed by literature-based assignment of cell types to clusters based on the identified marker genes[12]. The PDX scRNAseq reference contains 8 cell types, of which 6 are malignant, and 3 cell types intersect with the snRNAseq reference.

Since neither reference dataset contained annotations for Pro- and Anti-Tumor Macrophages, we subclustered the Macrophages cluster of the primary tumour snRNAseq dataset prior to performing deconvolution. We performed Leiden clustering[35] of the Macrophages

based on expression of Pro- and Anti-Tumor Macrophage marker genes (*CD163, MRC1L1, HLA-DMA, HLA-DMB, HLA-DOA, HLA-DOB, HLA-DPA1, HLA-DPB1, HLA-DQA1, HLA-DQB1, HLA-DRA, HLA-DRB1, CD80, CD86*). This separated the Macrophages into three well-defined clusters, with cluster 1 expressing *CD163* but not *HLA-DRB1*, cluster 2 expressing *HLA-DRB1* but not *CD163*, and cluster 3 not expressing any of the marker genes. Cluster 1 was annotated as Pro-Tumor Macrophages, cluster 2 as Anti-Tumor Macrophages, and cluster 3 as (unpolarized) Macrophages.

### Spatial correlations between cell types
Spatial correlations were calculated by fitting a Gaussian Process model to cell2location deconvolution results as described previously[44].

### Integration of PDX samples
PDX samples were integrated using scVI[45].

### Spatial clone mapping
Clones in Visium data were assigned based on the expression levels of genes that were previously found to differ in copy number between clones[12] in matched samples. For the human data, we used all genes on chromosomes 8 and 12. We used only those genes to perform spatial clustering with SpatialDE2[22] and adjusted the clustering parameters such that only two clusters were obtained. We found that the average expression of genes on chromosomes 8 and 12 was consistently higher in one of the clusters, confirming that the clustering did indeed separate clones and allowing us to assign clone labels to clusters. As an additional control, we performed Leiden clustering of Visium spots using only the expression levels of genes on chromosomes 8 and 12. Again, we adjusted the clustering parameters to obtain only two clusters. The overall pattern (see Supplementary Fig. 16) was very similar to the SpatialDE2 clustering, and we used the latter for further analysis.

PDX samples were analysed as above, but using the following genes, which were identified as located in chromosome regions that differ between the two major clones based on previous single-cell DNA sequencing data for the same samples: *HOXC13, HOXC11, NAB2, GLI1, DDIT3, COL2A1, PRPF40B, SMARCD1, ATF1, NACA, STAT6, CDK4, LRIG3, WIF1, PTPRB, KMT2D, ERBB3, HMGA2, ARID2, MDM2, BTG1, USP44, CHST11, SH2B3, ALDH2, PTPN11, TBX3, HNF1A, SETD1B, BCL7A, ZCCHC8, POLE, CLIP1, NCOR2, ARHGEF10, LEPROTL1, NRG1, FGFR1, IKBKB, HOOK3, HEY1, MYC, RECQL4, PLAG1, CHCHD7, COX6C, RAD21, NDRG1, TCEA1, PREX2, NBN, RUNX1T1, CDH17, UBR5, EIF3E, EXT1, NCOA2, CNBD1, PABPC1, CSMD3, FAM135B, RSPO2*

### Cell-cell communication analysis
We used COMMOT[26] with the built-in human CellChat ligand-receptor database. We set the maximum communication distance to 5 times the minimal distance between spot centres.

### Differential gene expression analysis
DGE comparing LFS with sporadic patient samples was performed in pseudo-bulk (using the sum of counts for each gene across all spots in a sample) using DESeq2[46]. False discovery rate was controlled using the Benjamini-Hochberg procedure with independent hypothesis weighting[47]. DGE comparing minimal residual disease with regrown tumours in the PDX dataset was performed between region 3 of sample PDX B (normal tumour), region 0 of sample PDX D1 (resistant), and region 0 of sample PDX E2 (regrown). Each Visium spot was treated as a replicate. DGE analysis was performed in R using DESeq2[46]. False discovery rate was controlled using the Benjamini-Hochberg procedure.

### Nuclear morphology quantification
Assignments of spots to clones based on (spatial) clustering were somewhat noisy. To generate contiguous tissue areas belonging to individual clones, nearest-neighbour graphs of Visium spots belonging to each clone were constructed. The largest connected component of each graph was defined as the tissue area belonging to that clone, and its outline was defined as the concave hull of its Visium spots calculated using the alphashape Python package.

To obtain segmentation masks and locations of cell nuclei, the DAPI-stained image was segmented using Cellpose[40,41] using its pre-trained 'nuclei' model. Segmentation masks were assigned to the clone whose tissue area contained the mask's centroid. The area of a nucleus was defined as the area of the corresponding segmentation mask.

### Deconvolution of immune cell types in tumours
Immune cell infiltration was inferred using CIBERSORTx[48] where parameters were set as default. LM22 was used as the single-cell signature matrix to impute the fractions and types of immune cells. The *TP53* status and the corresponding bulk gene expression profiles of the medulloblastoma patients (n = 763) were derived from Cavalli et al[5]. and Gene Expression Omnibus (GEO: GSE85217) respectively. For the pan-cancer immune cell infiltration analysis, we retrieved the transcriptomic profiles from The Cancer Genome Atlas[49,50]. The *TP53* status was derived from PanCanAtlas (https://gdc.cancer.gov/about-data/publications/pancanatlas). The patients with silent *TP53* mutation were excluded from the analysis. Wilcoxon rank-sum test was used to test the immune cell infiltration difference between the *TP53* mutant and *TP53* wild-type tumours.

### Copy number analysis from whole genome sequencing
The somatic copy number ratio was estimated using the GATK 4.1.3.0 workflow[51]. Briefly, the human reference genome (hg37) was partitioned into intervals with bin length equals 10000 bp. The read counts from the patient-derived xenograft samples (shortly after irradiation and long-term after irradiation) were collected based on the interval list. The untreated control patient-derived xenograft sample was used as a reference. The copy number ratios of the tumours were then denoised (DenoiseReadCounts) and segmented (ModelSegments). The alternate segments were coloured in grey and white; the medians of the segments were shown as black lines.

### Statistics & Reproducibility
No statistical method was used to predetermine sample size for spatial transcriptomics experiments. Two human samples were excluded from the analyses due to poor quality. All animals used for treatment experiments with carbon ions and PARPi were randomised into treatment groups based on tumour volume measurements. The investigators were not blinded during animal follow up and outcome assessment. The statistical tests used are indicated in the respective figure legends and in the main text, where appropriate.

### Reporting summary
Further information on research design is available in the Nature Portfolio Reporting Summary linked to this article.

## Data availability
The sequencing data generated in this study have been deposited at the European Genome-phenome Archive with accession number EGAS00001007128. The data is available under restricted access because partly personal data cannot be publicly available, due to the European General Data Protection Regulation (GDPR) and the German General Data Protection Regulation (GDPR) and to respect the patient consent forms. For data sharing, a Data Transfer Agreement (DTA) has to be legally settled between the requesting institute and the providing institute. Once the data access has been granted, the access is usually available for 5 years, depending on the individual patient consent forms. We will do our best to process the requests and get DTAs in place as fast as possible. The remaining data are available within the Supplementary Information. Source data are provided with this paper.

## Code availability

Jupyter notebooks reproducing the analyses are available at https://github.com/ilia-kats/medulloblastoma-paper and on Zenodo[52] (https://doi.org/10.5281/zenodo.13933624).

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

## Acknowledgements
The DKFZ Light Microscopy facility, and especially Damir Krunic, is acknowledged for help with macro design. Hermann Stammer, Viktoria Eichwald, Inna Babushkina, and Sabrina Kirschner are acknowledged for help with MRI. The DKFZ NGS Core facility is acknowledged for support with sequencing. Caroline Pabst is acknowledged for sharing NRGS mice and Kerstin Dell for advice on animal experiments. Stephan Brons is acknowledged for support of the irradiation experiments and Aleksandra Jurek for advice for the spatial transcriptomics experiments. We thank Norman Mack for support of the mouse experiments, Moritz Przybilla for sharing annotations, Aurelio Teleman and Peter Lichter for discussions. Aurélie Ernst received financial support from the German Cancer Aid and from the German Research Foundation (DFG). Oliver Stegle was supported by the German Federal Ministry of Education and Research (BMBF) through the project "DeepSC2 - Bioinformatische Methoden auf der Basis von tiefen Netzen für die Analyse einzelner Zellen in der Onkologie" (031L069A) and by Wellcome Leap as part of the Delta Tissue programme in collaboration with The Wellcome Sanger Institute, The Francis Crick Institute and The University of Cambridge.

## Author contributions
Ilia Kats performed the vast majority of the computational analyses and generated figures. Milena Simovic-Lorenz performed the xenograft experiments, animal treatment, follow-up of the animals, immunofluorescence analyses and generated figures. Hannah Schreiber performed the spatial transcriptomics experiments, immunofluorescence analyses, contributed to the follow-up of the animals and generated figure panels. Pooja Sant and Jan-Philipp Mallm contributed to the spatial transcriptomics experiments. Albert Li performed deconvolution analyses from bulk RNA sequencing and whole-genome sequencing analyses. Verena Körber performed the cancer cell fraction calculations. Pravin Velmurugan performed immunofluorescence analyses and quantifications and generated figures. Sophie Heuer performed tumour microtube quantifications. Luisa Kües performed live-cell imaging of medulloblastoma cells. Frauke Devens performed the FISH analysis. Martin Sill performed the analysis of DNA methylation array data. Manfred Jugold supported the MRI analyses. Mahmoud Moustafa supported the carbon ion irradiation experiments. Amir Abdollahi supervised the carbon ion irradiation experiments. Frank Winkler supervised the tumour microtube analyses. Andrey Korshunov performed neuropathological evaluation of the tumours. Stefan Pfister contributed to conceptualization and interpretation. Oliver Stegle and Aurélie Ernst conceived the study, jointly supervised the work and acquired the financial support for the project (experimental: A.E., computational: O.S.). All authors discussed the results and contributed to the final manuscript.

## Funding

## Competing interests
O.S. is a paid advisor of Insitor.INC. The remaining authors declare no competing interests.

## Additional information

Ilia Kats [1,22], Milena Simovic-Lorenz [2,3,22], Hannah Sophia Schreiber [2,3,4,22], Pooja Sant [5], Jan-Philipp Mallm [5], Verena Körber [6], Albert Li [2,3], Pravin Velmurugan [2,3,7], Sophie Heuer [3,8,9,10], Luisa Kües [3,8,9,10], Frauke Devens [2,3], Martin Sill [3,11,12], Manfred Jugold [13], Mahmoud Moustafa [3,14,15,16,17], Amir Abdollahi [3,14,15,16], Frank Winkler [3,8,9,10], Andrey Korshunov [3,11,18,19], Stefan M. Pfister [3,10,11,12,20], Oliver Stegle [1,21] ✉ & Aurélie Ernst [2,3] ✉

[1]Division of Computational Genomics and Systems Genetics, German Cancer Research Centre (DKFZ), Heidelberg, Germany. [2]Group Genome Instability in Tumors, German Cancer Research Centre (DKFZ), Heidelberg, Germany. [3]German Cancer Consortium (DKTK), Heidelberg, Germany. [4]Faculty of Medicine,

Heidelberg University, Heidelberg, Germany. [5]Single Cell Open Lab, German Cancer Research Center (DKFZ), Heidelberg, Germany. [6]MRC Molecular Hematology Unit, Weatherall Institute of Molecular Medicine, Radcliffe Department of Medicine, University of Oxford, Oxford, United Kingdom. [7]Faculty of Biosciences, Heidelberg University, Heidelberg, Germany. [8]Neurological Clinic, Heidelberg University Hospital (UKHD), Heidelberg, Germany. [9]Clinical Cooperation Unit Neurooncology, German Cancer Research Center (DKFZ), Heidelberg, Germany. [10]National Center for Tumor Diseases (NCT), Heidelberg, Germany. [11]Hopp Children's Cancer Center Heidelberg (KiTZ), Heidelberg, Germany. [12]Division of Pediatric Neurooncology, German Cancer Research Center (DKFZ), Heidelberg, Germany. [13]Core Facility Small Animal Imaging Center, German Cancer Research Center (DKFZ), Heidelberg, Germany. [14]Clinical Cooperation Unit Translational Radiation Oncology, National Center for Tumor Diseases (NCT), Heidelberg University Hospital (UKHD) and German Cancer Research Center (DKFZ), Heidelberg, Germany. [15]Heidelberg Institute of Radiation Oncology (HIRO), National Center for Radiation Oncology (NCRO), German Cancer Research Center (DKFZ), Heidelberg, Germany. [16]Division of Molecular and Translational Radiation Oncology, Heidelberg Faculty of Medicine (MFHD), Heidelberg University Hospital (UKHD) and Heidelberg Ion-Beam Therapy Center (HIT), Heidelberg, Germany. [17]Department of Clinical Pathology, Suez Canal University, Ismailia, Egypt. [18]Clinical Cooperation Unit Neuropathology, German Cancer Research Center (DKFZ), Heidelberg, Germany. [19]Department of Neuropathology, Heidelberg University Hospital (UKHD), Heidelberg, Germany. [20]Department of Pediatric Oncology, Hematology and Immunology, Heidelberg University Hospital (UKHD), Heidelberg, Germany. [21]Genome Biology Unit, European Molecular Biology Laboratory (EMBL), Heidelberg, Germany. [22]These authors contributed equally: Ilia Kats, Milena Simovic-Lorenz, Hannah Sophia Schreiber. ✉e-mail: o.stegle@dkfz.de; a.ernst@dkfz.de

