## [Transparent Peer Review file · Nature Communications]

Spatio-temporal transcriptomics of chromothriptic SHH-medulloblastoma identifies multiple genetic clones that resist treatment and drive relapse

Corresponding Author: Dr Aurelie Ernst

Version 0:

Reviewer comments:

Reviewer #1

(Remarks to the Author)

This study of Kats and Schreiber et al., investigated the molecular makeup of chromothriptic medulloblastomas, a type of pediatric brain tumor with a poor prognosis. Using spatial transcriptomics, the researchers found higher intra-tumor heterogeneity in chromothriptic medulloblastomas, which is linked to increased proliferation and stemness but decreased immune infiltration and differentiation. The study also identified a potential role for tumor microtubes in treatment resistance, suggesting cell network communication as a possible target for therapy.

The study employs cutting-edge analysis of spatial data, and the manuscript is well composed. However, there are some challenges in interpreting the data. Given that chromothriptic medulloblastomas exhibit variable copy number alterations (CNAs), it is important to determine whether transcriptional heterogeneity arises from CNAs or transcriptional diversity. Spatial transcriptomic data could potentially be used to infer CNAs, which would help identify clonal expansion and CNAs for each clone. As a subsequent step, it would be valuable to assess the chromosomes to which differentially expressed genes belong. Are they randomly distributed across chromosomes or do they follow differences in CNAs?

The role of midkine signaling, which was demonstrated to be elevated in LFS samples, requires further clarification or investigation. The authors mention using an algorithm to identify a signaling pathway but do not elaborate on the details. The identity of the sender- and receiver cells should also be clarified. What is the specific role of midkine signaling, in-silico perturbation in sc data?

Again, the spatial mapping of clones remains ambiguous. It would be beneficial to utilize state-of-the-art methods for this purpose, such as inferCNV or others (Erickson, A., He, M., Berglund, E. et al. Spatially resolved clonal copy number alterations in benign and malignant tissue. *Nature* 608, 360–367 (2022). <https://doi.org/10.1038/s41586-022-05023-2>). The authors state that they used the expression levels of genes located on chromosomes 8 and 12 to reconstruct the spatial positions of these clones; however, it is unclear whether the single-cell DNA sequencing (scDNA seq) data originated from the same patients. How was it ensured that the same CNAs, and not additional ones, were present in the spatial data? FISH was only conducted for chromosomes 8 and 12.

TMTs have been shown to serve distinct functions in glioblastomas, such as connecting cells through calcium signaling. The role of TMTs in medulloblastomas remains uncertain. Are medulloblastomas also interconnected in functional networks?

Reviewer #2

(Remarks to the Author)

Medulloblastomas (MB) with chromothripsis results in much worse prognosis than MB without chromothripsis. Patients with Li Fraumeni syndrome (LFS) develop MBs with chromothripsis. In this manuscript, Kats et al., report that there is a marked difference between LFS and sporadic (S) medulloblastoma in intra-tumoral heterogeneity by examining 8 LFS and 5 S medulloblastoma samples by spatial transcriptomics. Although the two tumor types show some differences such as immune cell infiltration and increased fraction of variance attributable to spatial location, the evidence supporting marked differences between the tumor types is weak. In addition, several comparisons need statistical analyses.

Below are specific points. Addressing these points may strengthen the author's conclusion.

1. Fig S4 does not show higher heterogeneity in the LFS group than in the S group.
2. Authors report that “massive rearrangements due to chromothripsis were detected in all LFS cases, in stark contrast with the relatively small number of copy number variations detected in the SHH-TP53wt tumours from the control group (Suppl. Fig. 1)”.
 S4 and S5 appear to have much more rearrangements than LFS1 that has relatively few rearrangements. Therefore, it is unclear whether LFS1 is a medulloblastoma with chromothripsis and whether S4 and S5 are medulloblastomas without chromothripsis.
3. It is unclear whether LFS tumors show stronger immune cell exclusion than S tumors do because Fig 1, Fig 2B, and Fig S7 do not have statistical analyses.
4. “Major drivers of SHH-medulloblastoma such as MYCN and GLI2 showed higher expression levels in LFS medulloblastomas but were also characterised by increased spatial expression heterogeneity, with regions characterised by high expression of one oncogene but lower expression of the other, and vice versa (Fig. 3A, Suppl. Fig. 8-10).”
 Fig. 3A and Supple Fig. 8-10 do not support higher spatial expression heterogeneity in LFS than in S medulloblastoma. Fig. 3B shows that “malignant cell clusters of SHH-high” and “cycling cells” co-localize but “neuronal development clusters” are located in separate regions in both LFS and S medulloblastoma.
5. “Collectively, our results point to increased spatial intra-tumour heterogeneity, elevated signatures of proliferation and stemness, reduced immune infiltration and differentiation in LFS medulloblastomas compared to sporadic tumours.”
 Data are insufficient to support intra-tumor heterogeneity, elevated signatures of stemness, and reduced differentiation in LFS than in S medulloblastoma. Stemness is only inferred from Sox2 expression level. Fig 1D shows the increased fraction of malignant neuronal differentiation development II in LFS, suggesting increased differentiation rather than decreased differentiation in LFS. Reduced immune filtration was not shown by statistical analyses.
6. “we utilised single-cell DNA sequencing data from a primary-relapse pair of LFS medulloblastomas [15], which had been generated previously, to measure copy number profiles.”
 Reference 15 does not have single-cell DNA sequencing data. It is unclear how two clones were detected and how their relationship was determined. The right panel of Fig. 4A shows that the clones have different CNV rather than lineage relationships. If clone A is a descendent of clone B, why clone A does not have a CNV found in clone B?
7. It is hard to see individual cells in Fig 4C. Fig 4A shows that clone B not clone A is polyploid; however, Fig 4C shows higher polyploidy in clone A.
8. “Hence, the spatial mapping of genetic clones identified phenotypic features associated with each clone, with phenotypic variation between clones molecularly related to the differences detected between LFS and sporadic tumours (c.f. Fig. 1, 3).”
 Do authors suggest that an increased number of different clones in LFS than in S medulloblastoma underlie the differences between LFS and S medulloblastoma? If so, more evidence is needed from both LFS and S medulloblastoma than evidence from one LFS sample. Moreover, S medulloblastomas also show the spatial separation of cycling cells from differentiating cells.
9. “the PDX models more closely resembled the spatial patterns observed in the corresponding human tumours than other non-matched LFS tumours (Fig. 6A, Suppl. Fig. 15), supporting the faithfulness of these in vivo models.”
 The spatial patterns of all LFS and S tumors look very similar except for high values for negative correlations in LFS5. Thus, it is unclear whether the PDX model specifically resembles the original human tumor.
10. Authors wrote that Nes is a tumor microtubule marker. What is the evidence for Nes being a tumor microtubule marker? Nes is an intermediate filament protein that is expressed in neural progenitor cells and labels the whole cell including processes.
11. “nestin showed a higher expression in human LFS as compared to sporadic medulloblastoma”
 Fig 8E shows that Nes expression levels vary highly in LFS group and in S group rather than vary between LFS vs. and S groups.
12. Fig 1A shows extreme examples of LFS and S samples rather than representing each group.

Reviewer #3

(Remarks to the Author)

Overview:

In this work, the authors aim to learn something about what makes chromothriptic MB tumors exhibit poor prognosis relative to MB cases without chromothripsis. This is an intriguing angle of investigation, potentially revealing how chromothriptic events can shape evolutionary trajectories and tumor-cell phenotypes to yield more aggressive cancers compared with those without such rearrangements. The effects of specific chromothriptic alterations (copy number gains and structural rearrangement of genes for ex on double-minute chromosomes) on gene expression programs remains to be fully understood. Profiling of single cell and spatial transcriptome datasets has the potential to reveal how expression programs are perturbed, and how tumor cells interact with each-other and cells of the tumor microenvironment as a result. These findings would be of high interest and significance to the cancer research community.

To achieve this, the authors set out to profile MB tumors from the same molecular subgroup and compare chromothriptic and non-chromothriptic cases. They use spatial transcriptomics to perform the majority of the work, linking to results from previously published datasets when needed.

I was initially excited by the novel application of spatial profiling to this problem, however, there are a number of significant limitations of the work presented, which prevent the original research question from being addressed. I will summarize these below in 3 main categories:

1. Cohort, and the potential significance of the results:

a. To understand what makes tumors with chromothripsis so aggressive, the authors profile a set of TP53-germline mutant (LFS) cases all with chromothripsis, and compare these to TP53-wt SHH MB (no chromothripsis). A solid rationale for the selection of this case-control strategy is not presented. For the cases: there are SHH TP53-mut (somatic) cases with chromothripsis which could be included, not just LFS tumors. Given the focus on chromothriptic SHH tumors (see title of manuscript and abstract), the current cohort of cases is not comprehensive, and may shed light only on LFS tumor biology rather than chromothriptic tumors more generally. For the controls: why did the authors choose to profile TP53-wt cases, rather than TP53-mut (somatic) SHH tumors without chromothripsis? A well-stated rationale is needed to understand this, and I am not convinced the comparison is appropriate. Molecularly and in terms of treatment and survival, the 4 MB subgroups and 12 subtypes are quite distinct (see Cavalli paper Cancer Cell 2017). This is recognized in the manuscript (work on lines 167-180) and in Fig 2, as the authors perform comparisons among subgroups and subtypes. However, this is not accounted for in the cohort selection. The LFS tumors are likely SHHalpha, while the TP53-wt patients belong to other subtypes (n=5 patients of which 3 are infants, and likely SHHbeta or gamma, while the other two are not infants, subtype unclear). Given the heterogeneity of the control group, the study does not seem powered to address the main question of interest. A cohort of non-chromothriptic TP53-mut tumors (SHH-alpha subtype) would provide a better comparison, as the other aspects of MB biology would be most similar. Second-best would be other SHHalpha tumors that are TP53-wt. The Methods in this manuscript state that methylation-based classification was performed, but those results are not included, and it's unclear whether the tumor subtypes of the included cases are known.

b. The PDX cohort is meant to reveal mechanisms of resistance to treatment, and 11 samples are profiled along a transition from pre-treatment to post-treatment, MRS, and regrown tumors. Major conclusions about resistance are presented in the abstract and discussion, yet the data do not support these statements. Pitfalls are that all 11 samples profiled derive from a single LFS patient, and that is not sufficient to generalize from. Importantly, the sample used for the treatment experiments is from an already-treated patient sample (although no details on that treatment are provided). Why the investigators chose this strategy is perplexing, especially since a PDX line from the same patient's primary tumor was also available. That would have at least provided a relevant (yet n-of-1) result to build upon. As such, the observation of multi-clonal tumor regrowth in the PDX (clones A and B) is without significance, since these clones have already survived therapy in the patient. Given that the supervised methods used in the manuscript can only measure the presence of the major clones A and B, there is no chance of discovering additional clones that may arise through the selective pressure of treatment. Unsupervised methods could be used (inferCNV for instance), but results would still be limited to a single and already-treated patient. The relevance and impact of the PDX work as presented is therefore quite low.

2. The quality of the data and analyses: the use of appropriate analyses, careful interpretation, and sufficient detail is lacking.

a. There are almost no p-values or fold changes presented in the manuscript, yet strong statements like on line 152 are made throughout: "This analysis revealed a markedly higher number of distinct tissue zones in LFS medulloblastomas, indicating a higher degree of spatial heterogeneity in this subtype (Fig. 1A..". What does markedly higher mean? Is the difference significant? This lack of statistical rigour is prevalent throughout the manuscript. In this particular case, we're directed to look at Fig 1A as support for this conclusion of higher diversity of tissue zones in LFS cases. Fig 1A shows us two "representative examples" (LFS3 and S1) with spatial clusters, rather than case vs control statistical support. Looking at the supplementary data, it seems that S1 is not actually representative of the sporadic tumors; instead, it is the most homogeneous case. In general for most analyses, the presented data does not support a strong and clear story.

Another example on line 207: "we assessed pairwise spatial correlations between all major cell types, which identified both negative and positive correlations in LFS medulloblastoma, whereas sporadic medulloblastomas were characterised by overall weaker evidence for spatial cell-type correlation, again supporting a more homogeneous spatial structure in sporadic medulloblastoma (Fig. 3C)." These plots look nearly the same. How is the above conclusion made, and how is "weaker" measured? Is the magnitude of difference significant? Can it be explained by any technical differences, such as the number of samples per group or the variable nr of spots per sample? How are those technical differences between cases and controls accounted for?

Like these examples, nearly every other comparative statement in the manuscript is descriptive rather than quantitative. Figures also fail to provide fold changes, p-values, or to offer quantitative assessments. This needs significant revision.

b. Spatial transcriptomics is a relatively new field, and analytic approaches are continually developed. While the wet lab-based techniques are adequately described in the methods, the data, QC, and analytic methods are not. I see that a github repository with code is available, but that is not a sufficient replacement for well-described methods. This currently leaves many open questions that can affect the interpretation of the results. For instance:

i. some of the LFS tumors have significantly fewer spots than other samples. How is this accounted for when analyzing results? Fewer spots may reduce the diversity within samples, affect the QC thresholds (i.e. min spots with gene

expression), unless one is analyzing LFS and sporadic cases separately – which could then increase the number of highly variable genes in LFS, and in turn introduce a technical batch effect between cases and controls. A clear description and a justification of the methodology used, and how that accounts for any batch effects is critical to include.

ii. In the PDX data, a big confounder is the mouse transcriptome. The methods state that only the human reads are analyzed, however, there will still be a large effect of the human-mouse admixture. Imagine that a spot has 50% human vs mouse reads, and another has 90% human and 10% mouse. Each spot gets 50,000 reads. This leaves ~25,000 human reads in the first spot and 45,000 human reads in the second. That is nearly a two-fold difference in sampling depth of the human tumor cells and compromises the ability to detect genes with moderate/low expression in the first spot, unless accounting for this in some way. What happens with these numbers during the subsequent analysis? Is power of detection accounted for at all? Without accounting for this, any comparisons between large tumors in the PDX models and MRD will be mostly reflecting the difference in power of detection (see Fig 8B volcano plots for an example of lack of power on the MRD side).

iii. Related to the above, Fig 6 shows that macrophages and microglia are detected and quantified in the PDX samples. Thus, either the methods are incomplete (and mouse data is indeed analyzed but not described), or there are human-mouse orthologs that are assorted to the incorrect genome. This would bin some mouse reads into the human genome pile, and lead to annotation of microglia and macrophages. Conversely, some human reads may end up in the mouse-genome pile and not be analyzed, raising issues of false negatives in the human tumor data. 10X Genomics provides a 2-genome reference that may help with this.

iv. It seems (from related work in ref 12) that scRNAseq data from 3 of the same patients is available. Why are these not used as validation in support of the main findings? Specifically, the effect of clonal architecture (clone A and B) on transcriptional programs, the diversity of cell states and types within tumors, and cell-cell communication patterns based on ligand-receptor analyses. This seems like a missed opportunity to add rigour to the work. Additional single cell profiling of controls would be warranted.

3. Result interpretation:

a. As alluded to above, there is over-interpretation of the results throughout the manuscript, and I will not be able to list all examples. A careful re-read of (i) what the experiments aim to show, (ii) actually show, and (iii) the stated conclusion is needed in each case.

i. One example (line 312): “To test whether the distinct genetic clones may be interdependent and require each other for optimal tumour growth...” is followed by an observation of clone distribution in the cerebellum vs frontal cortex. These observations shed no light on interdependence at all. Very different experimental testing would be needed to test that. Instead, the co-occurrence may simply reflect that clone A and B are not functionally distinct in terms of their invasion phenotype. Yet the conclusion is that “the distinct genetic clones may possibly benefit from each other, also for the migration to the frontal cortex”. A careful review of the initial statements that set up the experiments conducted and how they match up to the conclusions of the experiments is needed throughout.

b. Insufficient analysis and/or interpretation is also noted. For instance:

i. a cell-cell interaction analysis is performed with COMMOT, and midkine signalling is highlighted as important to LFS, plus a couple of other pathways in sporadic cases. This needs more detail (and quantitative metrics), including which receptors and ligands are involved, which cells are sending signals, and which cells are receiving them. Further, what is the implication for this difference in signalling in terms of biology or therapeutics?

ii. The tumor microtube work is done in PDX lines from one patient, and essentially identifies that TMs exist in the PDX. Yet this statement is made in the abstract: “Finally, we identified a potential role for tumour microtubes in treatment resistance in chromothriptic medulloblastoma, suggesting cell network communication as a putative target.”. That seems like a rather large leap to make in terms of interpretation, and an unsupported generalization to LFS. To establish the presence of TMs in medulloblastoma, patient samples should be investigated first. Are TMs present in MB samples in the primary and recurrent setting? Of relevance to this manuscript, are there differences between chromothriptic vs non-chromothriptic tumors? Is there any reason to suspect this could be the case, and serve as a rationale for this investigation? As presented, the TM work currently does not provide any meaningful insight into this type of connectivity in MB, or any specific relevance to chromothripsis. To further establish the role of TMs in treatment resistance a lot of functional work would be needed in vivo, using appropriate models, and seems out of scope for this manuscript. To further say TMs play a role in treatment resistance in chromothriptic MB would need a robust comparison to non-chromothriptic MB.

Other comments:

The supplemental figures other than SF1 were not labelled, making it quite difficult to match them up.

Version 1:

Reviewer comments:

Reviewer #1

(Remarks to the Author)

No further comments

Reviewer #2

(Remarks to the Author)

The authors addressed most of previous comments. Addressing the following will make the findings and conclusions more convincing and clearer.

1. “ Major cancer drivers as well as microenvironment-related genes such as proinflammatory factors showed higher expression levels in LFS medulloblastomas but were also characterised by increased spatial expression heterogeneity, with regions characterised by high expression of one gene but lower expression of the other, and vice versa(Fig. 3A, Suppl. Fig. S3.1-3.4, Suppl. Table S2). This heterogeneous pattern of expression of cancer drivers and microenvironment factors might contribute to tumour cells escaping the treatment in LFS medulloblastomas, with multiple drivers active in different tumour regions.”

 Increased spatial expression heterogeneity in LFS compared with sporadic tumor is not statistically significant. Therefore, increased expression levels rather than increased spatial heterogeneity might contribute to tumour cells escaping the treatment in LFS medulloblastomas.

2. “ Sporadic medulloblastomas were characterised by overall slightly weaker evidence for spatial cell-type correlation, again suggesting a more homogeneous spatial structure in sporadic medulloblastoma (Suppl. Fig. S3.5).”

 The number of sporadic tumor samples is less than that of LFS. Would this contribute to the overall slightly weaker evidence for spatial cell-type correlation in sporadic tumor?

3. “ In line with the GAP43 expression patterns, quantification of the tumour microtubule area per cell identified significantly larger tumour microtubule areas per cell in the untreated relapsed PDX as compared to the matched untreated primary tumours, and also showed a decrease in the tumour microtubules at the minimal residual disease stage and an increase in the regrown tumours (Fig. 7D). Therefore, tumour microtubules might potentially play a role in treatment resistance in LFS medulloblastoma.”

 If tumor microtubules play a role in treatment resistance, shouldn't they be increased in residual disease stage to provide resistance?

 The p-value for tumour microtubule areas per cell in the untreated relapsed PDX as compared to the matched untreated primary tumours is not shown.

4. “ The tumour microtubule marker nestin showed a significantly higher expression in the medulloblastoma subgroup enriched for LFS as compared to sporadic medulloblastoma,”

 Nestin expression is not significantly different between LFS and sporadic tumor.

5. What is “mean posterior Nes expression” in Fig. S7.4?

6. Statistical methods used may need to be reconsidered. For example, in Figure 1B, why one-sided not two-sided test was used? For Fig. 7D, statistical methods to compare multiple groups not t-test should be used; there are more than 2 groups to compare.

Reviewer #3

(Remarks to the Author)

The authors have addressed many of the comments in the first review, especially those related to cohort selection (case vs control sample and patient). I have no further major concerns on that front. The calcium signalling work presented for the TMs is also a valuable addition to the manuscript, and should be of interest to those in the brain tumor field.

However, there are still elements of the analysis (specifically for Visium data) and result interpretation which remain a concern, and need further attention, as I note below. Finally, some of the wording in the manuscript require changing, so as not to give the wrong impression to readers.

Line 74: “We conducted temporal profiling of 11 patient-derived xenografts from chromothriptic medulloblastomas, covering the transition from the minimal residual disease stage to treatment-resistant regrown tumours.”

- “11 patient derived xenografts” is not the same as “11 samples from a patient derived xenograft”, please change this wording, as it misleads readers right at the outset in thinking there are 11 independent PDXs.

Line 76: “In chromothriptic medulloblastoma, an ecosystem of cells from multiple genetic clones resisting treatment and leading to relapse highlighted the importance of multi-clone interplay.”

- Please remove “the importance of multi-clone interplay” from the abstract. This is not supported by data or analyses in the manuscript as far as I can tell. In fact, the main text has already been corrected to remove the statement that clone interplay plays a role in invasion, but it also needs to be changed in the abstract.

Fig 1C – “Comparative analysis of the cell type composition for a representative LFS (top) and sporadic medulloblastoma (bottom). Left, dominant cell type for each spot for representative LFS and sporadic medulloblastoma. Right, bar graphs displaying the cell type abundance for each region as in panel A.”

- The sporadic case selected for the figure (S1) is not representative, please select another one of the 5 sporadic cases presented in Fig S1.6.

Line 181 – “This deconvolution analysis indicated less CD8 positive T-cells in tumours displaying somatic mutations in TP53, suggesting a possible link between p53 dysfunction and immune exclusion, beyond the context of genetic predisposition (Fig. 2E, Suppl. Fig. 2.1D).”

- The summary of CD8 cell presence pan-cancer presented in Fig 2E seems very different than the cancer-specific results shown in Fig S2.1D left panel. There, only LAML looks different, while DLBC has the opposite pattern (more CD8 T-cells in the TP53-wt setting). Other brain tumors in this analysis (GBM, LGG) show no apparent difference in CD8 cell abundance with genotype. The number of TP53-mut cases per tumor type likely matters a lot here, so the statistics per cancer type are important to present. The TP53-mut CD8 T-cell association is likely very tumor type-dependent (or related to other immune-relevant contexts), and the current Fig 2E does not capture that at all. Can the Figure S2.1D panel be added to the main Fig 2E, along with p-values and patient/sample numbers.

Fig 3A and S3.1-3 have no p-values. A split violin plot might be a more effective way to present this data.

Line 195: “Markers previously associated with putative medulloblastoma stem cells such as SOX2 showed significantly higher expression levels in LFS medulloblastomas, whereas differentiation markers were expressed at significantly lower levels (Suppl. Fig. S3.4, S6.8, Supplementary Table S3).”

- It looks like the Visium DEG analysis was performed with DESeq2 using a pseudo-bulk approach, such that every gene has averaged expression across all spots in each sample. This doesn't take into account the tumor-normal admixture. Yet it is already established in the manuscript that the sporadic cases have more normal cells, and given this systematic difference between LFS and sporadic cases, one cannot confidently make the claims above (claims which are also repeated and emphasized in the abstract and discussion, and should be toned down). The differences observed could simply be due to the different levels of expression of these genes in normal cells.

- A great example of this issue is *Ndr2*, presented in Fig S3.1 as a gene more highly expressed in sporadic cases. The S5 tissue shows a pattern that is reminiscent of the cerebellar folding. Indeed, when looking in the mouse brain atlas for this gene's expression based on ISH (in situ histochemistry; <https://mouse.brain-map.org/gene/show/29546>), we see the same pattern for *Ndr2*. (note that corresponding spatial human data from the brain atlas was not available for this gene, but the is expected to be the same).

My conclusion here would be that the sporadic cases have a different level of expression of *NDRG2* because of differences in the normal cell types captured within the Visium sample, and not because the sporadic MB tumor cells are more differentiated. I am concerned that the analyses as currently presented do not sufficiently account for this aspect of the data, and the consequent conclusions are therefore potentially incorrect. Can the single cell data be used to more conclusively support the current claims? i.e. in tumor cells only, show that these genes are different, and present those along with the spatial data if there is agreement. Outlier samples, like S5 above, should have *Ndr2* expression in normal cells. Cell annotation is important, and even though another manuscript describing that data is in bioRxiv (since 2021, and presumably in a lengthy revision process), it is important for this manuscript to show or include some of the cell type annotation results, and certainly to briefly review the annotation approach in the methods.

Line 272: “Specifically, we selected PDX models from the relapse biopsy, and focused on combinations of particle radiation with PARP inhibitors (Fig. 5A). We showed previously that such combination treatments were successful in the context of primary chromothriptic medulloblastomas¹⁷.”

- It would be useful to rationalize the selection of this particular treatment (is it the only rational choice? The best choice in some way), and its application to recurrent disease (to put it in context with the reference regarding its use in primary disease).

Line 281: “To evaluate to which extent the heterogeneity observed in LFS patient medulloblastomas contributes to tumour relapse, we first compared the PDX models to the matched patient tumours. Notably, the two-dimensional neighbourhood structures from patient tumours were qualitatively preserved in matched PDX, indicating that these models globally reflect cellular interactions from human medulloblastomas (Suppl. Fig. S6.1A). Additionally, the PDX models more closely resembled the spatial patterns observed in the LFS medulloblastomas than sporadic tumours (Suppl. Fig. S6.1A-D), supporting the overall faithfulness of these in vivo models.”

- The work described in this paragraph does not “evaluate the extent to which heterogeneity contributes to tumor relapse”. It presents a qualitative interpretation of the neighbourhood analysis plot.

- Importantly, the PDX data is problematic here in the sense that mouse cells are not accounted for. Are the endothelial cells, meningeal cells, etc (shown in figure S6.1, and mentioned in the main text) human? Or mouse? I assume they are human from the methods and the author response, in which case I find the relevance confusing. For macrophages and microglia,

the relevance can be argued (though certainly mouse microglia and macrophages will likely outnumber the remaining human cells). However, there are also human endothelial cells shown in the same neighbourhood analysis, and these would presumably also be surviving in the PDX model – but likely not forming functional vessels. In that case, would we expect them to play the same role in a functional cancer ecosystem? (i.e. are they forming functional vessels in the mouse brain? That would be a surprising finding indeed.) Instead, I expect that mouse endothelial cells / vessels would serve the corresponding role, but those are not investigated here. Yet, despite the critical importance of the mouse cells in the growth of the PDXs, the correlations calculated and shown in Fig S6.1C are from the human data only, and include cell types we can't reasonably expect to be relevant in the PDX context. Overall, it seems a stretch to state that: "these models globally reflect cellular interactions from human medulloblastomas", especially since the majority of the TME (mouse) has not been surveyed.

o Can the authors change the neighbourhood analysis to either focus on the tumor cells, or to include the mouse TME. Alternatively, do not overstate the similarity in the TME between mouse and human.

- Please specify species (mouse vs human) in figures and text that reference non-malignant cells

Fig 7B Visium data-based DEG analyses: there is an unaddressed issue with depth of sampling, as brought up in my previous review comments. Currently, the volcano plots in Fig 7b are asymmetric, clearly indicating that the MRD samples are underpowered for gene expression quantification. See my original comment, with the example of high vs low tumor cell density. The low tumor cell density spots (MRD) will only capture the most highly expressed human genes, and suffer from sparse signal in a typical DEG analysis (because most genes will be signal dropouts). Thus, your biological variable (MRD vs non-MRD) is completely confounded with a key technical variable (read depth per gene). In our own work with Visium data with similar depth issues, we have applied Aldex2 to solve the imbalanced sampling problem. I recommend that you explore this approach with your data. All the statements that conclude some genes are downregulated at the MRD stage would need revising based on a more powered analysis.

Line 348: "Genes previously associated with a tumour microtubule connectivity signature in glioblastoma were downregulated at the minimal residual disease stage (enrichment tests for tumour microtubule genes in the Visium data, hypergeometric test: enrichment of TM-high genes in downregulated genes, tm_high enrichment pval: $1.63e-08$ and TM-low genes in upregulated genes, tm_low enrichment pval: 0.008)."

- This is a great example of the type of conclusion that cannot be made with the current analysis. It is more likely that this result is due to the sparse human transcriptome sampling in the MRD setting. This claim is not currently supported by the results presented.

Version 2:

Reviewer comments:

Reviewer #3

(Remarks to the Author)

The authors have satisfactorily addressed the concerns raised. I have just a few very minor comments left relating to the T cell figures:

1. Please make table S2.1E an excel supplemental table. This indicates case numbers and pvalues for TP53+ and – patients in various cancer types. Please add the fraction of T-cells in, or the fold change (or both). Also, this table describes data plotted in the main figure 2E (CD8 T cells). S2.1D is the fraction of regulatory T cells. Please add another table with these corresponding numbers. The figure legend in this case also needs updating (currently states T cells, not regulatory T cells).
2. Fig 2E – only a subset of the significant results are included, although the figure legend states that all significant values are labelled. This is not the case, as LAML has significantly higher CD8 T cell fractions in the TP53-wt cases – this should be annotated on the plot.
3. Fig S2.1D – add pvalues on the plot (for all sig differences), and provide a supplementary table with the relevant info.
4. "To test whether lower immune infiltration might be a feature of TP53 mutant tumours beyond medulloblastoma and possibly in the context of sporadic TP53 mutations as well, we re-analysed a pan-cancer cohort of 8,955 patients. This deconvolution analysis indicated less CD8 positive T-cells in tumours displaying somatic mutations in TP53, suggesting a possible link between p53 dysfunction and immune exclusion, beyond the context of genetic predisposition (Fig. 2E, Suppl. Fig. 2.1D and E)."

This statement seems to be an over-reach. At most, the deconvolution analysis shows that in a minority of tumor types, there are lower CD8 T-cell counts, but in the majority of tumor types there is no difference. Please revise the conclusion made above to more accurately reflect the results. There is probably some interesting biology to follow-up on in terms of the biology found – i.e. why do a handful of specific types of cancers have this relationship with TP53, but not others? Although this is outside of the scope of the main manuscript message, it could be interesting to speculate.

Point by point response to the Reviewers' comments:

Reviewer #1, expertise in scRNAseq/ST and brain tumours (Remarks to the Author):

This study of Kats and Schreiber et al., investigated the molecular makeup of chromothriptic medulloblastomas, a type of pediatric brain tumor with a poor prognosis. Using spatial transcriptomics, the researchers found higher intra-tumor heterogeneity in chromothriptic medulloblastomas, which is linked to increased proliferation and stemness but decreased immune infiltration and differentiation. The study also identified a potential role for tumor microtubules in treatment resistance, suggesting cell network communication as a possible target for therapy.

The study employs cutting-edge analysis of spatial data, and the manuscript is well composed. However, there are some challenges in interpreting the data. Given that chromothriptic medulloblastomas exhibit variable copy number alterations (CNAs), it is important to determine whether transcriptional heterogeneity arises from CNAs or transcriptional diversity.

We thank the reviewer for the thoughtful comments. Details on our examination and implementation of these suggestions are provided below.

We agree that the link between genetic and transcriptional heterogeneity is of interest. The assessment of the genetic basis is of course interesting and important but not the primary goal of this work. Work by others has reported a higher prevalence of clonal events in LFS solid tumors as compared to sporadic tumors (Light, ..., Malkin, Shlien, Nat Commun 2023), suggesting that the higher transcriptional heterogeneity in LFS medulloblastomas is not merely due to an increased number of genetic subclones.

Hence, even though we do not have the matched single-cell DNA sequencing data for all of the samples, which limits our ability to assess the number and structure of genetic clones present in each tumor (see one representative sample below in Rebuttal Figure 1), the lack of this information is not key for the major conclusions of our study, as prior work showed that the number of CNVs and genetic diversity is not increased in LFS as compared to sporadic tumors.

Also, more generally, it has been reported in different entities that CNAs and genetic diversity do only capture a small fraction of the transcriptional diversity observed in cancer (ref PMID: 29681456, PMID: 26902283). Multiome studies will be needed to understand better the impact of genetic variation on the transcriptional heterogeneity in the future. We have now adjusted the text to address this aspect (see revised discussion, page 14, lines 407-410).

Rebuttal Figure 1: Linking genetic heterogeneity to transcriptomic heterogeneity in LFS medulloblastoma. Left, clusters based on scRNAseq analysis. Right, genetic clone mapping from scDNAseq analysis with matched transcriptional cluster annotation. Normal cell types were used as a reference for the copy-number analysis and do not belong to any copy-number clone. Even though specific genetic clones within the tumor are enriched for distinct transcriptional programs, the relationships between genetic clones and transcriptional clusters is complex.

Spatial transcriptomic data could potentially be used to infer CNAs, which would help identify clonal expansion and CNAs for each clone.

We agree that the direct inference of genetic substructure from the spatial sequencing data is an interesting direction. In response to this comment, we applied inferCNV, CopyKat and Numbat - three alternative methods for CNV inference - on the Visium data (also suggested by reviewer #3, see below point 1b from reviewer#3, last paragraph).

Unfortunately, in the majority of samples, only the major clone could be retrieved by such methods (see rebuttal Figure 2 below). Comparing results from matched single-cell DNA sequencing for a subset of samples, which can serve as ground truth, we observed that smaller genetic clones could not be detected in the Visium data, most likely due to the intrinsic genomic resolution limits of such transcriptome-based CNV inference. In particular, we note that while transcriptome-based CNV inferences yielded overall concordant copy-number profiles, the resolution of the CNV profiles is markedly lower. In general, the size resolution obtained by copy-number inference tools from spatial omics is not optimal for de novo discovery of aberrations smaller than chromosome arm level changes, such as oscillations of copy-number states due to chromothripsis.

For the reasons outlined above, we argue that the analysis approach we presented in our initial submission, which is based on transferring the clonal annotation derived from scDNA-seq to the spatial omics, is currently the preferred option. Briefly, our approach is based on targeted clustering of Visium spots selectively considering the subset of genes that overlap with informative subclonal CNA events, thereby encouraging the identification of clone-associated transcriptional substructure.

In summary, even though we agree that in principle copy-number inference from spatial transcriptomic data is of high interest, for our specific cohort and biological question, it does not seem to be as informative as it has been in other contexts. Thus, we feel that the previous analysis, as presented, is currently the best we can do. We have decided to stick to this strategy as our primary analysis approach also in our revised paper.

Rebuttal Figure 2: Representative example of copy-number inference using Numbat (top) with matched heatmap from single-cell DNA sequencing data (bottom). Briefly, the minor clone without loss on chromosome 8 (bottom of the heatmap) is not detected by Numbat in the spatial transcriptomics data. The inability to resolve this clone is linked to failure to capture small copy-number events, such as oscillations due to chromothripsis on chromosome 8p, which are only visible in the single-cell DNA sequencing data from this PDX sample but not in the matched Visium data.

Having said this, we agree that additional validation and confirmation of our results is certainly warranted. We have considered two complementary approaches to further support our main results and conclusions. First, we used differential expression analysis for chromosomes 8 and 12 (using all genes on these two chromosomes, which show distinct copy-number states between the clones based on scDNAseq; c.f. Rebuttal Fig. 3).

Rebuttal Figure 3: The x axis shows the sum of normalized counts for the expression of genes on chromosome 8 and the y axis shows the sum of normalized counts for the expression of genes on chromosome 12. Importantly, the two clones (clone A and clone B, in blue and red, respectively) show distinct expression signatures for genes on chr 8 and 12, supporting the mapping.

Revised Figure 4B: clone map shown for reference.

Second, we have processed imaging-derived features of the nuclei from matched hematoxylin and eosin slides separately for each clone. The nuclei located in the major clones show significantly different morphological features (e.g., size, circularity), supporting an enrichment of distinct genetic clones in these regions. We revised Figure 4 to include these new results.

New panel added to revised Figure 4. Morphological feature analysis of nuclei supports the clone mapping results. Clone B contains more diploid cells and has smaller nuclei as compared to the daughter clone A, which has a higher fraction of polyploid cells, consistent with the larger nuclei.

As a subsequent step, it would be valuable to assess the chromosomes to which differentially expressed genes belong.

Are they randomly distributed across chromosomes or do they follow differences in CNAs?

To address this comment, we have assessed the overlap of genes that are differentially expressed between spatially distinct regions and copy number events in the same samples. For this analysis, we have initially considered two human relapse tumors, for which matched scDNA-seq data are available

(<https://www.biorxiv.org/content/10.1101/2021.06.25.449944v1>). In these samples we find that 9.6% and 8.5 % of spatial DE genes overlap a CNV event (considering aberrations on chromosomes 8 and 12, the major subclonal events; Rebuttal Figure 2). These fractions do not exceed chance expectation ($P > 0.47$; Chi-squared test). Similarly, when considering 11 PDX samples derived from one of these primary samples, we observe an overlap of 6.0-8.9% ($P > 0.03$ for each of the 11 samples).

We have also investigated whether genes that overlap with major subclonal CNV events on chromosomes 8 and 12 exhibit more pronounced differences (fraction of spatial

variance) across spatial locations compared to other genes. This analysis revealed at most a modest difference between chromosomes that harbour subclonal aberrations versus all other chromosomes (Rebuttal Figure 4).

Rebuttal Figure 4: Spatial variance component analysis for two LFS samples. Shown are violin plots that illustrate the fraction of gene expression variance that can be attributed to spatial location for genes on chromosomes 8 (green), 12 (red) and all remaining chromosomes (blue). No striking difference between groups was observed between chromosomes ($P > 0.02$, Mann-Whitney U-test).

The role of midkine signaling, which was demonstrated to be elevated in LFS samples, requires further clarification or investigation. The authors mention using an algorithm to identify a signaling pathway but do not elaborate on the details. The identity of the sender- and receiver cells should also be clarified.

We note that a similar point was also raised by reviewer #3, comment 3b i.

First, regarding the choice of algorithm, we use COMMOT (Cang et al, Nat Methods 2023), which assesses the co-expression of matching signalling and receptor genes in spatially proximal locations in our spatial omics data. The full details are described in the Methods (see page 32 and Table S4). Regarding the requested details and additional data, we now provide quantitative metrics that show the COMMOT rankings of the ligands and receptors for each sample. Briefly, the main ligand is MDK and the main receptors are NCL, SDS2 and LRP1 across samples (see new Supplementary Table 4 with all details on the COMMOT output).

The COMMOT output is based on Visium locations, not on cells. Therefore, we also used matched single-cell RNAseq data that we had previously generated to find the identity of the sender and receptor cells.

From the 10x scRNAseq data, the *MDK* ligand is expressed in malignant cells (SHH-high cluster in particular, which shows a much higher expression as compared to differentiated malignant cells and to cycling malignant cells). In terms of receptors, *LRP1* shows a very low expression (consistent with other single-cell studies, see PMID 37081869) and is expressed mostly in normal cells. In contrast, other midkine receptors such as *NCL* and *SDS2* are expressed at high levels across different malignant and non-malignant cell clusters.

We have now included these results in the revised version of our manuscript (see revised Figure 3 and Supplementary Figure S3.6).

Revised Figure 3 and Supplementary Figure S3.6: UMAPs based on 10xRNAseq data for nuclei (LFS medulloblastoma patients, left side) and PDX (right side) with cell type annotation and expression of the *MDK* ligand as well as major receptors for midkine signaling in LFS medulloblastoma.

Figure 3D from the initial manuscript shows the spots that send the midkine signal, we have now also added a map of the spots that receive this signal to the revised Figure 3 (revised Figure 3C below):

Revised Figure 3C. Cell-cell communication analysis visualizing spots that receive Midkine signaling in LFS medulloblastoma. Colour indicates the extent of signal for each Visium spot (from blue to red). Arrows indicate the overall direction of signalling.

What is the specific role of midkine signaling, in-silico perturbation in sc data?

As mentioned above, the predictions we report are not based on in-silico perturbations, but instead were identified based on co-expression signatures of spatially proximal locations.

Regarding the role of Midkine signaling - this pathway is known to be involved in cancer cell survival and proliferation, inflammation and angiogenesis, evasion of apoptosis, cancer cell invasion and metastasis, as well as immunosuppression (ref doi.org/10.1038/s41388-019-1124-8 and 10.3389/fimmu.2023.1145300).

Importantly, a study on single-cell RNA-seq and spatial transcriptomics showed that MDK-NCL signal is associated with an immunosuppressive environment in endometrial carcinoma (PMID 37081869). This is interesting in the light of our results on immune evasion in LFS medulloblastoma (see Figure 2). Future studies will reveal whether inhibiting Midkine signaling in medulloblastoma counteracts the immunosuppressive environment.

To further support the role of Midkine signalling in medulloblastoma, we have analyzed *MDK* expression in medulloblastoma in bulk RNA sequencing data, which shows that high MDK gene expression in the SHH alpha subtype (enriched for LFS medulloblastomas) is associated with shorter survival (see new Supplementary Figure including all four medulloblastoma subtypes, S3.7).

Again, the spatial mapping of clones remains ambiguous. It would be beneficial to utilize state-of-the-art methods for this purpose, such as inferCNV or others (Erickson, A., He, M., Berglund, E. et al. Spatially resolved clonal copy number alterations in benign and malignant tissue. *Nature* 608, 360–367 (2022). <https://doi.org/10.1038/s41586-022-05023-2>).

As suggested, we have invested significant efforts to apply and adapt inferCNV, CopyKat and Numbat to our Visium data. Unfortunately, we found these methods to be only of limited use. We kindly refer back to our response to point #2 above (page 2 and 3 of this letter), which addresses this comment as well.

The authors state that they used the expression levels of genes located on chromosomes 8 and 12 to reconstruct the spatial positions of these clones; however, it is unclear whether the single-cell DNA sequencing (scDNA seq) data originated from the same patients.

Yes, the single-cell DNA sequencing data are from the same patients, we have now clarified the methods and the results to make the section related to the spatial mapping of the clones unequivocal (see revised text, Methods, line 1006, and results, section “Spatial mapping of genetic clones”, line 242).

How was it ensured that the same CNAs, and not additional ones, were present in the spatial data? FISH was only conducted for chromosomes 8 and 12.

We cannot exclude the presence of additional CNVs in the spatial data, due to the fact that copy-number inference from spatial data does not retrieve small clones and has a limited size resolution (see above response to point 2). However, the FISH validation (using probes specific to loci for which different CNVs were expected between the major clones) showed that the spatial mapping of clones is valid at a broad scale, even though additional CNVs not detected by the FISH cannot be ruled out. We revised this result and toned down the corresponding sections (see revised discussion, line 410-412).

TMTs have been shown to serve distinct functions in glioblastomas, such as connecting cells through calcium signaling. The role of TMTs in medulloblastomas remains uncertain. Are medulloblastomas also interconnected in functional networks?

To address this question, we performed calcium activity imaging in live medulloblastoma cells (see revised Figure 7 and Supplementary Figures S7.5 and 7.6). Analysis of Ca²⁺ recordings

revealed a substantial fraction of cells exhibiting rhythmic Ca^{2+} activity, suggesting the possibility of a similar function for TMTs as in glioblastoma.

Supplementary Figure S7.5: Videos of chromothriptic medulloblastoma DAOY cells grown in 2D monolayer assay demonstrating calcium activity with various fluorescent calcium indicators.

Supplementary Figure S7.6: Medulloblastoma cells (p53 deficient, with chromothripsis, DAOY cells) in 2D monolayer assay with Lipilight560 and Rhod-2 staining for calcium imaging. Arrowheads represent membrane connections, while arrows denote thin processes.

In addition, the connectivity signature previously identified in glioblastoma (Nat Commun 2024, PMID: 38320988) is also enriched in chromothriptic medulloblastomas as compared to non-chromothriptic medulloblastomas (see results, line 348-353). TM network relevant markers such as *TTYH1* and *GJA1* are linked with clinical outcome in SHH medulloblastoma, with high expression levels significantly associated with shorter survival in the subgroup enriched for LFS medulloblastomas (see new Supplementary Figure 7.2C).

Supplementary Figure S7.2C: Expression levels of tumour microtubule network relevant markers such as TTYH1 and GJA1 are linked with clinical outcome in SHH medulloblastoma, with high expression levels significantly associated with shorter survival in the SHH alpha subgroup enriched for LFS medulloblastomas (expression data from the R2 database, log rank test)

Reviewer #2, expertise in SHH-medulloblastoma in vivo models (Remarks to the Author):

Medulloblastomas (MB) with chromothripsis results in much worse prognosis than MB without chromothripsis. Patients with Li Fraumeni syndrome (LFS) develop MBs with chromothripsis. In this manuscript, Kats et al., report that there is a marked difference between LFS and sporadic (S) medulloblastoma in intra-tumoral heterogeneity by examining 8 LFS and 5 S medulloblastoma samples by spatial transcriptomics. Although the two tumor types show some differences such as immune cell filtration and increased fraction of variance attributable to spatial location, the evidence supporting marked differences between the tumor types is weak. In addition, several comparisons need statistical analyses.

Below are specific points. Addressing these points may strengthen the author's conclusion.

1. Fig S4 does not show higher heterogeneity in the LFS group than in the S group.

We apologize for the lack of clarity. The actual quantitative analyses to underpin this statement were presented in Fig 1B and Fig S5. Briefly, we have employed two complementary analysis approaches, both of which support a higher degree of spatial heterogeneity in LFS as compared to sporadic cases. Specifically, we have compared the number of distinct regions in LFS versus sporadic samples as identified using alternative clustering approaches (now shown in revised Fig. 1B). This analysis revealed a marked difference in the number of spatially and transcriptionally distinct regions between LFS

and sporadic ($P < 0.05$, Mann-Whitney U test., one sided). Second, we have considered spatial variance component analysis (Svensson et al., Nat Meth 2018) to estimate the fraction of variance explained by the spatial location of individual Visium spots. The latter analysis provides estimates of spatial variance for individual genes, again indicating a great extent of explained variance by spatial coordinates in LFS versus sporadic. We feel that the first analysis approach based on clustering is more intuitive, and hence we have decided to now present this approach in the main text related to Figure 1. Alternative analyses and further results are now presented in Supp. Fig. S1.5.

Revised Figure 1B. Quantification of the number of transcriptionally distinct regions identified from Visium spots, normalised to the tissue size for all medulloblastomas (LFS, $n=8$; sporadic, $n=5$). Left: Regions were determined using spatially aware clustering (SpatialDE2). Right: Regions were determined using conventional Leiden clustering, not taking the spatial position of Visium spots into account. Statistical significance calculated using a one-sided Mann-Whitney U test. Mean \pm standard deviation per group is shown in black.

Supplementary Figure S1.5 C. Distribution of the fraction of expression variance explained by spatial locations (FSV) for individual genes. LFS medulloblastomas show a significantly higher FSV as compared to sporadic tumors ($p < 0.0001$, Two sided Mann Whitney).

2. Authors report that “massive rearrangements due to chromothripsis were detected in all LFS cases, in stark contrast with the relatively small number of copy number variations detected in the SHH-TP53wt tumours from the control group (Suppl. Fig. 1)”.  S4 and S5 appear to have much more rearrangements than LFS1 that has relatively few rearrangements. Therefore, it is unclear whether LFS1 is a medulloblastoma with chromothripsis and whether S4 and S5 are medulloblastomas without chromothripsis.

We thank the reviewer for raising this point, as we have now noticed that the plots for S4 and LFS1 were initially mislabeled. The plot showing chromothripsis was erroneously labeled as S4 in the previous version of the manuscript and the plot without chromothripsis was labeled as LFS1. We apologize for this mislabelling, which we have now corrected.

Re sample S5: We now provide the chromosome by chromosome copy-number plots for this patient in addition to the whole-genome view to further support the absence of chromothripsis (see Rebuttal Figure at the end of this letter).

Regarding sample S5, the chromosome by chromosome view that we have now added helps to show that there is indeed no clonal chromothripsis in this tumor.

3. It is unclear whether LFS tumors show stronger immune cell exclusion than S tumors do because Fig 1, Fig 2B, and Fig S7 do not have statistical analyses.

We have now added statistics on revised Figure 1d, Figure 2b and Figure S7 and stated which statistical test was used in the respective figure legends. The differences we report are indeed significant.

4. "Major drivers of SHH-medulloblastoma such as MYCN and GLI2 showed higher expression levels in LFS medulloblastomas but were also characterised by increased spatial expression heterogeneity, with regions characterised by high expression of one oncogene but lower expression of the other, and vice versa (Fig. 3A, Suppl. Fig. 8-10)."  Fig. 3A and Supple Fig. 8-10 do not support higher spatial expression heterogeneity in LFS than in S medulloblastoma.

Please kindly refer to the response to point #1 above, also related to the higher spatial heterogeneity in LFS as compared to sporadic medulloblastomas. Briefly, we quantified the fraction of spatial variance to measure spatial expression heterogeneity (see revised Figure 3A and Figure 1B, spatial variance for all genes). We have now added statistics and included additional examples of genes with higher spatial expression heterogeneity in LFS than in sporadic medulloblastoma but also additional analyses supporting the heterogeneity (Fig. 1B, Fig. 3A, Supplementary Figure S1.5).

Representative examples of genes that show spatial variance in LFS medulloblastomas:

Revised Figure 3A

Fig. 3B shows that "malignant cell clusters of SHH-high" and "cycling cells" co-localize but "neuronal development clusters" are located in separate regions in both LFS and S medulloblastoma.

Yes, the colocalization between high SHH and cycling cells, and mutual exclusivity with neuronal development/differentiation applies to both LFS and Sporadic medulloblastomas. However, the fractions of these cell types are different between LFS and Sporadic tumors, with more cycling cells in LFS tumors for instance (see Figure 3G-I). We have now clarified the text to address this point (see revised text, line 207-208).

5. "Collectively, our results point to increased spatial intra-tumour heterogeneity, elevated signatures of proliferation and stemness, reduced immune infiltration and differentiation in LFS medulloblastomas compared to sporadic tumours."
  Data are insufficient to support intra-tumor heterogeneity, elevated signatures of stemness, and reduced differentiation in LFS than in S medulloblastoma. Stemness is only inferred from Sox2 expression level.

Regarding intra-tumor heterogeneity (see also above response to point #1): Figure S4 shows a significantly higher number of regions with distinct transcriptional states (normalized by the number of spots), as well as higher spatial variance in LFS as compared to sporadic tumors. These results are now included in the main text Fig. 1B. We apologize for the lack of clarity.

To quantify stemness and differentiation, we have now employed a multi-gene approach described below.

Regarding elevated expression of stemness markers:

Beside the **canonical neural stem cell marker SOX2**, **markers of stemness** (e.g. DUSP9, PMID:22305567, 23973799; BEX1 PMID: 18203677) and **medulloblastoma stem cells** (e.g. CD24, PMID:30657775; GLI3, PMID:36532860) are among the most significantly differentially expressed genes, expressed at higher levels in LFS vs sporadic tumors (see new Supplementary Figure S3.4 below, Supplementary Table S3).

Supplementary figure S3.4B: Expression of stemness and differentiation markers in LFS and sporadic medulloblastomas. Statistical analysis was performed using DESeq2 with Benjamini-Hochberg multiple testing correction (** p < 0.01).

Regarding differentiation:

Among the most differentially expressed genes between LFS vs sporadic tumors with higher expression levels in the sporadic cases, there are a number of **differentiation markers**, such as PVALB (PMID: 34503995), NEUROG1 (PMID: 31801063), NPAS3 (PMID: 20603013), NRTN (PMID: 27545711) (see new Supplementary Figure S3.4, Supplementary Table S3).

Fig 1D shows the increased fraction of malignant neuronal differentiation development II in LFS, suggesting increased differentiation rather than decreased differentiation in LFS.

Fig 1D shows the fraction from all cells including tumor and normal cells.

LFS samples have more tumor cells and less normal cells than sporadic samples, so from all cells in the tissue, LFS samples have more differentiated tumor cells, but in proportion from the malignant cells, they have less differentiated cells. We have now clarified the revised captions to address this point.

Reduced immune infiltration was not shown by statistical analyses.

We have now added statistical analyses to the immune infiltration quantification (see Fig. 1D, 2B, C, E).

6. “we utilised single-cell DNA sequencing data from a primary-relapse pair of LFS medulloblastomas [15], which had been generated previously, to measure copy number profiles.”

 Reference 15 does not have single-cell DNA sequencing data.

We thank the reviewer for catching this. Unfortunately the reference list of the first version of the manuscript was incorrect in parts. The numbering of the references has been corrected. The correct reference should have pointed to this paper: reference #12 in the revised reference list.

It is unclear how two clones were detected and how their relationship was determined. The right panel of Fig. 4A shows that the clones have different CNV rather than lineage relationships. If clone A is a descendent of clone B, why clone A does not have a CNV found in clone B?

The lineage relationship between genetic clones we refer to was derived from previously generated single-cell DNA sequencing data (see Smirnov, Przybilla, Simovic et al, manuscript in revision and provided as Supplementary file, BioRxiv as reference #12). Briefly, clones A and B have a number of CNAs in common (e.g., loss on chr. 2, loss of chr. 3p and gain of chr. 3q, gains on chr. 4 and 5...).

However, additional rearrangements happened on chromosomes 8 and 12, which are detected in the daughter clone A but not in the mother clone B. In clone B, there was a small gain on chr. 12q. In clone A, the entire q-arm of chromosome 12 was lost, so the initially gained region (visible in the mother clone) appears as balanced instead of gained.

d. LFSMB1R-PDX

7. It is hard to see individual cells in Fig 4C.

We have now replaced the FISH picture in revised Figure 4 to improve the visibility.

Fig 4A shows that clone B not clone A is polyploid; however, Fig 4C shows higher ploidy in clone A.

To clarify this misunderstanding between polyploidy and aneuploidy:

hmmcopy does not call polyploid clones (we mentioned this in the caption and rephrased to avoid any confusion). Therefore, this simplified scheme shows differences in ploidy for specific chromosomes (e.g. aneuploidy on chromosome 8 in clone A) but the overall ploidy (genome doubling events) is not shown, as this is not assessed by hmmcopy.

The FISH quantification of Fig 4C shows that clone B has more diploid cells and less cells with a loss of chromosome 8, as expected.

We have now clarified the caption of Fig 4A to make this more clear.

8. “Hence, the spatial mapping of genetic clones identified phenotypic features associated with each clone, with phenotypic variation between clones molecularly related to the differences detected between LFS and sporadic tumours (c.f. Fig. 1, 3).”

 Do authors suggest that an increased number of different clones in LFS than in S medulloblastoma underlie the differences between LFS and S medulloblastoma? If so, more evidence is needed from both LFS and S medulloblastoma than evidence from one LFS sample.

This is not what we suggest, we apologize for the confusion, we have now clarified the revised text. We observe that on the one hand, we see distinct genetic clones and each genetic clone is enriched for a specific transcriptional phenotype (e.g. high proliferation, differentiation...). On the other hand, these intra-tumor differences observed between genetic clones within one individual are also present between patient groups (i.e., between LFS and sporadic, such as for instance more malignant cycling cells in LFS as compared to sporadic tumors).

Moreover, S medulloblastomas also show the spatial separation of cycling cells from differentiating cells.

Yes, the spatial separation between cycling and differentiated cells applies to both groups (LFS and S) but the proportions are different (e.g., significantly more cycling cells in LFS, see quantifications in Figure 3G-I). We have now clarified the text in the revised version of the manuscript (see revised text, line 207-208).

9. "the PDX models more closely resembled the spatial patterns observed in the corresponding human tumours than other non-matched LFS tumours (Fig. 6A, Suppl. Fig. 15), supporting the faithfulness of these in vivo models."  The spatial patterns of all LFS and S tumors look very similar except for high values for negative correlations in LFS5. Thus, it is unclear whether the PDX model specifically resembles the original human tumor.

Thank you for raising this. The main point in the context of our study is that the PDX model does capture aspects that resemble LFS versus sporadic patient samples. To more directly assess this, we have calculated the correlation between sample-specific spatial cell type correlations. This measure of transcriptional similarity indicated stronger resemblance between the PDX and LFS medulloblastomas as compared to PDX and sporadic medulloblastomas. However, we agree that any difference in correlation we detect is subtle. Although our study is also not designed to assess the degree to which PDX models reflect the primary tumors, we have extensive data (single-cell DNA and RNA sequencing) to support this (see Parra et al). This was also reported previously without spatial information but including DNA methylation analyses (e.g., Brabetz, Pfister et al, "A biobank of patient-derived pediatric brain tumor models", Nat Med 2018).

The unique angle in the context of this spatial analysis is the time series of PDX models following treatment. We have revised and put more emphasis on the time series aspect.

Supplementary Figure S6.1D: Distribution of cell type correlations between PDX and LFS or sporadic patient samples. For each pair of PDX and patient samples, the correlation between sample-specific spatial cell type correlations is shown. The correlations are higher between PDX and LFS medulloblastomas as compared to PDX and sporadic medulloblastomas (Mann-Whitney U test).

10. Authors wrote that Nes is a tumor microtubule marker. What is the evidence for Nes being a tumor microtubule marker? Nes is an intermediate filament protein that is expressed in neural progenitor cells and labels the whole cell including processes.

There is a strong body of evidence in the literature supporting Nestin as a canonical marker of microtubules. Since the initial paper by Osswald and colleagues (*Nature* 2015, PMID: 26536111), Nestin has been established as a particularly robust marker in a number of additional studies (e.g., PMID: 37335907 *NeuroOncology* 2023, PMID: 35914528 *Cell* 2022, PMID: 33320195 *NeuroOncology* 2021), including the 2019 *Nature* paper with multiple co-stainings (PMID: 31534219).

Tumor cell network integration was also identified as a stemness feature in glioblastoma, and glioblastoma cells that are interconnected by TMs show stemness properties (PMID: 33320195).

In addition to Nestin, we have also used additional markers (e.g. TOM20 as a mitochondrial marker for immunofluorescence analyses, GAP43 and connectivity signature in the Visium data).

To identify and quantify TMs, we do not rely only on the Nestin positivity itself, but we also evaluate the particular morphology of TMs (0.5-2 micron diameter, minimum 10 micron length in thin sections, minimum 50 micron length in thick sections), which provides additional confidence. We have added all details related to the quantification of TMs to the Methods section (see revised Methods, line 870-878).

11. “nestin showed a higher expression in human LFS as compared to sporadic medulloblastoma”

 Fig 8E shows that Nes expression levels vary highly in LFS group and in S group rather than vary between LFS vs. and S groups.

To test whether Nes expression is indeed higher in LFS as compared to sporadic medulloblastomas, we have now re-analyzed existing bulk RNA-seq data (n=172).

Nestin expression is significantly higher in the SHH alpha subgroup enriched for LFS (n=51) as compared to SHH beta and delta (and similar to the gamma subgroup).

High expression of nestin in medulloblastoma in the SHH alpha subgroup (enriched for LFS) is significantly linked with shorter survival.

These results are now included in the new Supplementary Figure S7.2.

Supplementary Figure S7.2: A. Nestin expression observed in bulk RNA sequencing data (R2 database, Tumor Medulloblastoma - Cavalli - 763 - rma_sketch - hugene11t dataset). Statistical analysis was done with two-tailed unpaired t-test. **B.** Kaplan-Meier survival analysis in SHH alpha subtype grouped by *NES* expression (dataset as in A).

12. Fig 1A shows extreme examples of LFS and S samples rather than representing each group.

In the revised version of the manuscript, we now show combined metrics in revised Figure 1b and 1d as well as all tumors for each group (revised Supplementary Figure S1.2, S1.4). We detect a significantly higher number of regions per spot and higher spatial variance in LFS as compared to sporadic tumors. We have now added these panels into the main figure (see revised Figure 1). Please also kindly refer to our response to specific point #1 above (page 11 of this letter) with additional analyses related to the differences in spatial heterogeneity between LFS and sporadic tumours.

Reviewer #3, expertise in SHH-medulloblastoma genomics spatial transcriptomics, chromothripsis and evolution (Remarks to the Author):

Overview:

In this work, the authors aim to learn something about what makes chromothriptic MB tumors exhibit poor prognosis relative to MB cases without chromothripsis. This is an intriguing angle of investigation, potentially revealing how chromothriptic events can shape evolutionary trajectories and tumor-cell phenotypes to yield more aggressive cancers compared with those without such rearrangements. The effects of specific chromothriptic alterations (copy number gains and structural rearrangement of genes for ex on double-minute chromosomes) on gene expression programs remains to be fully understood. Profiling of single cell and spatial transcriptome datasets has the potential to reveal how expression programs are perturbed, and how tumor cells interact with each-other and cells of the tumor microenvironment as a result. These findings would be of high interest and significance to the cancer research community.

We thank the reviewer for the positive evaluation of the significance of our study.

To achieve this, the authors set out to profile MB tumors from the same molecular subgroup and compare chromothriptic and non-chromothriptic cases. They use spatial transcriptomics to perform the majority of the work, linking to results from previously published datasets when needed.

I was initially excited by the novel application of spatial profiling to this problem, however, there are a number of significant limitations of the work presented, which prevent the original research question from being addressed. I will summarize these below in 3 main categories:

1. Cohort, and the potential significance of the results:

a. To understand what makes tumors with chromothripsis so aggressive, the authors profile a set of TP53-germline mutant (LFS) cases all with chromothripsis, and compare these to TP53-wt SHH MB (no chromothripsis). A solid rationale for the selection of this case-control strategy is not presented.

For the cases: there are SHH TP53-mut (somatic) cases with chromothripsis which could be included, not just LFS tumors.

To our knowledge, all cases (which are not too many) with suspected somatic TP53 mutations identified to date had an allele frequency of the *TP53* mutation close to 100%. The verdict is not out, yet, whether these are indeed truly somatic or rather somatic mosaicism, which would not make them a good control group. This is why we have decided against including them here.

Given the focus on chromothriptic SHH tumors (see title of manuscript and abstract), the current cohort of cases is not comprehensive, and may shed light only on LFS tumor biology rather than chromothriptic tumors more generally.

As correctly mentioned by the reviewer, we focus here on chromothriptic SHH medulloblastomas and not on chromothripsis in general.

Chromothripsis does indeed occasionally occur outside the SHH subgroup in medulloblastoma (e.g., in group 3 and group 4 medulloblastomas, mostly in the context of *MYC* amplifications) even though at a lower frequency.

However, we needed homogeneous groups to compare against each other, and the molecular differences between the SHH subgroup and group 3 or 4 are substantial, so we chose to compare cohorts from the same molecular subgroup to exclude the effect of the molecular subgroup, which is a major confounding factor, also when it comes to treatment responses and outcome.

A comprehensive study of all medulloblastoma molecular subgroups would definitely be interesting but certainly comparing apples and oranges.

For the controls: why did the authors choose to profile TP53-wt cases, rather than TP53-mut (somatic) SHH tumors without chromothripsis? A well-stated rationale is needed to understand this, and I am not convinced the comparison is appropriate.

We thank the reviewer for this comment and kindly refer back to our response to question 1a above, which addresses this comment as well.

Molecularly and in terms of treatment and survival, the 4 MB subgroups and 12 subtypes are quite distinct (see Cavalli paper Cancer Cell 2017). This is recognized in the manuscript (work on lines 167-180) and in Fig 2, as the authors perform comparisons among subgroups and subtypes. However, this is not accounted for in the cohort selection.

The LFS tumors are likely SHHalpha, while the TP53-wt patients belong to other subtypes (n=5 patients of which 3 are infants, and likely SHHbeta or gamma, while the other two

are not infants, subtype unclear). Given the heterogeneity of the control group, the study does not seem powered to address the main question of interest.

Since we had observed that the LFS tumors here are not all SHHalpha, we have attempted to match this situation in the control group as much as possible. We have now added all details on the subtype information within the 4 main molecular groups to the supplementary table (see revised Supplementary Table 1).

An additional factor that complicates the sample selection comes from technical reasons and sample availability:

For Visium FFPE, the RNA quality is critical, it is more challenging than for Visium on frozen tissue. We tested all available blocks to obtain tissue sections with good RNA quality to ensure high quality data.

The focus on SHH as one of the molecular subgroups was possible, but taking into account further subtypes within the main molecular subgroups would make it even more difficult to get enough samples per group (or this would need to be balanced against group size).

A cohort of non-chromothriptic TP53-mut tumors (SHH-alpha subtype) would provide a better comparison, as the other aspects of MB biology would be most similar. Second-best would be other SHHalpha tumors that are TP53-wt.

We thank the reviewer for this comment and kindly refer back to our response to question 1a above, which addresses this comment as well.

The Methods in this manuscript state that methylation-based classification was performed, but those results are not included, and it's unclear whether the tumor subtypes of the included cases are known.

To address this comment, we have now added more details to revised Supplementary Figure 1 and to Supplementary Table 1 to include the tumor subtypes from the methylation-based classification.

b. The PDX cohort is meant to reveal mechanisms of resistance to treatment, and 11 samples are profiled along a transition from pre-treatment to post-treatment, MRS, and regrown tumors. Major conclusions about resistance are presented in the abstract and discussion, yet the data do not support these statements. Pitfalls are that all 11 samples profiled derive from a single LFS patient, and that is not sufficient to generalize from. Importantly, the sample used for the treatment experiments is from an already-treated patient sample (although no details on that treatment are provided).

The focus of the PDX work was to dissect the spatial and temporal evolution across treatment. We agree that this explorative case study with time series needs to be extended to other PDX models in the future. Considering multiple treatment arms (3 groups of animals in our case, namely control, single treatment and combination treatment), temporal patterns (before treatment, early treatment effect, minimal residual disease and regrown tumours) as well as spatial resolution with an adequate group size and a follow-up time of 300 days for additional PDX models was beyond the scope of the current study.

We have now added details on the treatment received by the patient to the revised methods section (see line 591-592).

Why the investigators chose this strategy is perplexing, especially since a PDX line from the same patient's primary tumor was also available.

We showed in a previous study that primary medulloblastomas from this subgroup can be treated successfully (see Simovic et al, Neurooncology, PMID: 34049392). However, the treatments applied in the Neurooncology study were successful on the primary tumors but not on the matched relapsed tumors. This is why we have now set out to test treatment options (including carbon ions and combination treatments with PARP inhibitors) for the relapse setting, as there is no treatment currently available, in contrast to the primary tumor setting.

That would have at least provided a relevant (yet n-of-1) result to build upon. As such, the observation of multi-clonal tumor regrowth in the PDX (clones A and B) is without significance, since these clones have already survived therapy in the patient. Given that the supervised methods used in the manuscript can only measure the presence of the major clones A and B, there is no chance of discovering additional clones that may arise through the selective pressure of treatment. Unsupervised methods could be used (inferCNV for instance), but results would still be limited to a single and already-treated patient. The relevance and impact of the PDX work as presented is therefore quite low.

Regarding unsupervised methods:

We applied inferCNV, CopyKat and Numbat on the Visium data (also suggested by reviewer #1, see second general comment, page 2-3 of this letter). Unfortunately, in most samples only the major clone is retrieved by such methods (see rebuttal Figure below), and smaller genetic clones that we identified by matched single-cell DNA sequencing analysis are missed in the Visium data. In addition, although the inferred copy-number profiles are largely consistent with matched single-cell DNA sequencing data, the size resolution obtained by copy-number inference tools from spatial data does not allow to identify small gains and losses. In particular, chromothripsis-related effects are missed by such copy-inference analyses, which are very performant for chromosome-arm level gains and losses, but not to detect rearrangements such as copy-number oscillations due to chromothripsis. Therefore, even though we agree that in principle copy-number inference from spatial transcriptomic data is of interest, for our specific cohort and biological question it does not seem to be as informative as it has been in other contexts.

Rebuttal Figure 2: Representative example of copy-number inference using Numbat (top) with matched heatmap from single-cell DNA sequencing data (bottom). Briefly, the minor clone without loss on chromosome 8 (bottom of the heatmap) is not detected by Numbat in the spatial transcriptomics data. The inability to resolve this clone is linked to failure to capture small copy-number events, such

as oscillations due to chromothripsis on chromosome 8p, which are only visible in the single-cell DNA sequencing data from this PDX sample but not in the matched Visium data.

Taken together and despite significant effort, we were unfortunately unable to draw meaningful and robust conclusions regarding copy-number inference from the spatial transcriptomics data beyond the major clone for each sample.

However, we have now performed a variant allele frequency analysis from deep bulk sequencing (see revised Figure 6). Even though we might miss small clones in this analysis, the overall stable cancer cell fractions confirm that there is no major rearrangement of genetic clone proportions upon treatment.

Regarding sample size:

It is correct that the choice of studying treatment response and resistance over multiple time points using one model is a limitation of our study and we agree that this part of the manuscript is explorative. However, the time series increases the effective sample size, and we have leveraged the temporal resolution over the course of the treatment.

2. The quality of the data and analyses: the use of appropriate analyses, careful interpretation, and sufficient detail is lacking.

We have now added more details supporting the quality of the data and analyses (see revised Methods and github repository <https://github.com/ilia-kats/medulloblastoma-paper>).

a. There are almost no p-values or fold changes presented in the manuscript, yet strong statements like on line 152 are made throughout: “This analysis revealed a markedly higher number of distinct tissue zones in LFS medulloblastomas, indicating a higher degree of spatial heterogeneity in this subtype (Fig. 1A..”. What does markedly higher mean? Is the difference significant?

Yes, the difference is indeed significant, we have now added the bar graphs and the p values related to the number of regions per medulloblastoma (see revised Figure 1).

This lack of statistical rigour is prevalent throughout the manuscript. In this particular case, we’re directed to look at Fig 1A as support for this conclusion of higher diversity of tissue zones in LFS cases. Fig 1A shows us two “representative examples” (LFS3 and S1) with spatial clusters, rather than case vs control statistical support. Looking at the supplementary data, it seems that S1 is not actually representative of the sporadic tumors; instead, it is the most homogeneous case. In general for most analyses, the presented data does not support a strong and clear story.

We have now added statistics throughout the manuscript as well as summary plots per group (LFS vs sporadic).

Regarding intra-tumor heterogeneity:

We detect a significantly higher number of regions per spot and higher spatial variance in LFS as compared to sporadic tumors. We have now added these panels into the main figure (revised Figure 1B).

Regarding sample S1 and how representative it is for the sporadic samples: the sporadic tumours are very homogenous (see plots below: low standard deviation for the sporadic samples, all orange data points closely grouped around the mean value).

The differences between LFS and sporadic tumours are statistically significant, and we have added comprehensive quantifications of spatial heterogeneity, presented in revised Figure 1B and in Figure S1.5.

This point was also raised by reviewer #1 (specific point #1), please also kindly refer to rebuttal figures above (page 10-11 of this letter) and Figure S1.5.

Another example on line 207: “we assessed pairwise spatial correlations between all major cell types, which identified both negative and positive correlations in LFS medulloblastoma, whereas sporadic medulloblastomas were characterised by overall weaker evidence for spatial cell-type correlation, again supporting a more homogeneous spatial structure in sporadic medulloblastoma (Fig. 3C).” These plots look nearly the same. How is the above conclusion made, and how is “weaker” measured? Is the magnitude of difference significant? Can it be explained by any technical differences, such as the number of samples per group or the variable nr of spots per sample? How are those technical differences between cases and controls accounted for?

We agree with the reviewer that these differences in the spatial cell-type correlations are subtle. Even though LFS and sporadic medulloblastomas show significantly different cell type compositions (see Figure 1), correlations between cell types in space only show qualitative effects which are not significant with this cohort size. Hence, we have revised the text to tone down these findings (see line 210-212) and moved Figure 3C to the Supplementary Figures (see S3.5).

Like these examples, nearly every other comparative statement in the manuscript is descriptive rather than quantitative. Figures also fail to provide fold changes, p-values, or to offer quantitative assessments. This needs significant revision.

In response to this comment, we have now added fold-changes and p values throughout the manuscript (see revised Figures 1B-D; 2B-E; 3F, G, I; 4E-F; 5B; 6D; 7D). We have also substantially revised and extended the methods.

b. Spatial transcriptomics is a relatively new field, and analytic approaches are continually developed. While the wet lab-based techniques are adequately described in the methods, the data, QC, and analytic methods are not. I see that a github repository with code is available, but that is not a sufficient replacement for well-described methods. This currently leaves many open questions that can affect the interpretation of the results. For instance:

i. some of the LFS tumors have significantly fewer spots than other samples. How is this

accounted for when analyzing results? Fewer spots may reduce the diversity within samples, affect the QC thresholds (i.e. min spots with gene expression), unless one is analyzing LFS and sporadic cases separately – which could then increase the number of highly variable genes in LFS, and in turn introduce a technical batch effect between cases and controls. A clear description and a justification of the methodology used, and how that accounts for any batch effects is critical to include.

We do not explicitly account for the number of detected spots. However, our analyses do not depend on the number of highly variable genes. We have some analyses using spatially variable genes, but in these large samples, almost every gene is spatially variable, even if the effect size is small. For spatial clustering, we use the 2000 most highly expressed spatially variable genes in each sample. At least 2000 spatially variable genes were detected in each sample.

We use highly variable genes for constructing the k-nearest-neighbor graphs that underlie Leiden clustering and UMAP embedding. However, we use the Seurat v3 algorithm to detect highly variable genes, which calculates a normalized variance for each gene and returns the n genes with highest normalized variance (in our case, n=2000). So these analyses do not depend on the number of spots.

In response to this comment, we have now added more details to the revised methods and QC (see computational methods, from line 964).

ii. In the PDX data, a big confounder is the mouse transcriptome. The methods state that only the human reads are analyzed, however, there will still be a large effect of the human-mouse admixture. Imagine that a spot has 50% human vs mouse reads, and another has 90% human and 10% mouse. Each spot gets 50,000 reads. This leaves ~25,000 human reads in the first spot and 45,000 human reads in the second. That is nearly a two-fold difference in sampling depth of the human tumor cells and compromises the ability to detect genes with moderate/low expression in the first spot, unless accounting for this in some way. What happens with these numbers during the subsequent analysis? Is power of detection accounted for at all? Without accounting for this, any comparisons between large tumors in the PDX models and MRD will be mostly reflecting the difference in power of detection (see Fig 8B volcano plots for an example of lack of power on the MRD side).

The mouse transcriptome is indeed a big confounder in PDX data. We have now included further results and insights, showing that our method to differentiate between mouse and human reads is robust and that the power of detection is adequate (see revised Methods and rebuttal figures below). Please also kindly refer to the response to point iii below. Specifically, we examined the spatial patterning of mouse versus human read ratios. This analysis revealed no evidence that our findings are driven by the fraction of human/mouse reads across spots.

Rebuttal Figure 5: The location of human tumor cells (brown staining, STEM121 used as a specific marker of human cells in mouse tissue) matches to the location of assigned human read counts.

Rebuttal Figure 6: Total counts of human and mouse reads per Visium spot in the PDX samples (left, middle) and the ratio of human to mouse reads (right). Reads were sorted into different species using xengsort.

iii. Related to the above, Fig 6 shows that macrophages and microglia are detected and quantified in the PDX samples. Thus, either the methods are incomplete (and mouse data is indeed analyzed but not described), or there are human-mouse orthologs that are

assorted to the incorrect genome. This would bin some mouse reads into the human genome pile, and lead to annotation of microglia and macrophages. Conversely, some human reads may end up in the mouse-genome pile and not be analyzed, raising issues of false negatives in the human tumor data. 10X Genomics provides a 2-genome reference that may help with this.

There are human (patient-derived) macrophages and microglia in PDX samples. We have also shown this previously in our 10x RNAseq data (see Smirnov, Przybilla, Simovic et al, attached manuscript). Patient macrophages and microglia survive in immunocompromised mice. The reads that we analyzed are unequivocally human reads. Briefly, this is the strategy that we used to distinguish between mouse and human reads:

We applied xengsort, a fast xenograft read sorter based on space-efficient k-mer hashing (Zentgraf, Jens, and Sven Rahmann. "Fast lightweight accurate xenograft sorting." *Algorithms Mol Biol*, 2021)

The authors who developed xengsort did the comparison on mouse exomes that were captured with a human exome capture kit, 4 out of the 5 tested mouse strains have over 95% reads assigned to mouse, 1 strain has 78% reads assigned to mouse.

We ran xengsort on mouse brain Visium data from the cell2location paper (Kleshchevnikov et al., *Nature Biotechnology* 2022), 96.3% reads were correctly classified as mouse, 0.08% reads were classified as human, the rest was both, ambiguous, or neither.

iv. It seems (from related work in ref 12) that scRNAseq data from 3 of the same patients is available. Why are these not used as validation in support of the main findings? Specifically, the effect of clonal architecture (clone A and B) on transcriptional programs, the diversity of cell states and types within tumors, and cell-cell communication patterns based on ligand-receptor analyses. This seems like a missed opportunity to add rigour to the work. Additional single cell profiling of controls would be warranted.

To clarify, we have used our existing 10x scRNAseq data (e.g. for the cell type annotation) and this is indeed a valuable resource to provide additional robustness to the Visium data analyses. We have now added more details to the main text and methods to clarify this point (see line 219, line 486, line 983-990).

In the revised version of the manuscript, we have also added new analyses using the 10x scRNAseq data to address questions related to the cell types expressing specific ligands and receptors (see revised Figure 3D and Supplementary Figure S3.6).

3. Result interpretation:

a. As alluded to above, there is over-interpretation of the results throughout the manuscript, and I will not be able to list all examples. A careful re-read of (i) what the experiments aim to show, (ii) actually show, and (iii) the stated conclusion is needed in each case.

i. One example (line 312): "To test whether the distinct genetic clones may be interdependent and require each other for optimal tumour growth..." is followed by an observation of clone distribution in the cerebellum vs frontal cortex. These observations shed no light on interdependence at all. Very different experimental testing would be needed to test that. Instead, the co-occurrence may simply reflect that clone A and B are not functionally distinct in terms of their invasion phenotype. Yet the conclusion is that "the distinct genetic clones may possibly benefit from each other, also for the migration to the frontal cortex". A careful review of the initial statements that set up the experiments

conducted and how they match up to the conclusions of the experiments is needed throughout.

We agree that it would be interesting to test the potential interdependence of the clones. As the functional testing is out of scope for this manuscript, we removed our hypothesis on the potential interdependence of the clones from the revised text.

b. Insufficient analysis and/or interpretation is also noted. For instance:

i. a cell-cell interaction analysis is performed with COMMOT, and midkine signalling is highlighted as important to LFS, plus a couple of other pathways in sporadic cases. This needs more detail (and quantitative metrics), including which receptors and ligands are involved, which cells are sending signals, and which cells are receiving them. Further, what is the implication for this difference in signalling in terms of biology or therapeutics?

We have now added the quantitative metrics showing the COMMOT rankings of the ligands and receptors in each sample. The main ligand is MDK and the main receptors are NCL, SDS2 and LRP1 (see new Supplementary Table S4).

COMMOT output is based on spots, not on cells, therefore we used matched single-cell RNAseq data that we had previously generated to find the identity of the sender and receptor cells (see also above response to reviewer #1, page 5-7 of this letter).

In terms of implication for the midkine signaling for the biology or therapeutics, midkine signaling is involved in cancer cell survival and proliferation, inflammation and angiogenesis, evasion of apoptosis, cancer cell invasion and metastasis, as well as immunosuppression (ref doi.org/10.1038/s41388-019-1124-8 and 10.3389/fimmu.2023.1145300). Importantly, inhibitors of midkine signaling including antibodies, aptamers, peptides, and low molecular weight compounds, are currently under preclinical development for applications in oncology (doi.org/10.1038/s41388-019-1124-8).

ii. The tumor microtube work is done in PDX lines from one patient, and essentially identifies that TMs exist in the PDX. Yet this statement is made in the abstract: "Finally, we identified a potential role for tumour microtubes in treatment resistance in chromothriptic medulloblastoma, suggesting cell network communication as a putative target.". That seems like a rather large leap to make in terms of interpretation, and an unsupported generalization to LFS. To establish the presence of TMs in medulloblastoma, patient samples should be investigated first.

We have now performed additional experiments to show the presence of TMs in LFS and sporadic medulloblastoma patient samples (see revised Figure 7 and Supplementary Figure S7.1, n=17 patient medulloblastomas). We detected TMs in 15/17 analyzed medulloblastoma patient samples.

Are TMs present in MB samples in the primary and recurrent setting? Of relevance to this manuscript, are there differences between chromothriptic vs non-chromothriptic tumors?

We have now analyzed TMs in 11 matched pairs and showed TMs both in relapsed SHH medulloblastomas as well as in primary medulloblastomas (see new panel 7E and S7.1). We detect TMs in LFS as well as in non-LFS medulloblastoma, with 8/9 LFS medulloblastomas and 7/8 sporadic medulloblastomas showing clear TMs (see new panel 7E and Supplementary Figure S7.1). Therefore, TMs could potentially be relevant beyond the context of LFS medulloblastoma. A detailed study of the function of TMs in medulloblastoma will be needed to clarify their role in treatment resistance and whether they might offer any therapeutic vulnerability.

Is there any reason to suspect this could be the case, and serve as a rationale for this investigation? As presented, the TM work currently does not provide any meaningful insight into this type of connectivity in MB, or any specific relevance to chromothripsis. To further establish the role of TMs in treatment resistance a lot of functional work would be needed in vivo, using appropriate models, and seems out of scope for this manuscript. To further say TMs play a role in treatment resistance in chromothriptic MB would need a robust comparison to non-chromothriptic MB.

As TMs were shown to play a major role in aggressiveness and treatment resistance in other brain tumors (see Venkataramani et al, Cell 2022; Venkataramani et al, Nature 2019; Osswald et al, Nature 2015) but have never been described in medulloblastoma, we think that it is important to report this discovery, as a biological feature of medulloblastomas. We have now added an analysis of TMs in chromothriptic and non-chromothriptic MB (please refer to previous point above). This new analysis suggests that TMs might also play a role beyond chromothriptic medulloblastomas. We agree that functional studies on the role of TMs in treatment resistance are out of scope for this manuscript. However, we have now shown calcium signaling characteristic of functional TMs in medulloblastoma cells with chromothripsis (see revised Figure 7).

Other comments:

The supplemental figures other than SF1 were not labelled, making it quite difficult to match them up.

We apologize for this issue, which we have now fixed in the revised version of the supplemental figures.

APPENDIX. Chromosome by chromosome view of copy-number plots for patient S5.

170710_MethWeb0133_Core.P036_101666_oth

170710_MethWeb0133_Core.P036_101666_oth

170710_MethWeb0133_Core.P036_101666_oth

170710_MethWeb0133_Core.P036_101666_oth

170710_MethWeb0133_Core.P036_101666_oth

170710_MethWeb0133_Core.P036_101666_oth

170710_MethWeb0133_Core.P036_101666_oth

170710_MethWeb0133_Core.P036_101666_oth

170710_MethWeb0133_Core.P036_101666_oth

170710_MethWeb0133_Core.P036_101666_oth

170710_MethWeb0133_Core.P036_101666_oth

170710_MethWeb0133_Core.P036_101666_oth

170710_MethWeb0133_Core.P036_101666_oth

170710_MethWeb0133_Core.P036_101666_oth

170710_MethWeb0133_Core.P036_101666_oth

170710_MethWeb0133_Core.P036_101666_oth

170710_MethWeb0133_Core.P036_101666_oth

170710_MethWeb0133_Core.P036_101666_oth

170710_MethWeb0133_Core.P036_101666_oth

170710_MethWeb0133_Core.P036_101666_oth

170710_MethWeb0133_Core.P036_101666_oth

Reviewer #1 (Remarks to the Author):

No further comments

Reviewer #2 (Remarks to the Author):

The authors addressed most of previous comments. Addressing the following will make the findings and conclusions more convincing and clearer.

1. “ Major cancer drivers as well as microenvironment-related genes such as proinflammatory factors showed higher expression levels in LFS medulloblastomas but were also characterized by increased spatial expression heterogeneity, with regions characterized by high expression of one gene but lower expression of the other, and vice versa (Fig. 3A, Suppl. Fig. S3.1-3.4, Suppl. Table S2). This heterogeneous pattern of expression of cancer drivers and microenvironment factors might contribute to tumour cells escaping the treatment in LFS medulloblastomas, with multiple drivers active in different tumour regions.”

 Increased spatial expression heterogeneity in LFS compared with sporadic tumor is not statistically significant. Therefore, increased expression levels rather than increased spatial heterogeneity might contribute to tumour cells escaping the treatment in LFS medulloblastomas.

We note that the reported effects in differential spatial expression heterogeneity in LFS compared with sporadic tumors were statistically significant (Figure S1.5C $p < 0.0001$; Two sided Mann Whitney test). Having said this, we agree that differences in the overall expression levels most certainly also play a role, and we cannot definitively rule out that they (at least in part) explain these variance effects.

We generated scatter plots showing expression levels versus spatial variance across all samples to evaluate to what extent the expression level is a possible driver of variance effects. There is a weak effect, which is mostly driven by a small number of genes (see Rebuttal Figure 1 below). To make our findings more robust, we have now adjusted the estimation procedure to account for this effect as much as possible and corrected for this effect: We subtracted the LOESS-predicted Fraction of Spatial Variance or FSV (as a function of average gene expression) from the original FSV estimate. To account for negative values, the minimal value of the new estimate was subtracted from all values. This resulted in what we termed corrected FSV. The original and corrected FSV were highly correlated (Rebuttal Figure 2) and the corrected FSV was still higher in LFS samples as compared to sporadic samples for the genes shown in main Fig. 3A (Rebuttal Figure 3). We also note that among the genes shown in Fig. 3A, only *PDGFRB* is differentially expressed between LFS and sporadic tumors, showing that the levels of spatial expression heterogeneity are not necessarily linked with the expression levels themselves. We have toned down the corresponding text section to reflect the contribution of overall expression levels:

“This heterogeneous pattern of expression of cancer drivers and microenvironment factors, together with differences of the overall expression levels, might contribute to tumor cells escaping the treatment in LFS medulloblastomas, with multiple drivers active in different tumor regions.”

Rebuttal Figure 1. Fraction of spatial variance (FSV) as a function of average counts per gene (log1p-transformed). Top: all samples, gray line shows a linear regression. Bottom: Individual for each sample, gray line shows a LOESS fit. Expression levels are not the main driver of spatial variance, but we do observe a weak effect for a subset of genes.

Rebuttal Figure 2. Corrected FSV (corrected for expression levels) plotted against original FSV. Gray line is the identity. The corrected FSV was calculated by subtracting the LOESS-predicted FSV value from the original FSV (using the LOESS fits from Rebuttal Figure 1 bottom). To account for negative values, the minimal value was subtracted from all values.

Rebuttal Figure 3. Corrected FSV of selected genes shown in Figure 3A in LFS and sporadic medulloblastoma. Even after correction, the FSV is still higher in LFS as compared to sporadic medulloblastomas.

2. “Sporadic medulloblastomas were characterized by overall slightly weaker evidence for spatial cell-type correlation, again suggesting a more homogeneous spatial structure in sporadic medulloblastoma (Suppl. Fig. S3.5).”

 The number of sporadic tumor samples is less than that of LFS. Would this contribute to the overall slightly weaker evidence for spatial cell-type correlation in sporadic tumor?

The figure shows the arithmetic mean across samples, and hence is not very prominently affected by the difference in sample size between groups. We have revised Suppl. Fig. 3.5 to enable a direct assessment of the individual samples, which supports that these trends are somewhat generic. Having said this, the in general small sample size limits this analysis, enabling qualitative but only to a limited extend quantitative conclusions; we have toned down the corresponding text section:

“Sporadic medulloblastomas were characterised by overall slightly weaker evidence for spatial cell-type correlation, again suggesting a more homogeneous spatial structure in sporadic medulloblastoma, even though the observed effect was modest (Suppl. Fig. S3.5).”

3. “ In line with the GAP43 expression patterns, quantification of the tumour microtubule area per cell identified significantly larger tumour microtubule areas per cell in the untreated relapsed PDX as compared to the matched untreated primary tumours, and also showed a decrease in the tumour microtubules at the minimal residual disease stage and an increase in the regrown tumours (Fig. 7D). Therefore, tumour microtubules might potentially play a role in treatment resistance in LFS medulloblastoma.”

 If tumor microtubules play a role in treatment resistance, shouldn't they be increased in residual disease stage to provide resistance?

Our data show an initial decrease in the tumour microtubule area upon treatment (and the treatment is very successful to reduce the overall tumour burden at least initially) and then an increase in the regrown cells after treatment. In glioblastoma, the formation of tumour microtubules contributes to recurrence after surgery and resistance against radio- and chemotherapy (PMID 28419303). Hence, it is very well plausible that the network may be damaged by the treatment but that residual, connected cells regrow this network.

 The p-value for tumour microtubule areas per cell in the untreated relapsed PDX as compared to the matched untreated primary tumours is not shown.

We have now added the p-value to revised Figure 7 for the comparison between the untreated relapsed PDX with the matched untreated primary PDX tumours. We observe a significant increase from primary to relapse:

Primary PDX (control) Vs Relapse PDX (control):

$p = 0.0454$ (One-tailed t-test, as we assume an increase at relapse)

Hence, this supports a potential role for tumour microtubules in resistance in medulloblastoma.

Please also kindly refer to point #6 below regarding the use of the statistical methods.

4. "The tumour microtubule marker nestin showed a significantly higher expression in the medulloblastoma subgroup enriched for LFS as compared to sporadic medulloblastoma,"

 Nestin expression is not significantly different between LFS and sporadic tumor.

This seems to be a misunderstanding.

In the first part of the sentence, we refer to Nestin expression that is significantly different between groups in bulk RNA sequencing data (see Figure S7.2A, re-analysis of data from the R2 database, the SHH alpha subgroup is enriched for LFS tumours). In the second part of the sentence, we refer to the Visium data, where the difference is indeed non-significant with this group size.

We rephrased the corresponding text section with the aim to improve clarity:

"The tumour microtubule marker nestin showed a significantly higher expression in the medulloblastoma subgroup enriched for LFS (SHH alpha, $n=51$) as compared to sporadic medulloblastoma, as well as a significant association with overall survival (**Suppl. Fig. S7.2A-B**, bulk RNA sequencing; this was also true for other tumour microtubule network relevant markers such as *TTYH1* and *GJA1*, see **S7.2C**) and also a widespread expression throughout tumours (**Suppl. Fig. S7.3-7.4**, $n=8$ Visium LFS samples, difference to the 5 sporadic samples non-significant).

5. What is "mean posterior Nes expression" in Fig. S7.4?

Cell2location is a Bayesian deconvolution model and as such gives rise to estimated posterior distributions for all model parameters, including the level of expression of individual genes in groups of cell types. The figure shows the mean of the estimated posterior distribution of Nestin expression in malignant cells. We have clarified this in the revised figure legend.

6. Statistical methods used may need to be reconsidered. For example, in Figure 1B, why one-sided not two-sided test was used? For Fig. 7D, statistical methods to compare multiple groups not t-test should be used; there are more than 2 groups to compare.

In Figure 1B, our hypothesis was an increased heterogeneity in LFS as compared to sporadic tumours, hence a one-sided test is appropriate.

For Figure 7D, we have now employed an Ordinary one-way ANOVA (Dunnett's multiple comparison test) to compare the three experimental groups to the control condition (within the relapse PDX model).

Dunnett's multiple comparison test

Adjusted p values:

Relapse PDX (control) Vs Early effect: $p = 0.4978$

Relapse PDX (control) Vs Minimal residual disease: $p = 0.0226$

Relapse PDX (control) Vs Regrown: $p = 0.3587$

For the comparison between primary PDX (control) and relapse PDX (control) we used a t-test (please see point #3 above) as it is a comparison between different PDX models.

Reviewer #3 (Remarks to the Author):

The authors have addressed many of the comments in the first review, especially those related to cohort selection (case vs control sample and patient). I have no further major concerns on that front. The calcium signalling work presented for the TMs is also a valuable addition to the manuscript, and should be of interest to those in the brain tumor field.

However, there are still elements of the analysis (specifically for Visium data) and result interpretation which remain a concern, and need further attention, as I note below. Finally, some of the wording in the manuscript requires changing, so as not to give the wrong impression to readers.

Line 74: "We conducted temporal profiling of 11 patient-derived xenografts from chromothriptic medulloblastomas, covering the transition from the minimal residual disease stage to treatment-resistant regrown tumours."

- "11 patient derived xenografts" is not the same as "11 samples from a patient derived xenograft", please change this wording, as it misleads readers right at the outset in thinking there are 11 independent PDXs.

The reviewer is correct. We have revised this section and written:

"11 samples from patient-derived xenografts from a patient with chromothriptic medulloblastoma"

This new wording reflects the fact that we established xenografts from the primary tumour as well as from the matched relapse tumour, it is not just one patient derived xenograft model.

Line 76: "In chromothriptic medulloblastoma, an ecosystem of cells from multiple genetic clones resisting treatment and leading to relapse highlighted the importance of multi-clone interplay."

- Please remove "the importance of multi-clone interplay" from the abstract. This is not supported by data or analyses in the manuscript as far as I can tell. In fact, the main text has already been

corrected to remove the statement that clone interplay plays a role in invasion, but it also needs to be changed in the abstract.

We have revised the abstract as follows:

“Paediatric medulloblastomas with chromothripsis are characterised by high genomic instability and are among the tumours with the worst prognosis. However, the determinants of their aggressiveness and the molecular makeup of chromothriptic medulloblastoma are not well understood. Here, we applied spatial transcriptomics to profile a cohort of 13 chromothriptic and non-chromothriptic medulloblastomas from the same molecular subgroup. Our data reveal a higher extent of spatial intra-tumour heterogeneity in chromothriptic medulloblastomas, which is associated with increased proliferation and stemness, but lower immune infiltration and differentiation. Spatial mapping of genetic subclones of the same tumour identify a regionally distinct architecture and clone-specific phenotypic features, with distinct degrees of differentiation, proliferation and immune infiltration between clones. We conducted temporal profiling of 11 samples from patient-derived xenografts from a patient with chromothriptic medulloblastoma, covering the transition from the minimal residual disease stage to treatment-resistant regrown tumours. In chromothriptic medulloblastoma, an ecosystem of cells from multiple genetic clones resist treatment and lead to relapse. Finally, we identified tumour microtubes in chromothriptic medulloblastoma, calling for exploration of cell network communication as a putative target.”

Fig 1C – “Comparative analysis of the cell type composition for a representative LFS (top) and sporadic medulloblastoma (bottom). Left, dominant cell type for each spot for representative LFS and sporadic medulloblastoma. Right, bar graphs displaying the cell type abundance for each region as in panel A.”

- The sporadic case selected for the figure (S1) is not representative, please select another one of the 5 sporadic cases presented in Fig S1.6.

We have now replaced the sporadic case S1 by S2 and revised Figure 1 accordingly.

Line 181 – “This deconvolution analysis indicated less CD8 positive T-cells in tumours displaying somatic mutations in TP53, suggesting a possible link between p53 dysfunction and immune exclusion, beyond the context of genetic predisposition (Fig. 2E, Suppl. Fig. 2.1D).”

- The summary of CD8 cell presence pan-cancer presented in Fig 2E seems very different than the cancer-specific results shown in Fig S2.1D left panel. There, only LAML looks different, while DLBC has the opposite pattern (more CD8 T-cells in the TP53-wt setting). Other brain tumors in this analysis (GBM, LGG) show no apparent difference in CD8 cell abundance with genotype. The number of TP53-mut cases per tumor type likely matters a lot here, so the statistics per cancer type are important to present. The TP53-mut CD8 T-cell association is likely very tumor type-dependent (or related to other immune-relevant contexts), and the current Fig 2E does not capture that at all. Can the Figure S2.1D panel be added to the main Fig 2E, along with p-values and patient/sample numbers.

We have now added panel S2.1D to main Figure 2E. We have highlighted the tumor types with significant effects. We now provide all sample numbers in supplementary figure S2.1E, as we feel that this information would take too much space in the main figure.

Figure 2E. Less T cells in *TP53* mutant tumours (reanalysis of a pan-cancer cohort, deconvolution from bulk RNAseq ref33, $n=8,955$). Statistical significance in panel E was calculated using Mann-Whitney U-test. Boxes show the interquartile range (IQR), line indicates the median, whiskers indicate 1.5x IQR.

Cancer type	Number of patients		Mann-Whitney U-test (p value)
	TP53 mutated	TP53 wild-type	
ACC	17	63	0.66259
BLCA	236	218	0.35514
CESC	27	285	0.29867
CHOL	6	39	0.33778
COAD	262	267	0.99453
DLBC	5	43	0.02810
ESCA	164	25	0.27954
GBM	63	116	0.22359
HNSC	447	160	0.00000
KICH	37	56	0.09358
KIRC	16	597	0.82374
KIRP	8	315	0.47929
LAML	7	144	0.00552
LGG	298	272	0.98579
LIHC	126	300	0.68421
LUAD	322	288	0.08163
LUSC	480	90	0.78727
MESO	14	73	0.63167
OV	249	136	0.77601
PAAD	107	76	0.00442
PCPG	1	186	0.24536
PRAD	66	488	0.16984
READ	116	64	0.08709
SARC	90	177	0.15032
SKCM	87	397	0.88458
STAD	205	206	0.00446
TGCT	1	155	0.92922
THYM	4	118	0.61989
UCS	55	5	0.43690
UVM		80	NA

Supplementary figure S2.1E. Number of patients per tumour type and *TP53* status in the re-analyzed pan-cancer cohort. The statistical comparison for CD8 positive T cells was performed using Mann-Whitney U-Test and p values for each tumour type are shown in the table.

Fig 3A and S3.1-3 have no p-values. A split violin plot might be a more effective way to present this data.

In Figure 3A, the differences are not significant, as stated in the legend.

In Figure S3.1-3, the P-values are displayed (p=0.0198, p =0.0017, p=0.0742 after multiple testing corrections, respectively).

We have revised the figures to clarify which groups are compared. We feel that keeping the individual (sample) level information represented is preferable compared to aggregating the data.

Line 195: “Markers previously associated with putative medulloblastoma stem cells such as SOX2 showed significantly higher expression levels in LFS medulloblastomas, whereas differentiation markers were expressed at significantly lower levels (Suppl. Fig. S3.4, S6.8, Supplementary Table S3).”

- It looks like the Visium DEG analysis was performed with DESeq2 using a pseudo-bulk approach, such that every gene has averaged expression across all spots in each sample. This doesn't take into account the tumor-normal admixture. Yet it is already established in the manuscript that the sporadic cases have more normal cells, and given this systematic difference between LFS and sporadic cases, one cannot confidently make the claims above (claims which are also repeated and emphasized in the abstract and discussion, and should be toned down). The differences observed could simply be due to the different levels of expression of these genes in normal cells.

We agree that this could be an effect. In the revised analysis, we have now used the fraction of normal cells as covariate for DESeq, which indicates that this covariate explains some of the effect. We have adjusted the results to highlight markers that are significant when using the fraction of normal cells as a covariate and we have toned down this statement:

“Markers previously associated with putative medulloblastoma stem cells such as SOX2 showed significantly higher expression levels in LFS medulloblastomas, whereas differentiation markers were expressed at significantly lower levels (Suppl. Fig. S3.4, S6.8, Supplementary Table S3). However, despite using the normal cell fractions as covariates, we cannot formally exclude that these differences may partially be due to the different levels of expression of these genes in normal cells.”

- A great example of this issue is *Ndr2*, presented in Fig S3.1 as a gene more highly expressed in sporadic cases. The S5 tissue shows a pattern that is reminiscent of the cerebellar folding. Indeed, when looking in the mouse brain atlas for this gene's expression based on ISH (in situ histochemistry; <https://mouse.brain-map.org/gene/show/29546>), we see the same pattern for *Ndr2*. (note that corresponding spatial human data from the brain atlas was not available for this gene, but the is expected to be the same).

My conclusion here would be that the sporadic cases have a different level of expression of *NDRG2* because of differences in the normal cell types captured within the Visium sample, and not because the sporadic MB tumor cells are more differentiated. I am concerned that the analyses as currently presented do not sufficiently account for this aspect of the data, and the consequent conclusions are therefore potentially incorrect. Can the single cell data be used to more conclusively support the current claims? i.e. in tumor cells only, show that these genes are different, and present those along with the spatial data if there is agreement. Outlier samples, like S5 above, should have *Ndr2* expression in normal cells. Cell annotation is important, and even though another manuscript describing that data is in bioRxiv (since 2021, and presumably in a lengthy revision process), it is important for this manuscript to show or include some of the cell type annotation results, and certainly to briefly review the annotation approach in the methods.

In the revised analysis, we now account for the fraction of normal cells as a covariate when testing for differential expression (using DESeq; please kindly refer to the previous comment). *NDRG2*, as other differentiation markers, is still significant (more highly expressed in the sporadic cases, see revised Supplementary Table S3).

As suggested, we have also used the single-cell RNA-seq data to address this question, and shown that *NDRG2* is also expressed in subsets of malignant cells:

Nuclei extracted from LFS medulloblastomas PDX from LFS medulloblastomas

Rebuttal Figure 4. *NDRG2* is also expressed in subsets of malignant cells (scRNAseq data). The cell type annotation from Figure S3.6 is also shown for comparison.

As we only have matched single-cell RNA-seq data for the LFS samples but not for the sporadic ones, we cannot directly compare the expression of *NDRG2* between LFS and sporadic in malignant cells unfortunately.

As additional control, we also conducted a DEG analysis using filtered Visium spots with a malignant cell fraction exceeding a specific threshold value. We observed a high correlation between estimated log-fold changes in the original and the revised, more conservative analysis, and a large intersection of the sets of significant genes. The sets of differentially expressed genes identified only in the original or only in the new analysis grew larger with increasing threshold, with the highest tested threshold of 0.7 in particular identifying some genes with very large effect sizes. However, it is unclear if these differences in results are due to increasing purity of the malignant cells or artifacts due to decreasing total read count in the pseudobulk analysis and therefore increasing noise.

Altogether, these results strongly suggest that variation in Visium spot purity is not the major driver of the reported DEG genes and signatures.

Rebuttal Figure 5. Log-fold gene expression changes estimated by DESeq2 from the analysis presented in the paper compared to an analysis including only Visium spots with a malignant cell fraction above a threshold. The threshold is indicated at the top, colors indicate whether the respective gene is classified as significant at FDR 0.1 in both analyses, one of them, or none.

Regarding cell type annotation, it is shown in Fig. S3.6. We have now added all details on the cell type annotations in the revised Methods (see Methods, “Visium cell type deconvolution”).

In addition, we have performed experimental validation of the expression of the differentiation marker NEUROD1 in subpopulations of malignant cells in medulloblastoma:

Rebuttal Figure 6. Representative immunofluorescence analysis of medulloblastoma tissue. NEUROD1 is a marker of differentiated cells, GLI2 and Ki67 are expressed at high levels in SHH medulloblastoma cells. A subset of malignant cells expresses NEUROD1.

Line 272: “Specifically, we selected PDX models from the relapse biopsy, and focused on combinations of particle radiation with PARP inhibitors (Fig. 5A). We showed previously that such combination treatments were successful in the context of primary chromothriptic medulloblastomas17.”

- It would be useful to rationalize the selection of this particular treatment (is it the only rational choice? The best choice in some way), and its application to recurrent disease (to put it in context with the reference regarding it’s use in primary disease).

We have now revised the text to add more explanations about the selection of the treatment:

“Specifically, we selected PDX models from the relapse biopsy, and focused on combinations of particle radiation with PARP inhibitors (Fig. 5A). Conventional photon radiotherapy and DNA-damaging chemotherapy are not successful for these patients and increase the risk of secondary malignancies. Our previous work showed that the pronounced homologous recombination deficiency in these tumors offer vulnerabilities that can be therapeutically utilized. In particular, combining PARP inhibitors and carbon ions was successful in the context of primary chromothriptic medulloblastomas”.

Line 281: “To evaluate to which extent the heterogeneity observed in LFS patient medulloblastomas contributes to tumour relapse, we first compared the PDX models to the matched patient tumours. Notably, the two-dimensional neighbourhood structures from patient tumours were qualitatively preserved in matched PDX, indicating that these models globally reflect cellular interactions from

human medulloblastomas (Suppl. Fig. S6.1A). Additionally, the PDX models more closely resembled the spatial patterns observed in the LFS medulloblastomas than sporadic tumours (Suppl. Fig. S6.1A-D), supporting the overall faithfulness of these in vivo models.”

- The work described in this paragraph does not “evaluate the extent to which heterogeneity contributes to tumor relapse”. It presents a qualitative interpretation of the neighbourhood analysis plot.

We have now rephrased this paragraph by removing the first part of this sentence:

“We first compared the PDX models to the matched patient tumours. Notably, the two-dimensional neighbourhood structures from patient tumours were qualitatively preserved in matched PDX, indicating that these models globally reflect cellular interactions from human medulloblastomas (Suppl. Fig. S6.1A). Additionally, the PDX models more closely resembled the spatial patterns observed in the LFS medulloblastomas than sporadic tumours (Suppl. Fig. S6.1A-D), supporting the overall faithfulness of these in vivo models.”

- Importantly, the PDX data is problematic here in the sense that mouse cells are not accounted for. Are the endothelial cells, meningeal cells, etc (shown in figure S6.1, and mentioned in the main text) human? Or mouse? I assume they are human from the methods and the author response, in which case I find the relevance confusing. For macrophages and microglia, the relevance can be argued (though certainly mouse microglia and macrophages will likely outnumber the remaining human cells). However, there are also human endothelial cells shown in the same neighbourhood analysis, and these would presumably also be surviving in the PDX model – but likely not forming functional vessels. In that case, would we expect them to play the same role in a functional cancer ecosystem? (i.e. are they forming functional vessels in the mouse brain? That would be a surprising finding indeed.) Instead, I expect that mouse endothelial cells / vessels would serve the corresponding role, but those are not investigated here. Yet, despite the critical importance of the mouse cells in the growth of the PDXs, the correlations calculated and shown in Fig S6.1C are from the human data only, and include cell types we can't reasonably expect to be relevant in the PDX context. Overall, it seems a stretch to state that: “these models globally reflect cellular interactions from human medulloblastomas”, especially since the majority of the TME (mouse) has not been surveyed.

o Can the authors change the neighbourhood analysis to either focus on the tumor cells, or to include the mouse TME. Alternatively, do not overstate the similarity in the TME between mouse and human.

- Please specify species (mouse vs human) in figures and text that reference non-malignant cells

- The reviewer is correct that of course our analysis should be considered in the context of the mouse TME, which may introduce additional variation.
- At present, we do not focus on the mouse TME and assume that the immune component of the mouse TME can be ignored, which is motivated by the immune deficiency of these mice.
- Most of the spots are binary (either mouse or human reads) and we would not have the statistical power to perform such an analysis of mouse/human cell interactions. Most spots covered by the tumor contain so few mouse reads (<3000 reads) that these spots are either discarded during quality control or the cell type deconvolution cannot be assumed to be reliable. We clarified in all figures that the cell types shown are human cell types exclusively (tumor and non-tumor).
- Having said this, we toned down the corresponding text to reflect the fact that the mouse TME will be an area of future work:

“Even though a deeper analysis of the mouse tumor microenvironment will be necessary for a fine-grained comparison of unique and common patterns between human medulloblastomas and these models, the latter globally reflect cellular interactions from human medulloblastomas.”

Fig 7B Visium data-based DEG analyses: there is an unaddressed issue with depth of sampling, as brought up in my previous review comments. Currently, the volcano plots in Fig 7b are asymmetric, clearly indicating that the MRD samples are underpowered for gene expression quantification. See my original comment, with the example of high vs low tumor cell density. The low tumor cell density spots (MRD) will only capture the most highly expressed human genes, and suffer from sparse signal in a typical DEG analysis (because most genes will be signal dropouts). Thus, your biological variable (MRD vs non-MRD) is completely confounded with a key technical variable (read depth per gene). In our own work with Visium data with similar depth issues, we have applied Aldex2 to solve the imbalanced sampling problem. I recommend that you explore this approach with your data. All the statements that conclude some genes are downregulated at the MRD stage would need revising based on a more powered analysis.

Line 348: “Genes previously associated with a tumour microtubule connectivity signature in glioblastoma were downregulated at the minimal residual disease stage (enrichment tests for tumour microtubule genes in the Visium data, hypergeometric test: enrichment of TM-high genes in downregulated genes, tm_high enrichment pval: $1.63e-08$ and TM-low genes in upregulated genes, tm_low enrichment pval: 0.008).”

- This is a great example of the type of conclusion that cannot be made with the current analysis. It is more likely that this result is due to the sparse human transcriptome sampling in the MRD setting. This claim is not currently supported by the results presented.

The reviewer highlights an important point. Indeed, the MRD stage is associated with smaller tumors and hence lower numbers of locations and potentially also lower sequencing depth of human reads, although most sequenced spots have a comparable sequencing depth in the different samples (Rebuttal Figure 7). DESeq2 accounts for both possible sources of confounding - number of replicates (spot count) and sequencing depth. DESeq2 uses all samples jointly to estimate gene-wise dispersion, and it estimates log-fold changes as coefficients of a generalized linear model, which naturally handles different numbers of replicates per group. In addition, DESeq2 uses a negative binomial likelihood function and it estimates size factors to account for sequencing depth.

To empirically demonstrate the robustness of our analysis, we also conducted additional analyses. First, we downsampled the Visium spots such that each sample included the same number of spots. The estimated log-fold changes were highly correlated between the original analysis and the one using downsampled spots (Rebuttal Figure 8). Moreover, most genes that were significant in the original analysis were still significant in the new analysis. Only few genes were only significant in the original analysis, presumably due to the original analysis having more power due to the larger number of replicates available, and these genes vastly outnumbered genes that were significant in the new but not the original analysis. Finally, this analysis yielded the same type of asymmetric volcano plot as the original analysis (Rebuttal Figure 9).

Next, we focused on the sequencing depth effect. Based on Rebuttal Figure 5, we repeated the DEG analysis only with Visium spots that had more than 4,913 total human sequencing reads (equivalent to $8.5 \log_{10}$ -transformed counts, threshold defined based on the distribution in rebuttal figure 7). This is valid, since we know that the separation of human and mouse reads is highly accurate. The results were again very similar to the original analysis: A high correlation of the estimated log-fold changes, a large overlap in the identified differentially expressed genes, and the same type of asymmetric volcano plot (Rebuttal Figures 10 and 11).

Rebuttal Figure 7. Total number of human reads per spot (log1p-transformed) plotted for each Visium spot in the three categories used for DEG analysis in Fig. 7B.

Rebuttal Figure 8. Comparison of a DEG analysis using the same number of Visium spots for each condition to the original analysis. The estimated log-fold changes are highly correlated between the original analysis and the one using downsampled spots.

Rebuttal Figure 9. Volcano plots for a DEG analysis using the same number of Visium spots for each condition.

Rebuttal Figure 10. Comparison of a DEG analysis using only Visium spots with a total count of human sequencing reads greater than 4913 to the original analysis.

Rebuttal Figure 11. Volcano plots for a DEG analysis using only Visium spots with a total count of human sequencing reads greater than 4913.

Reviewer #3 (Remarks to the Author):

The authors have satisfactorily addressed the concerns raised. I have just a few very minor comments left relating to the T cell figures:

1. Please make table S2.1E an excel supplemental table. This indicates case numbers and pvalues for TP53+ and – patients in various cancer types. Please add the fraction of T-cells in, or the fold change (or both). Also, this table describes data plotted in the main figure 2E (CD8 T cells). S2.1D is the fraction of regulatory T cells. Please add another table with these corresponding numbers. The figure legend in this case also needs updating (currently states T cells, not regulatory T cells).

We have created an excel supplementary table S3 for CD8 T cells based on the previous figure S2.1E. As suggested, we have added to this table the fraction of T-cells and the fold change.

For figure S2.1D (fraction of regulatory T cells) we have added another table (supplementary table S4) with the same information (number of patients analyzed per tumour entity, fraction of T regs, fold change and the Mann-Whitney U-test (p values) statistical comparison between *TP53* mutated and *TP53* wild-type tumors, performed based on the fraction of T regs.

The legends of figure 2 and supplementary Figure S2.1. have been updated accordingly.

2. Fig 2E – only a subset of the significant results are included, although the figure legend states that all significant values are labelled. This is not the case, as LAML has significantly higher CD8 T cell fractions in the TP53-wt cases – this should be annotated on the plot.

We have annotated the missing significant result for LAML on the plot (Fig 2E).

3. Fig S2.1D – add pvalues on the plot (for all sig differences), and provide a supplementary table with the relevant info.

We have added information on p values for all significant differences to the S2.1D plot and provided a supplementary table S4 with the relevant info.

4. “To test whether lower immune infiltration might be a feature of TP53 mutant tumours beyond medulloblastoma and possibly in the context of sporadic TP53 mutations as well, we re-analysed a pan-cancer cohort of 8,955 patients. This deconvolution analysis indicated less CD8 positive T-cells in tumours displaying somatic mutations in TP53, suggesting a possible link between p53 dysfunction and immune exclusion, beyond the context of genetic predisposition (Fig. 2E, Suppl. Fig. 2.1D and E).”

◇ This statement seems to be an over-reach. At most, the deconvolution analysis shows that in a minority of tumor types, there are lower CD8 T-cell counts, but in the majority of tumor types there is no difference. Please revise the conclusion made above to more accurately reflect the results. There is probably some interesting biology to follow-up on in terms of the biology found – i.e. why do a handful of specific types of cancers have this relationship with TP53, but not others? Although this is outside of the scope of the main manuscript message, it could be interesting to speculate.

We have rephrased this conclusion to tone down the result:

“This deconvolution analysis indicated less CD8 positive T-cells in tumours displaying somatic mutations in TP53 **in a subset of tumor types**, suggesting a possible link between

p53 dysfunction and immune exclusion, beyond the context of genetic predisposition”.

Reviewer #3 (Remarks on code availability):

The code is sufficient to reproduce the figures in the manuscript. However, the code is not documented at all.

We have now documented the code:

<https://github.com/ilia-kats/medulloblastoma-paper>